# A cell-type-specific error-correction signal in the posterior parietal cortex

Jonathan Green[1✉], Carissa A. Bruno[1,3], Lisa Traunmüller[1,3], Jennifer Ding[1,3], Siniša Hrvatin[1,2], Daniel E. Wilson[1], Thomas Khodadad[1], Jonathan Samuels[1], Michael E. Greenberg[1] & Christopher D. Harvey[1✉]

Neurons in the posterior parietal cortex contribute to the execution of goal-directed navigation[1] and other decision-making tasks[2–4]. Although molecular studies have catalogued more than 50 cortical cell types[5], it remains unclear what distinct functions they have in this area. Here we identified a molecularly defined subset of somatostatin (Sst) inhibitory neurons that, in the mouse posterior parietal cortex, carry a cell-type-specific error-correction signal for navigation. We obtained repeatable experimental access to these cells using an adeno-associated virus in which gene expression is driven by an enhancer that functions specifically in a subset of Sst cells[6]. We found that during goal-directed navigation in a virtual environment, this subset of Sst neurons activates in a synchronous pattern that is distinct from the activity of surrounding neurons, including other Sst neurons. Using in vivo two-photon photostimulation and ex vivo paired patch-clamp recordings, we show that nearby cells of this Sst subtype excite each other through gap junctions, revealing a self-excitation circuit motif that contributes to the synchronous activity of this cell type. These cells selectively activate as mice execute course corrections for deviations in their virtual heading during navigation towards a reward location, for both self-induced and experimentally induced deviations. We propose that this subtype of Sst neurons provides a self-reinforcing and cell-type-specific error-correction signal in the posterior parietal cortex that may help with the execution and learning of accurate goal-directed navigation trajectories.

Recent efforts have catalogued over 50 molecularly defined neuronal cell types in the cerebral cortex[5]. Emerging studies have started to characterize the functions of broad cell type classes, such as Sst and parvalbumin neurons, for sensory processing in sensory cortices[7–9] and, to a lesser extent, during cognitive computations[10–14]. However, little is known about the subtypes that comprise these classes. We investigated this topic in the posterior parietal cortex (PPC) in the context of goal-directed navigation, a task for which the PPC is required[1]. Individual PPC neurons activate sequentially during navigational trajectories and at specific combinations of spatial position and navigational movements, for example, when turning left at an intersection[1,15,16]. Collectively, data indicate a role for the PPC in planning and guiding navigational actions, among other important functions[17–21]. However, little is known about how specific cortical cell types, or even broad cell classes, contribute to these functions. Using a viral tool[6], we found that a subtype of Sst neurons in the PPC activate synchronously as mice course-correct for deviations in their navigational trajectories, highlighting an error-correction signal in a specific cell type.

## Molecular profile of Sst44 cells

We previously identified an enhancer (*Sst44*) that drives gene expression specifically in a subset of cortical Sst cells (referred to here as Sst44 cells)[6]. To characterize this subset, we used single-cell assay for transposase-accessible chromatin using sequencing (ATAC-seq) to measure genomic accessibility across inhibitory neurons as a proxy for enhancer activity. The *Sst44* enhancer was predominantly accessible in one cluster of Sst neurons that was positive for *Calb2* and *Hpse* and a small cluster that was positive for *Chodl* (Fig. 1a,b and Extended Data Fig. 1a). When packaged in an adeno-associated virus (AAV), this enhancer drove expression in *Calb2*+, *Hpse*+ and *Chodl*+ Sst cells, as assayed using in situ RNA hybridization (Fig. 1c–f and Extended Data Fig. 1b). As *Chodl*+ Sst cells are very rare, the majority of Sst44 cells were *Calb2*-positive and *Hpse*-positive (Fig. 1e). Consistent with these subtypes[22], Sst44 cells were mainly located in layers 2/3 and 4 and, on the basis of single-cell reconstructions, sent processes to layer 1 (Fig. 1g and Extended Data Fig. 1c,d). Below, we focus on Sst44 cells in layer 2/3, which are mostly *Calb2*-positive (Extended Data Fig. 1d (shaded region)).

[1]Department of Neurobiology, Harvard Medical School, Boston, MA, USA. [2]Present address: Whitehead Institute, MIT, Cambridge, MA, USA. [3]These authors contributed equally: Carissa A. Bruno, Lisa Traunmüller, Jennifer Ding. ✉e-mail: jonathan_green@hms.harvard.edu; harvey@hms.harvard.edu

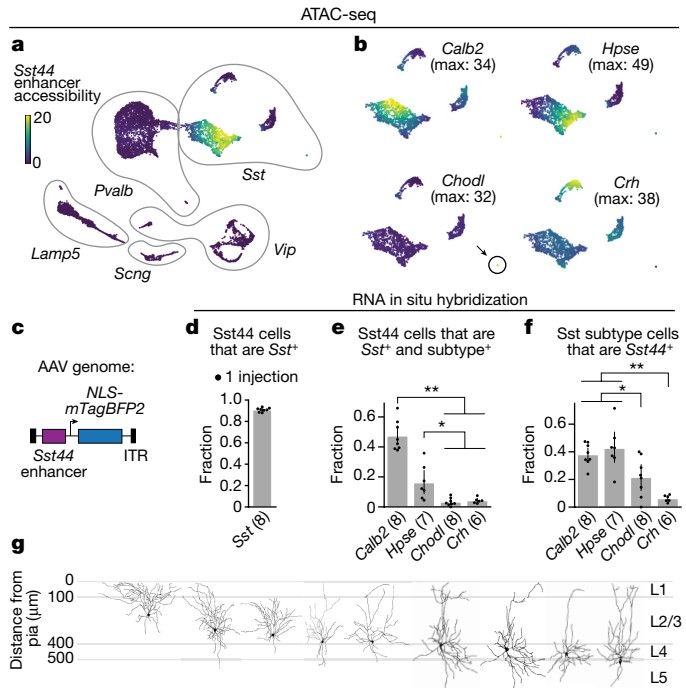

**Fig. 1 | The *Sst44* enhancer drives expression in *Calb2*⁺ and *Hpse*⁺ Sst neurons. a**, UMAP projection of 10,375 inhibitory cortical neurons showing *Sst44* enhancer accessibility. **b**, Accessibility of Sst subtype marker genes across Sst neurons. Max, maximum. **c**, AAV genome with the *Sst44* enhancer driving *NLS-mTagBFP2*. ITR, inverted terminal repeat. **d–f**, The specificity of viral *Sst44* expression was assessed using in situ RNA hybridization. **d**, The specificity of the *Sst44* enhancer for Sst cells. **e**, The fraction of Sst44 cells that are positive for each subtype. **f**, The fraction of each Sst subtype that is *Sst44*-positive. Statistical analysis was performed using Wilcoxon rank-sum tests; **$P < 0.01$, *$P < 0.05$. **g**, The morphology of single Sst44 neurons filled through a whole-cell patch. Data are mean ± bootstrapped 95% confidence intervals.

Note that the *Sst44* enhancer does not drive gene expression in all *Calb2*⁺ and *Hpse*⁺ cells (Extended Data Fig. 1b,c), indicating that *Sst44*-enhancer-driven expression overlaps with, but does not strictly mirror, the expression of these marker genes. We therefore refer to the neurons targeted by this enhancer empirically as Sst44 neurons.

## Sst44 cells activate as a group

To investigate the role of Sst44 cells during goal-directed navigation, we imaged their activity in the PPC using a two-photon microscope as mice navigated a virtual T-maze[1] (Fig. 2a,b). We trained head-fixed mice to run on an air-supported ball, the pitch and yaw rotations of which were used to move forwards and turn in a virtual environment projected onto a visual display (Fig. 2a). We rewarded mice for turning left at the T-intersection if a black cue was displayed on the maze walls and for turning right if a white cue was displayed (Fig. 2b). We imaged neural activity once the mice performed this task with an accuracy of greater than 85%. We also imaged neural activity in a similar maze in which the cue was omitted in the latter part of the maze (delay maze). We expressed jGCaMP7f from the synapsin promoter to image calcium activity from neurons and tdTomato in Sst neurons using the *Sst-cre* allele and a transgenic reporter. We expressed mTagBFP2 from the *Sst44* enhancer (Fig. 2c). This configuration enabled us to compare the activity of Sst44 neurons with the activity of other Sst and non-Sst neurons side-by-side in the same mouse, which importantly controlled for behavioural variability that might drive differences in activity that

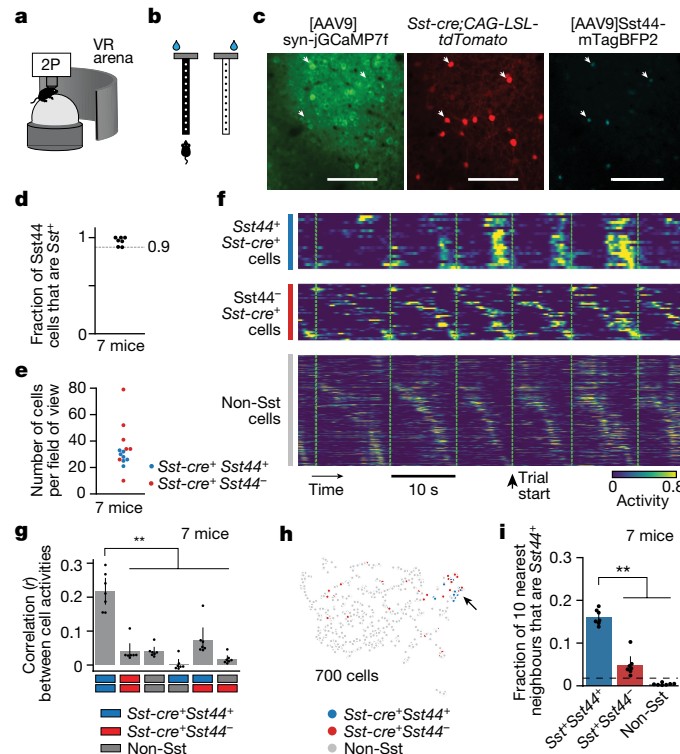

**Fig. 2 | Sst44 cells activate synchronously in a cell-type-specific pattern. a**, Imaging activity during virtual navigation. 2P, two-photon scanning microscope; VR, virtual reality. **b**, T-maze virtual environments. **c**, Example cropped field of view. Scale bars, 100 μm. **d**, The specificity of the *Sst44* enhancer. **e**, The number of Sst cells per field of view across three planes. **f**, Sample activity trace showing synchronous activity in Sst44 neurons. Each row is the activity of one cell. **g**, Pearson correlation between cells of each cell type. **h**, UMAP projection of each cell's activity from one session, showing clustering of Sst44 neurons. **i**, The fraction of the ten nearest neighbours in activity space that are *Sst44*-positive. The dashed line shows the mean after shuffling cell type identities. Activity was smoothed with a 0.25 s Gaussian filter. Statistical analysis was performed using Kolmogorov–Smirnov tests. Data are mean ± bootstrapped 95% confidence intervals.

are not due to cell type. The non-Sst neurons were mostly excitatory neurons and included other inhibitory neuron types. We confirmed the *Sst44* enhancer was greater than 90% specific for Sst cells (Fig. 2d). On the basis of previous research, the remaining Sst44 cells that are not Sst-positive are mainly parvalbumin inhibitory neurons[6]. To remove the contributions of these non-Sst cells, in our imaging and photo-stimulation analyses below, we analysed only Sst44 cells that were also positive for *Sst-cre*, which we confirmed is specific for Sst cells in the PPC (90 ± 3%, mean ± bootstrapped s.e., 4 mice). We detected around 20–40 Sst44 cells across three 650 μm × 650 μm fields of view spaced along the dorsal–ventral axis in layer 2/3 (46 ± 4% of all Sst cells, mean ± bootstrapped s.e., 7 mice) (Fig. 2e).

During the goal-directed navigation task, Sst44 cells tended to activate in unison in contrast to other Sst and non-Sst cells, which tended to activate at different points during the trial (Fig. 2f). We quantified this difference by computing the correlation in activity between single cells. Activity was more strongly correlated among Sst44 cells compared with among other Sst cells or non-Sst cells and across cell types (Fig. 2g). Moreover, the average activity of Sst44 cells was not correlated with the average activity of non-Sst cells, arguing against a role in normalization or gain control (Extended Data Fig. 2a–c). Consistently, the activity of Sst44 cells was distinct from that of most other Sst and non-Sst cells when visualizing cell activities in a uniform manifold approximation

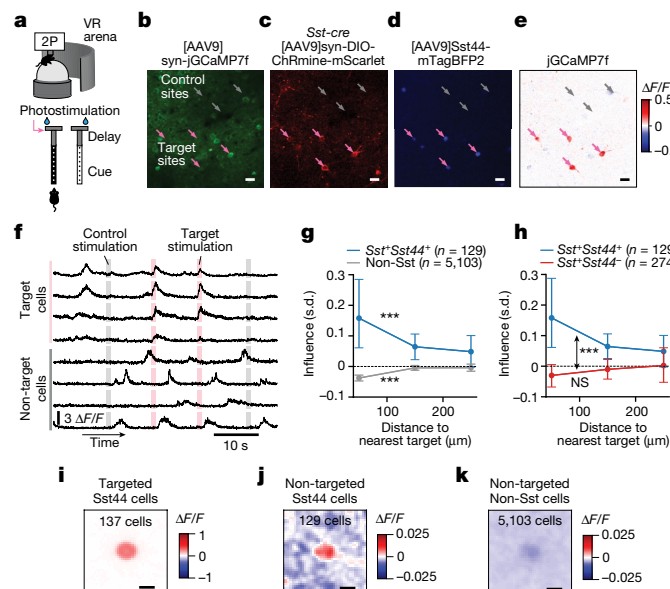

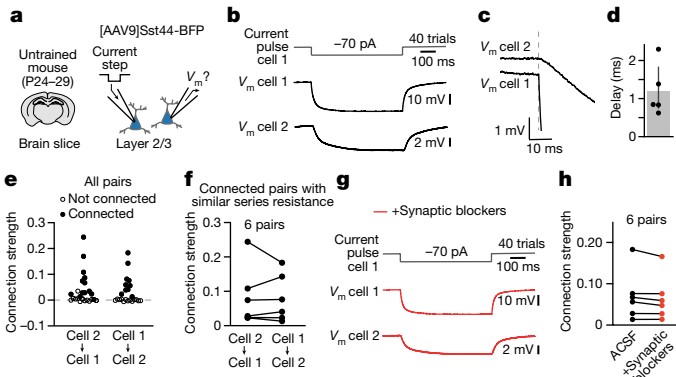

**Fig. 3 | Sst44 cells on average excite each other and inhibit other neurons during navigation. a**, Photostimulation of Sst44 cells during navigation, triggered on entering the T-junction. **b–d**, Example cropped sample field of view. **b**, jGCaMP7f in neurons. **c**, ChRmine–mScarlet in Sst cells. **d**, Nuclear-localized mTagBFP2 in Sst44 cells. **e**, Photostimulation-induced activity relative to the control trials. **f**, Sample trace of activity during photostimulation. **g**, Influence (in units of s.d., Methods) versus distance from the nearest target for *Sst44+* and non-Sst cells. Statistical analysis was performed using Wilcoxon signed-rank tests versus zero; ***$P < 1 \times 10^{-4}$. **h**, Influence versus distance from the nearest target as in **g**, but for *Sst44+* and *Sst44–Sst+* cells. The blue curve (*Sst44+*) is the same in **g** and **h** for comparison. Statistical analysis was performed using a Wilcoxon rank-sum test to compare *Sst44+* and *Sst44-Sst+* cells (double-sided arrow), and a Wilcoxon signed-rank test to compare *Sst44-Sst+* cells versus zero (not significant (NS)). Cells 40–200 μm from nearest target were selected for statistical tests. **i**, Average target trial minus control trial d*F/F* image cropped and centred on each photostimulated cell. **j,k**, Same as **i**, but for each influenced Sst44 cell (**j**) and each non-Sst cell (**k**). Data are mean ± bootstrapped 95% confidence intervals. n = 4 mice, 20 sessions. Scale bars, 20 μm (**b–e**) and 10 μm (**i–k**).

and projection (UMAP) plot (Fig. 2h). Indeed, the nearest neighbours of Sst44 cells in activity space were enriched for other Sst44 cells (Fig. 2i). Note that this clustering was not perfect—we observed that a minority of *Sst44– Sst+* cells clustered with Sst44 cells (24 ± 4%, mean ± bootstrap s.e.; Methods). This minority probably includes other cells of the same Sst44 subtype that, by chance, were not well transduced by the virus. It may also include another Sst subtype that responds similarly to Sst44 cells. We observed similar activity properties in mazes with and without the delay maze segment (Extended Data Fig. 2d–g). Sst44 cells therefore activate in a synchronous, cell-type-specific pattern, indicating that the *Sst44* enhancer defines a subset of functionally distinct Sst cells.

## Sst44 cells excite each other in vivo

We hypothesized that this synchronous and distinct activity pattern could be generated in part by interactions between Sst44 cells. We used influence mapping[23] to measure the effect of photostimulating groups of Sst44 cells on the activity of non-stimulated Sst44 cells, other Sst cells and non-Sst cells (Fig. 3a). To photostimulate neurons, we expressed soma-localized ChRmine—a red-shifted depolarizing opsin—fused to mScarlet in all Sst neurons. We used a spatial light modulator to generate multiple spots that were each scanned in a

**Fig. 4 | Sst44 cells are coupled through gap junctions. a**, Patch-clamp electrophysiology analysis of pairs of Sst44 cells in brain slices. **b**, The mean membrane voltage ($V_m$) from an example connected pair. **c**, Expanded view of **b** to display the delay. The $V_m$ for cells 1 and 2 are offset for visualization. **d**, The delay in the $V_m$ of the follower cell for strong connections (Methods). Mean: 1.2 ms, 5 connections. **e**, Connection strength ($\Delta V_m$ in the follower cell/$\Delta V_m$ in the driver cell) from all pairs. A total of 14 out of 26 pairs (54%) were connected. Statistical analysis was performed using Wilcoxon signed-rank tests with Benjamini−Hochberg correction; $P < 0.01$. **f**, The connection strength for connected pairs with similar series resistance (<30% difference), showing that the connection strength is reciprocal. **g**, The mean $V_m$ for the pair in **b** after adding inhibitors for NMDA (D-AP5), AMPA (NBQX) and GABA_A (gabazine) receptors. **h**, The connection strength before and after adding synaptic blockers.

spiral pattern approximately the size of a cell. In this way, we targeted 4–10 manually chosen neurons expressing both *Sst44*-enhancer-driven mTagBFP2 and *Sst-cre*-restricted ChRmine–mScarlet (target sites) (Fig. 3b–d). Photostimulation increased activity in targeted Sst44 neurons to a level comparable to high endogenous activity (Fig. 3e,f,i and Extended Data Fig. 3a). On the interleaved control trials, we targeted an equivalent number of sites that did not contain a ChRmine-expressing neuron (control sites).

To examine circuit interactions that are behaviourally relevant, we photostimulated groups of Sst44 cells as mice entered the T-intersection during the navigation task. We used calcium imaging to measure the effect on non-photostimulated cells, quantified as the change in activity in these cells after photostimulation of the targeted Sst44 cells relative to the control trials, normalized to the s.d. of their activity on control trials (Methods). As expected for an inhibitory cell type, we observed, on average, inhibitory influence on non-Sst neurons (Fig. 3g,k). Notably, photostimulation of Sst44 cells increased the activity of non-photostimulated Sst44 neurons (Fig. 3g,j) but not other Sst neurons (Fig. 3h). This influence decayed over 100–200 μm (Fig. 3g). To rule out off-target photostimulation effects, we confirmed that photostimulation directly drove activity only within 30 μm of the target sites[23] (Extended Data Fig. 3b), and therefore selected cells further than 40 μm from target sites for analysis. We also did not observe excitatory influence in *Sst44–Sst+* cells that also expressed ChRmine (Fig. 3h). Moreover, quantifying the influence of photostimulation relative to the control trials takes into account off-target photostimulation, and target and control sites were located at similar distances from non-photostimulated Sst44 cells (Extended Data Fig. 3c). Photostimulation did not induce noticeable changes in behaviour (Extended Data Fig. 3d,e), indicating that the excitatory influence among Sst44 cells is not due to a behavioural response to photostimulation. Although Sst44 cells, on average, did not significantly influence *Sst44– Sst+* cells, our imaging data revealed a weak correlation between the endogenous activity of these cell types (Fig. 2g) that may be the result of these cell types receiving similar inputs. Together, our results indicate that Sst44 cells excite each other, but not other Sst cells,

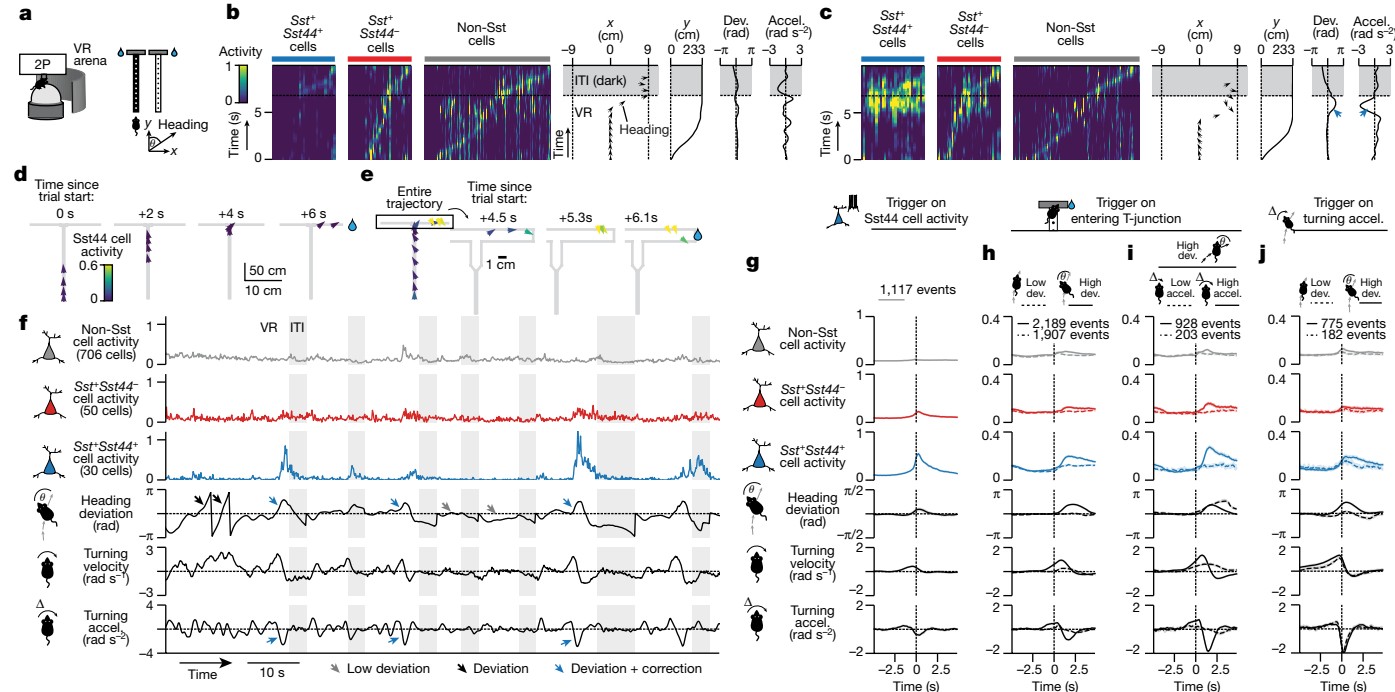

**Fig. 5 | Sst44 cells in the PPC activate during navigational course corrections.** **a**, Imaging activity during virtual T-maze navigation. **b**, Sst44 cell activity during a smooth trajectory towards the reward. Accel., turning acceleration; dev., heading deviation. **c**, Sst44 activity as in **b**, but during a trial with a course correction, highlighting high Sst44 cell activity. **d**,**e**, Trajectories of the trials in **b** and **c**, respectively. **f**, Sample trace of the mean cell type activities. **g**, Behaviour triggering on bursts of Sst44 cell activity. **h**, Activity triggered on entering the T-junction, split by large and small heading deviations. Left and right deviations were pooled after inverting behaviour for left deviations. **i**, Activity triggered on entering the T-junction as described in **h**, but splitting high-deviation trials

by strong and weak corrections. **j**, Activity triggered on turning accelerations, with high turning velocity (to match the turning profile across conditions), split by large and small deviations. Left and right turning accelerations were pooled after inverting behaviour for right accelerations. For **h**–**j**, statistical analysis was performed using Wilcoxon rank-sum tests across trials for Sst44 cell activity (solid versus dashed line, $P < 1 \times 10^{-10}$), and Wilcoxon signed-rank tests across trials for Sst44 cell activity versus the other cell types ($P < 1 \times 10^{-10}$). Activity averaged over 0.5 to 2.5 s relative to −1 to 0 s was used for statistical analyses. Data are mean ± bootstrapped 95% confidence intervals. $n = 7$ mice, 27 sessions.

further indicating that Sst44 cells are a functionally distinct group within the Sst population. This self-recruitment among Sst44 cells probably contributes to their distinct and synchronous activity during navigation.

## Sst44 cells are electrically coupled

Given that Sst44 cells are inhibitory, the excitatory influence among these cells suggests that this interaction is dominated by polysynaptic or electrical connections. To better understand the nature of this interaction, we performed patch-clamp electrophysiology analysis of pairs of Sst44 cells in layer 2/3 of the PPC in brain slices collected from untrained mice (Fig. 4 and Extended Data Fig. 3f–k). We injected a hyperpolarizing current in one cell (the driver cell) and measured the resulting deflection in the membrane potential of the second cell (the follower cell; Fig. 4a). In 54% of pairs (14 out of 26), we observed a hyperpolarizing deflection in the follower cell (Fig. 4b–e), suggesting that Sst44 cells are electrically coupled by gap junctions. If this hypothesis is true, we would expect that the connection is fast, reciprocal and unaffected by blockers of synaptic transmission. By contrast, a polysynaptic connection should be slow and strongly affected by synaptic blockers. Indeed, the follower cell responded rapidly with an average delay of 1.2 ms (Fig. 4c,d). Moreover, the connection was reciprocal and of similar magnitude in both directions (Fig. 4f). Finally, we observed no change in the connection strength after adding blockers for NMDA (*N*-methyl-D-aspartate), AMPA (α-amino-3-hydroxy-5-methyl-4-isoxazole propionic acid) and GABA_A (type A γ-aminobutyric acid) receptors (Fig. 4g,h). These results indicate that Sst44 cells are

electrically coupled through gap junctions, which could underlie the excitatory influence measured optogenetically in vivo (Fig. 3) and the synchronous activity of Sst44 cells (Fig. 2). These experiments do not rule out additional contributions from polysynaptic connections, for example, through reciprocal inhibitory connections between Sst and vasoactive intestinal peptide (Vip) neurons[13,24].

## Course-correction activity in Sst44 cells

Regarding when the synchronous activity bursts in Sst44 cells occur during goal-directed navigation, we first noticed that Sst44 cell activity varied widely across trials (Fig. 5a–f and Supplementary Videos 1 and 2). In one-third of the trials, Sst44 cells had little activity throughout the entire trial (Extended Data Figs. 4 and 5), indicating that these cells did not respond synchronously to the visual scene, to the mouse starting and continuing to run down the maze, to its position in the maze, to it turning into one of the T-arms, or to it stopping and consuming the reward. To identify what drove the activity of Sst44 cells, we characterized mouse behaviour during large bursts of activity in these cells. We found that, during these bursts, the mouse was often correcting for an error in navigating towards the reward zone (Fig. 5c,e,f and Extended Data Fig. 6). For example, some Sst44 cell events occurred in the T-arm as the mouse turned away from the reward zone and then corrected its course by turning back towards the reward zone (Fig. 5c,e,f and Supplementary Video 2 (mouse 1)). As another example, Sst44 cells were active in the T-stem when the mouse turned prematurely before the T-intersection and then corrected to continue its traversal down the

T-stem (Extended Data Fig. 6 and Supplementary Video 2 (mouse 2)). By contrast, we did not observe synchronous bursts of Sst44 cell activity when the mice navigated smoothly into the reward zone (Fig. 5b,d,f and Extended Data Figs. 4 and 5).

To systematically analyse these events, we quantified behaviour during bursts of Sst44 cell activity across mice. The bursts occurred when the mouse had turned away from its typical trajectory, resulting in a heading deviation, and increased its turning acceleration in the opposite direction, consistent with a corrective action to move back towards the typical trajectory (Fig. 5f (blue arrows) and 5g). We defined heading deviations as the difference in the mouse's virtual heading direction from its typical heading direction at the same maze position on trials with smooth navigational trajectories. The mouse also slowed its forward running speed before these Sst44 cell activity events and sped up afterwards, consistent with the mouse slowing down during the deviation and correction and speeding up after the correction (Extended Data Fig. 7a). This analysis triggered on Sst44 cell activity therefore showed that, on average, mice were correcting for a deviation in their trajectory during bursts of Sst44 cell activity.

We next analysed neural activity triggered on course corrections, which we defined as events with a heading deviation and a turning acceleration in the opposite direction. Although course corrections could occur at any point in the maze, they occurred most frequently as the mouse entered the T-intersection as here the mouse must make a sharp error-prone turn to reach the reward (Extended Data Fig. 5). Notably, the distribution of Sst44 cell activity events along the maze closely matched the spatial distribution of course corrections, with an enrichment near to the T-intersection (Extended Data Fig. 5). Sst44 cell activity was much greater when the mouse entered the T-junction with a large heading deviation compared with a low deviation (Fig. 5h (solid versus dashed lines)). When considering only the trials with a high heading deviation, we observed much more activity in Sst44 cells during deviations accompanied by a strong correction, as measured by the mouse's turning acceleration (Fig. 5f (blue arrows) and Fig. 5i (solid line)), compared with during deviations accompanied by a weak correction (Fig. 5f (black arrows) and Fig. 5i (dashed line)). Activity was intermediate for deviations that were accompanied by an intermediate correction (Extended Data Fig. 7c), consistent with a signal that depends on the strength of the correction. Thus, Sst44 cells are activated during course corrections rather than heading deviations alone.

Sst44 cells activated during course corrections specifically and not during the act of turning more generally. We could further distinguish between corrective and non-corrective turns because the mouse turned as part of its normal trajectory into one of the T-arms. Sst44 cells did not activate strongly during high turning accelerations when the mouse entered the T-junction with a low heading deviation (Extended Data Fig. 7d), or during velocity- and acceleration-matched turning events more generally (Fig. 5j), indicating that these cells do not generally respond to the act of turning. Moreover, Sst44 cells did not have high activity during large increases in forward velocity (Extended Data Fig. 7b). Sst44 cells therefore do not generally respond during forward or turning movements. Moreover, because turning controls rotations in the virtual environment, these analyses also indicate that Sst44 cells do not generally respond to visual flow, which we confirmed in the experiments described below.

Furthermore, we examined whether Sst44 cells activated similarly during sessions early in training. As a course correction implies knowledge of the reward location, a course-correction signal should be present only after learning. Indeed, during low-accuracy training sessions, Sst44 cell activity was low even during large turning accelerations towards the reward zone (Extended Data Fig. 7e). By contrast, Sst44 cell activity was moderate in sessions with moderate accuracy (Extended Data Fig. 7f). Thus, Sst44 cells activate only during corrective

turns once the mice have learned the reward location, consistent with a course-correction signal. These experiments also further demonstrate that Sst44 cells do not respond during the act of turning in general.

Thus, Sst44 cells activated weakly to heading deviations and to turning separately, but strongly to their combination (Fig. 5h–j), consistent with a course-correction signal. As Sst44 cells activate specifically during this combination, and because they are relatively silent for the entire length of smooth trajectories (Extended Data Fig. 4), our results highlight the specificity of Sst44 cell activity to course corrections rather than to other behaviours. Notably, this course-correction activity was strongest in Sst44 cells and much weaker in other Sst neurons and non-Sst cells (Fig. 5i), consistent with the distinct activity patterns in Sst44 cells described above. Non-Sst cells were active throughout the trial (with each cell active at different times), even when Sst44 cells were inactive[1] (Fig. 5b).

As a population, Sst44 cells were active during corrections for both leftward and rightward deviations (Extended Data Fig. 7g,h). Individual Sst44 cells also responded to both directions, with some responding more in one direction than the other (Extended Data Fig. 7i). Behaviourally, we observed a bias towards corrections for rightward deviations (Extended Data Fig. 7j,k), possibly due to our experimental set-up (Methods).

Although these analyses point to Sst44 cell activity during events characterized by a simple combination of heading deviation and an opposing turning acceleration, we also identified events of Sst44 cell activity that did not follow this rule but that were nevertheless consistent with a course correction. For example, some activity events occurred after the mouse turned 360° and slowed down its turning to prevent overshooting the correct navigational trajectory, in which case its turning acceleration was of the same sign as its heading deviation (Extended Data Fig. 6c,d and Supplementary Video 3). The heterogeneity of the mouse's behaviour across Sst44 cell activity events made it difficult to quantify all events in a single analysis and motivated us to consider experimental perturbations to test the conclusion that Sst44 cells are activated during navigational course corrections.

## Activity during navigation perturbations

We perturbed the mouse's heading by transiently introducing an angular drift that increased the mouse's virtual turning velocity in either direction (Fig. 6). This angular drift caused the mouse to rotate in the virtual environment and, therefore, introduced a heading deviation. In response to this heading perturbation, the mouse corrected by changing its turning in the opposite direction (Fig. 6a–f). Consistent with a course-correction signal, we observed a strong increase in Sst44 cell activity when the mouse corrected for this heading perturbation and little activity when the mouse did not strongly correct (Fig. 6b–g, Extended Data Fig. 8 and Supplementary Video 4). Furthermore, this response was strongest in Sst44 cells and was not strongly present in other Sst cells or non-Sst neurons (Fig. 6f,g).

Beyond this average response, we found that Sst44 cell activity depended both on the magnitude of the heading deviation and of the corrective action, consistent with a graded response to course corrections, and mirroring our analyses of spontaneous course corrections above (Extended Data Fig. 8c–f). Consistent with this idea, during perturbation trials with low turning acceleration and low Sst44 cell activity, the mouse nevertheless corrected, only on a slower timescale (Fig. 6g), indicating that Sst44 cells do not activate during all corrective actions, but only during stronger ones. These effects help to explain the variability in the magnitude of Sst44 cell activity during individual corrections.

As a second test, we introduced a navigational error by switching the visual cue on the maze walls, along with the location of the reward,

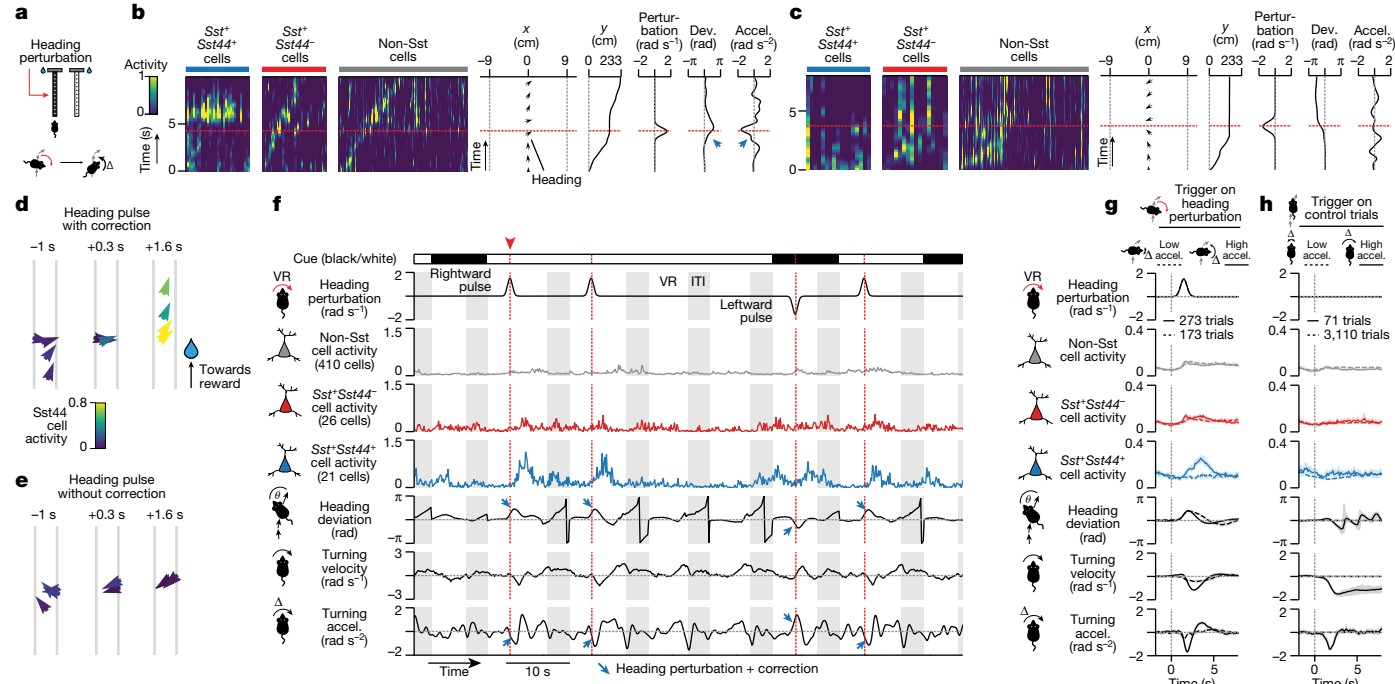

**Fig. 6 | Sst44 cells in the PPC activate during corrections for experimentally induced trajectory deviations. a**, Perturbation of virtual heading. **b**, Sst44 cell activity as the mouse corrects for a heading perturbation. **c**, Sst44 cell activity as in **b**, during a trial in which the mouse does not immediately correct. **d**,**e**, The trajectories of trials in **b** and **c**, respectively. **f**, Sample trace of mean activity in each cell type. **g**, Activity and behaviour during heading perturbations, split by strong and weak corrections. Left and right heading perturbations were pooled after inverting behaviour for leftward perturbations. **h**, Activity and behaviour as in **g**, at the same maze position on control trials. Statistical analysis was performed using Wilcoxon rank-sum tests comparing across trials the Sst44 cell activity (solid versus dashed line, $P = 3 \times 10^{-12}$ (**g**), $P = 0.9$ (**h**)); and Wilcoxon signed-rank tests comparing across trials the Sst44 cell activity versus other cell types ($P < 1 \times 10^{-8}$ (**g**), $P > 0.1$ (**h**)). Activity averaged over 2.5 to 4.5 s relative to 0 to 1 s was used for statistical tests. Turning velocities and accelerations refer to ball movements, not virtual movements (the latter also include the heading perturbation). Data are mean ± bootstrapped 95% confidence intervals. $n = 8$ mice, 16 sessions.

halfway through the T-stem (Extended Data Fig. 9). In these trials, the mice had to course-correct by transitioning, in the middle of a trial, from a trajectory towards one reward location to a trajectory towards the opposite reward location. These trials were interleaved with control trials without a cue switch. Consistent with a course-correction signal, we observed an increase in Sst44 cell activity time-locked to the cue switch. This activity emerged only after the mice learned to change trajectories (Extended Data Fig. 9b,c). This increase in activity for course corrections during cue-switch trials was mostly absent in other cell types (Extended Data Fig. 9b,c).

These perturbation experiments also confirmed that Sst44 cell activity in the PPC was not generally related to the act of turning. Sst44 cell activity was low when the mouse turned without a heading perturbation on the interleaved control trials (Fig. 6h), or when the mouse turned during the perturbation but while its heading deviation was low (Extended Data Fig. 8d). These experiments also showed that Sst44 cells do not generally respond to visual flow, as Sst44 cell activity was also low during uncorrected perturbations (Fig. 6g (dashed line)), where a rotation of the visual scene was still present. To confirm this conclusion, we played back the visual scene from heading perturbation trials in open loop (Methods). We observed little Sst44 cell activity in open loop playback, ruling out a response purely to the visual stimulus, including visual flow (Extended Data Fig. 10a,b).

## Sst44 cells do not signal all error types

Given their activity during navigational course corrections, we tested whether Sst44 cells also respond to an error in reward prediction by omitting rewards on a subset of correct trials. We observed little difference in Sst44 cell activity between the rewarded and unrewarded trials, arguing against a response that is tied to immediately predicting the reward (Extended Data Fig. 10c). We also investigated whether Sst44 cells respond to an error in predicting visual inputs by analysing Sst44 cell activity when the mouse was running parallel to the maze wall (where visual flow is expected and present) or perpendicular to the maze wall (where visual flow is expected but not present). Sst44 cell activity was similar between these two conditions, suggesting that their activity does not reflect an error in predicting visual flow (Extended Data Fig. 10d). Moreover, Sst44 cells did not respond to unexpected visual flow during heading perturbations when the mouse did not have a strong corrective turn (Fig. 6g (dashed line)). Thus, Sst44 cells in the PPC do not seem to carry a generalized error signal but, rather, respond specifically to course corrections while navigating towards a reward location.

## Sst44 cell activity in another area

Given that similar interneuron subtypes are present across cortical areas[5], Sst44 cells are expected to be found in areas beyond PPC. In the same mice in which we observed course-correction signals in the PPC (Fig. 5), we moved our imaging field of view on interleaved sessions to another navigation-related area, the retrosplenial cortex (RSC). Notably, we did not observe strong activity during course corrections in Sst44 cells in the RSC (Extended Data Fig. 11a–c). By contrast, these cells did activate synchronously and in a manner that was distinct from other cell types (Extended Data Fig. 11d–f). Thus, Sst44 cells in the PPC

and RSC are similar in that they both activate in a cell type-specific, synchronous pattern, but how this activity relates to behaviour or sensory inputs appears to be area-specific.

## Discussion

These results highlight a navigation error-correction signal in a rare cell type in the PPC. To our knowledge, a navigation error-correction signal has not been reported previously, either in the PPC or in other parts of the brain. Although Sst neurons have been studied in the context of sensory prediction errors[7], comparisons of this navigation error-correction signal to other work on Sst neurons is challenging because previous studies could not isolate this subtype of neuron.

Error signals are fundamental to learning algorithms in biological and artificial neural networks. Thus, one potential function for this error-correction signal is to act as a teaching signal that enables the learning of navigational trajectories to reach reward locations. Sst44 cells are well positioned to serve a function in plasticity because they primarily target dendrites, and project to layer 1 of the cortex, which receives feedback connections from other areas. Consistent with this function, Sst neurons in the motor cortex are necessary for learning and the regulation of the spine density in layer 1[25].

Error signals are also important for motor control. This error-correction signal could therefore also act as a command signal that contributes to the mouse executing corrections in real-time. In this scenario, Sst44 cells may suppress the activity of neurons favouring the current, incorrect turning direction, and allow for a change in activity favouring turning in the correct direction. This potential function is not mutually exclusive with a function in learning, and the two could operate concurrently. Owing to the temporal resolution of our calcium signal and of our behavioural measurements, which were limited by the inertia of the ball, we could not conclude whether Sst44 cell activity preceded or followed course-correction events, which may further help to distinguish between these two potential learning and command functions. We attempted an optogenetic inhibition experiment to test the involvement of Sst44 cells in moment-to-moment course corrections in well-trained mice, but further experiments will be needed in this direction (Extended Data Fig. 12).

A major feature of the error-correction signal is that it appears in synchronous bursts of activity in Sst44 cells, which are probably driven in part by gap junctions between these cells. These results are consistent with previous reports on gap junctions between Sst cells more generally[26] and within a subtype of layer 2/3 neurons in somatosensory cortex with similar anatomical properties as Sst44 neurons[27]. How the electrical coupling rate we observed among Sst44 cells compares with all Sst cells in layer 2/3 of the PPC remains to be determined. It may be that gap junctions contribute to computing the error-correction signal by averaging the activity of upstream cells that are also active during these events.

The finding of a navigation error-correction signal in the PPC helps to define this area's function. Previous research has proposed a role for the PPC in planning and executing navigational routes[1,15,16]. The fact that Sst44 cells activate during corrective actions is consistent with an action-centred role for the PPC in navigation, and a role for planning and executing actions more generally[20]. Although the error-correction signal in Sst44 cells depended on the strength of the error and the correction, one could also conceive of an error-correction signal that is either on or off. The fact that Sst44 cells carry a graded signal may imply that the magnitude of inhibition provided by these cells is important for their function. Further investigation into the functional significance of the error-correction signal carried by Sst44 neurons may offer additional understanding of the specific function of the PPC within a larger network of areas involved in goal-directed navigation[28].

Sst44 cells are expected to be present in multiple cortical areas[5]. Notably, even though Sst44 cells in the RSC are also synchronously active,

they do not activate strongly during course corrections, indicating that this activity is not present in all areas. Given these data, we speculate that Sst44 cells carry an error-related signal in different areas of the cortex, with this signal tailored to each area's function. If true, these cells could provide an error-related signal at the level of a cortical area, which would bridge the spatial gap between the error signals carried by dopamine neurons, which project throughout the brain[29], and synapse-level error signals that are instrumental in training artificial neural networks.

Identifying the error-correction signal in Sst44 neurons was made possible by new technology to target increasingly precise cell types using enhancer AAVs[6,30,31]. These findings also highlight that fine-scale distinctions between cell types at the molecular level matter at the circuit and physiological levels. The repeatable access provided by the *Sst44* enhancer has the potential to open a range of experimental studies on the functions and mechanisms of a cortex-wide cell type and a fundamental class of signals in the cerebral cortex.

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

## Methods

### Mice

All of the experimental procedures were approved by the Harvard Medical School Institutional Animal Care and Use Committee. The following mouse lines were used: *Gad2-cre* (Jax, 010802), *Sun1-GFP* (Jax, 021039), C57Bl/6J (Jax, 000664), *Sst-cre* (Jax, 013044)[32] and *Ai14* (Jax, 007914)[33]. Mice were housed under a reversed 12 h–12 h light–dark light cycle. Sample sizes were chosen on the basis of previous similar experiments. Trial types were randomly interleaved. Blinding is not relevant because comparisons were made within animals or samples.

### Viruses

pAAV2-Sst44-mTagBFP2 (pAAV2-2xSV40pA-Sst44-CMVminP-NLS-Flag-mTagBFP2-NLS-WPRE-SV40pA), pAAV2-A2-Sst44-mTagBFP2 (pAAV2-A2-2xSV40pA-Sst44-CMVminP-NLS-Flag-mTagBFP2-NLS-WPRE-SV40pA-A2) and pAAV2-A2-syn-jGCaMP7f (pAAV2-A2-syn-jGCaMP7f-WPRE-SV40pA-A2) were cloned in-house and verified by Sanger sequencing using Genewiz. The A2 insulator[34] was added in the last two constructs to test its effect on cell type specificity, but we observed similar specificity in both cases. pAAV2-syn-DIO-ChRmine-mScarlet (pAAV2/9-hSyn-DIO-ChRmine-mScarlet-Kv2.1-WPRE-hGHpA) was made by GenScript. pAAV2-Sst44-stGtACR2-mNeonGreen (pAAV2-2xSV40pA-Sst44-CMVminP-GtACR2-mNeonGreen-ST-WPRE-SV40pA) was cloned, packaged with AAV9 and titred by Vigene. AAV9-packaged syn-jGCaMP7f was obtained from Addgene (104488). Other custom constructs were packaged with AAV9 and titred by qPCR by the Boston Children's Hospital Viral Core. All viruses were diluted in PBS to the final titre indicated in each experiment.

### Single-cell ATAC-seq

**Nucleus isolation.** For nucleus isolation for single-cell ATAC analysis[35], we used two *Gad2-cre*[+/−];*Sun1-GFP*[+/−] female mice to isolate inhibitory neurons expressing SUN1–GFP on the nuclear membrane. This selection yielded higher-resolution cell type information from inhibitory neurons, which are numerically underrepresented in single-nucleus isolation protocols without enrichment procedures. Nuclei were isolated as previously described[6,36], with modifications for fluorescence-activate cell sorting (FACS). The posterior cortex (around 3 mm by 3 mm centred on PPC) was dissected in ice-cold choline solution (2.1 g $l^{-1}$ NaHCO$_3$, 2.16 g $l^{-1}$ glucose, 0.172 g $l^{-1}$ NaH$_2$PO$_4$·H$_2$O, 7.5 mM MgCl$_2$·6H$_2$O, 2.5 mM KCl, 10 mM HEPES, 15.36 g $l^{-1}$ choline chloride, 2.3 g $l^{-1}$ ascorbic acid, 0.34 g $l^{-1}$ pyruvic acid), and transferred to a Dounce homogenizer containing homogenization buffer (0.25 M sucrose, 25 mM KCl, 5 mM MgCl$_2$, 20 mM Tricine-KOH, pH 7.8, 1 mM DTT, 0.15 mM spermine, 0.5 mM spermidine and protease inhibitors). The tissue was dounced with a tight pestle until it was well homogenized (10–15 strokes). IGEPAL (final 0.15%) was added, followed by 5–10 more strokes. The homogenate was passed through a 40 μm filter. Tween-20 (final 0.1%) and BSA (final 1%) were added to the filtrate. Nuclei were centrifuged at 500*g* for 5 min at 4 °C and resuspended in 0.5 ml wash buffer (10 mM Tris-HCl pH 7.4, 10 mM NaCl, 3 mM MgCl$_2$, 1% BSA, 0.1% Tween-20). Nuclei from each sample were sorted by FACS (Sony, SH800Z) based on the SUN1–GFP signal into two wells in a cooled 96-well plate coated with wash buffer. We used a 96-well plate because the sorted volume was small. After sorting, the plate was centrifuged at 500*g* for 5 min at 4 °C, and the nuclei were resuspended in 20 μl of Wash Buffer.

**Library preparation and sequencing.** Approximately 15,000 nuclei from each sample were combined with the transposition mix (10x Genomics), and the manufacturer's protocol for single-cell ATAC (10x Genomics, CG000168 Rev A) was followed, using one Chromium Controller (10x Genomics) lane per sample for cell barcoding. The libraries were sequenced on the NextSeq 500 DNA sequencer (Illumina).

**Data pre-processing and analysis.** Reads were mapped to the mouse genome (mm10), and cell barcodes were processed using the Cell Ranger pipeline (10x Genomics) using the default parameters. We used SnapATAC[37] (v.1.0.0) to further process the data, including binning mapped reads across the genome into 5,000 bp bins, filtering barcodes on the basis of the number of unique fragments (1,000–100,000) and the fraction of reads in promoters (0.1–0.6), performing dimensionality reduction (30 dimensions) on the normalized Jaccard similarity matrix and constructing a *k*-nearest-neighbour graph ($k = 0.5 \times \sqrt{(\text{number of barcodes})}$) (see the SnapATAC pipeline for a complete description). Following published multiplet detection algorithms[38,39], we simulated doublets, triplets and quadruplets by summing random combinations of cells and removed cells on the basis of their multiplet score, defined as the fraction of nearest neighbours that were simulated multiplets (higher multiplet score indicates higher likelihood that the cell is not a singlet). We chose a multiplet score threshold based on the trough in their distribution and based on whether cells above this threshold tended to have higher fragment counts. We repeated this process a total of two times since we observed a large fraction of multiplets in the first round, which was consistent with nucleus clumping after FACS.

After removing cells that were not inhibitory neurons based on the accessibility of marker genes, our dataset included 10,375 inhibitory neurons with an average of 12,594 fragments per cell. Although stringent, this process significantly cleaned up the data and revealed all five inhibitory neuron classes (Sst, Pvalb, Vip, Lamp5 and Sncg; Extended Data Fig. 1a), with high enough resolution to distinguish precise cell types within these classes. We repeated dimensionality reduction (*n* = 24) and the construction of the k-nearest neighbour graph (*k* = 15) after this selection. We clustered cell types using the Leiden algorithm[40] (resolution = 0.7) and visualized cell type clusters with UMAP[41] (default parameters within SnapATAC).

### Imaging native fluorescence with in situ RNA hybridization

**Surgeries.** We injected four C57BL/6J mice bilaterally for a total of eight injected hemispheres. In each hemisphere, four injections were made 0.5 mm apart in a grid in the PPC (centred at 1.7 mm lateral and 2.0 mm posterior to bregma) at depths of 0.20 and 0.70 mm below the dura. Each injection was approximately 65 nl of AAV2/9-Sst44-mTagBFP2 diluted to $5 \times 10^{11}$ GC per ml in PBS. We considered each unilateral grid of injections to be one replicate.

**Sample preparation.** Mice were perfused with PBS and 4% paraformaldehyde (PFA) approximately 2 weeks (13–16 days) after injections. The brains were post-fixed in 4% PFA at 4 °C overnight before sectioning 50 μm coronal slices on a vibratome (Leica, VT1000S). The slices were stored in antifreeze solution (40% 1× PBS, 30% ethylene glycol, 30% glycerol) at −20 °C for up to 2 months.

**Dual imaging of native fluorescence and RNA hybridization overview.** To combine mTagBFP2[42] and RNAScope (ACDBio) imaging, we first imaged mTagBFP2 native fluorescence, then performed RNAScope analysis of the same slices, and imaged the RNA signal in the same area. This protocol enabled us to combine native fluorescence and RNA labelling, as the RNA labelling protocol involves a protein-degradation step.

**Imaging native fluorescence.** Slices were washed three times with 0.3% Triton X-100 in PBS (PBS-T) to permeabilize cell membranes before applying DRAQ7 (Abcam, ab109202) nuclear stain 1:500 in PBS-T for 7 min at room temperature. Slices containing injection sites were washed in PBS and then mounted onto Superfrost Plus glass slides (Thermo Fisher Scientific, 12-550-15) with a glass coverslip secured using ProLong Gold Antifade Mountant (Invitrogen, P36934).

Images of the injection sites were acquired using the Olympus FV1000 Confocal Microscope with a 0.4 NA ×10 air objective (Harvard

Medical School Neurobiology Imaging Facility) and encompassed an area of 1,272 μm by 1,272 μm. The slides were stored in 5× saline sodium citrate at room temperature overnight.

**In situ hybridization using RNAScope.** After gently sliding off the coverslips in 5× saline sodium citrate, the slides were washed in PBS then dried thoroughly. Tissue pretreatment and in situ hybridization was performed according to the protocol delineated by RNAScope Fluorescent Multiplex Reagent Kit v2 (ACDBio, 323110) for fixed-frozen tissue with the following modifications. After baking, the slices were fixed in 4% PFA in PBS for 30 min at 4 °C. To minimize tissue warping, hydrogen peroxide (ACDBio, 322381) incubation was limited to 5 min and target retrieval (ACDBio, 322000) was brought to a boil but then allowed to cool until boiling had stopped before submerging the slides for 15 min. Finally, the slices were incubated with Protease IV (ACDBio, 322336) for 15 min at 40 °C.

We used the following RNA probes manufactured by ACDBio: *Sst* (404631-C3), *Hpse* (412251-C2), *Chodl* (450211-C2), *Calb2* (313641) and *Crh* (316091-C2). The fluorophores Cyanine 3 Amplification Reagent (FP1170) and Fluorescein Amplification Reagent (FP1168) from PerkinElmer were diluted in TSA Buffer (ACDBio, 322810) at the following concentrations: Fluorescein 1:1,500 for *Sst*; Cyanine 3 1:375 for *Hpse*, *Chodl* and *Calb2*; and Cyanine 3 1:190 for *Crh*. We also performed experiments using the *Nr2f2* probe and other gene markers for the *Nr2f2*[+] Sst cell cluster but could not extract a reliable signal, and therefore do not show these results here.

After in situ hybridization, the slices were incubated with DRAQ7 1:25 in PBS-T for 10 min at room temperature and washed briefly in distilled water. Excess liquid was removed from the slides before securing the coverslips with ProLong Gold Antifade Mountant. Images were taken of the same area as the native fluorescence image by aligning the two using the DRAQ7 nuclear dye.

**Image registration and processing.** A registration matrix was computed based on the common DRAQ7 nuclear channels taken before and after RNAScope analysis of each slice using either the ACDBio HiPlex Registration software or by manually selecting control points (25–70 pairs) to compute a local weighted mean transformation in MATLAB. These registration matrices were applied to each channel in the RNAScope images.

We trained two Cellpose[43] models to segment NLS–mTag-BFP2-labelled nuclei and *Sst*-RNA-labelled cells. Masks generated by these Cellpose models were excluded from analysis if they had an area of less than 75 pixels for BFP masks and less than 100 pixels for *Sst* masks to eliminate the background noise, and if they were located below cortex. We also excluded BFP masks in regions that did not overlap with the RNAScope image. BFP and *Sst* masks with a Pearson's spatial correlation of less than 0.16 between DRAQ7 nuclear channels in images taken before and after RNAScope for an area of 50 × 50 pixels around the centre of the masks were also excluded from the analysis.

We used Fiji[44] to generate the images in each figure.

**Analysis of cell types and depth distribution.** To define *Sst*-positive BFP-labelled cells, we mapped BFP masks to *Sst* masks. A BFP mask was assigned as an *Sst* cell if it had at least 50% overlap with that *Sst* mask.

We used the *Sst* masks to define the intensity of each *Sst* subtype marker, calculated as the mean intensity inside the *Sst* mask. Cells were considered to be positive if the intensity was greater than 1.5× the 5th percentile of cell means (used as an estimate of background intensity) in each image.

Cells were considered to be BFP[+] if the BFP intensity was greater than 15% of the average intensity of the three brightest BFP masks from each image. For determining the percentage of subtype cells that are BFP[+], images were used only if they contained at least ten BFP[+] cells. For the percentage of BFP[+] cells that were subtype positive, injections (which could include multiple images) were only used if they contained at least ten BFP[+] cells across slices.

A line along the pia was drawn manually in Fiji ImageJ for each image. Cell depth was defined as the vertical distance from this line to the centre of a BFP or Sst mask.

To analyse the specificity of the *Sst-cre* line in the PPC, we counted the overlap between *Sst-cre* cells expressing tdTomato and the *Sst* probe using RNAScope.

## Surgeries for in vivo experiments

Surgeries were performed at 8–12 weeks old. Mice were injected with dexamethasone (2 mg per kg) 1–6 h before surgery. A titanium head-plate was fixed to the skull with dental cement (Metabond, Parkell) mixed with carbon powder to prevent light contamination, centred on (−1.7, −2 mm, medial–lateral (ML) and anterior–posterior (AP) axes, respectively, from bregma), on the left side. A 3.5 mm diameter craniotomy was performed centred on (−1.7 (ML), −2 mm (AP)). Virus was injected at nine sites, at 0.25 mm below the dura: four centred on (−1.7 (ML), −2 mm (AP)) spaced by 0.3–0.4 mm targeting the PPC, two centred on (−0.8 (ML), −2 mm (AP)) spaced along the anterior–posterior axis by 0.3–0.4 mm targeting the RSC, and three at (−2.3 (ML), −2.6 mm (AP)), (−1.9 (ML), −2.6 mm (AP)) and (−2.7 (ML), −3 mm (AP)) targeting the visual cortex and the surrounding visual areas. The last three injections were used only for retinotopic mapping. For each injection, a bevelled glass pipette was inserted 50 μm past the desired depth, and then retreated to the correct depth. Approximately 65 nl was injected over 3 min, after which we waited 3 min to allow the pressure to equilibrate before retracting the pipette. After a durotomy, a cranial window, consisting of two 3-mm-diameter coverslips and one 4-mm-diameter coverslip (#1 thickness, Warner Instruments) bonded using ultraviolet-curable optical adhesive (Norland Optics, NOA81), was inserted. An aluminium ring was cemented onto the titanium headplate as an adapter for preventing light contamination.

The headplate was mounted approximately parallel to the tangent plane of the left PPC. This meant that the mouse's head was tilted slightly to the right for all of the experiments. This tilt probably contributes to the behavioural bias of the mice to have more course corrections for rightward deviations (Extended Data Fig. 7j,k).

For imaging experiments, we used *Sst-cre*[+/−];*Ai14*[+/−] male mice to label Sst cells with tdTomato. We injected and trained five cohorts of mice. Imaging cohort 1 (7 mice) was used for imaging activity in the PPC (3, 4, 5, 7, 3, 3, 2 sessions per animal, 27 sessions total) and the RSC (4, 2, 4, 6, 3, 1, 3 sessions per animal, 23 sessions total). Imaging cohort 2 (8 mice) was used for imaging activity in the PPC during heading perturbations (2, 2, 3, 2, 2, 1, 2, 2 sessions per animal, 16 sessions total). We used 2 mice from cohorts 1 and 2 (4 mice total, 1, 1, 2, 2 sessions per mouse, 6 sessions total) for the reward-omission experiments. Imaging cohort 3 (6 mice) was used for imaging activity in the PPC during training before high-accuracy performance (2, 2, 2, 2, 2, 1 sessions per animal, 11 sessions total). Imaging cohort 4 (7 mice) was used for training on the cue switch (6, 5, 6, 6, 8, 7, 7 sessions per animal, 45 sessions total). Imaging cohort 5 (5 mice) was trained on a linear maze, which we used for the playback experiments (3, 3, 3, 4, 3 sessions per animal, 16 sessions total). The virus mixture consisted of AAV2/9-syn-jGCaMP7f (6.25 × 10[11] GC per ml) and AAV2/9-Sst44-mTagBFP2 (5 × 10[11] GC per ml) in all mice, except one mouse in cohort 1, in which we injected a mixture of AAV2/9-A2-syn-jGCaMP7f (9.4 × 10[11] GC per ml) and AAV2/9-A2-Sst44-mTagBFP2 (1 × 10[11] GC per ml). The only difference here is the addition of the A2 insulator[34], which we added to test its effect on specificity. Given that we did not observe a difference in specificity (Fig. 2d), we pooled the data together.

For photostimulation experiments, we injected and trained *Sst-cre*[+/−] male mice (*Sst-cre*[+/+] crossed to C57Bl/6J; 4 mice; 5, 3, 7, 5 sessions per mouse, 20 sessions total). The virus mixture consisted of

AAV2/9-syn-jGCaMP7f ($6.25 \times 10^{11}$ GC per ml), AAV2/9-Sst44-mTagBFP2 ($5 \times 10^{11}$ GC per ml) and AAV2/9-syn-DIO-ChRmine-mScarlet ($5 \times 10^{11}$ GC per ml).

## Behavioural training

**Behavioural set-up.** We used a miniaturized virtual reality system, as described previously[45]. Head-fixed mice ran on an air-supported 8-inch-diameter Styrofoam ball. Ball velocities were tracked by two optical sensors (ADNS-9800, Avago Technologies) and digitized (USB-6003, National Instruments). The ball's pitch and yaw velocity were used to control the mouse's forward movement and heading velocity (view angle), respectively, in a virtual environment that was projected on a parabolic screen covering ~180°. Light exiting the laser projector (PicoBit laser projector, 60 Hz, Celluon) was dimmed with a neutral density filter (NE10B-A, Thorlabs) and short-pass filtered (550 nm cut-off, FES0550, Thorlabs), which helped to improve the mouse's behaviour and minimize light contamination during imaging. The rewards were delivered through a metal spout, controlled by a solenoid valve, and consisted of 0.15 g per 100 ml acesulfame potassium (Prescribed For Life; an artificial sweetener) in tap water.

We constructed two training rigs, plus one rig that was used for imaging, and one rig that was used for photostimulation experiments. The rigs were made to be as identical as possible to each other. The head plate was positioned 1 inch behind the centre of the ball, with a distance 1 inch above the ball, and at the horizontal and vertical centre of the visual display. Mice were typically initially trained on the training rigs and then transferred to either the imaging or photostimulation rig.

**Virtual environment and training.** Starting 3 days before training, mice were given 1 ml total water per day, including rewards received during training. We monitored their body weight, and further supplemented with water to ensure it was above 75% of their pre-training weight. The virtual environment was generated by Virtual Reality Mouse Engine (ViRMEn)[46] in MATLAB (MathWorks). The environment consisted of a T-maze with a white or black cue on the walls, as previously described[1]. Once the mouse entered 9 cm into either arm, the trial ended. The mouse was then rewarded for turning left if a black cue was presented, and for turning right if a white cue was presented. After the trial ended, we displayed a dark screen during an intertrial interval of 3 s for correct trials and 7 s for incorrect trials, after which the mouse started at the start of a new, randomly chosen T-maze. Mice started in a short maze (~64 cm long, or 1 ball rotation) in which a tower at the end of the maze indicated the location of the reward, in addition to the cue on the walls. Once mice started to perform well on this maze, they were advanced to longer mazes (113 cm and 225 cm) that required better ball control for them to smoothly navigate to the end. Once they performed with high accuracy, a tower was added on both sides at the end of the T-maze, such that the mice could no longer run towards the tower but had to navigate based on the cue on the walls. We included crutch trials, in which one reward-locating tower was still present, as mice transitioned to this maze. Once proficient on the maze with two towers, a delay maze segment was added in between the cue and the T-junction. The delay maze segment was grey and had a different texture than the reward-associated white or black cue. The delay was introduced as a short segment at the end of the maze and was gradually lengthened to 27% or 50% of the maze length. If the mice displayed a strong bias for turning left or right, we implemented bias correction by setting the probability of a right-rewarded maze as one minus the fraction of trials where the mouse turned right on the last 20 trials.

For the heading perturbation experiments, we added a bias to the mouse's virtual heading velocity triggered when the mouse passed the halfway mark in the maze, on 30% of trials. Left and right virtual rotations occurred with 50% probability each. The added bias varied with a Gaussian profile (mu = 1.5 s, sigma = 0.375 s, peak = 0.03 rad s$^{-1}$), resulting in a rotation of 103° over 3 s (peak velocity at 1.5 s) if the mouse did

not correct. Note that this perturbation does not result in a pure open loop rotation but, rather, adds a bias on top of the normal closed-loop configuration. In other words, the mouse can still compensate for the perturbation by turning on the ball. We also note that we initially tried jumping the mouse's heading instantaneously, rather than in this smooth manner, but this tended to activate the entire circuit in a cell-type-independent manner, which made it more difficult to isolate the Sst44-cell-specific response.

For training mice on the cue-switch experiments, we first trained mice to perform the T-maze with the delay with high accuracy (>85% correct trials). We then imaged mice in a T-maze where on 50% of trials, the visual cue was changed at the halfway point in the maze (for one mouse, we changed the cue in the last quarter of the maze; data are always aligned to the cue switch location), either from black to white or from white to black. The change in cue was visible as the mouse approached it. The rewarded location changed along with the cue. Thus, the mice had to learn to change trajectory based on the second cue to receive a reward. The other 50% of trials were control trials in which the cue was constant, as in the original maze. There was no delay on any trial for this experiment.

For training mice on a linear track, the linear track consisted of a corridor of the same length as the first training T-maze (64 cm long). Mice received a reward at the end of the maze. Trials were also separated by an intertrial interval during which we displayed a dark screen. We trained the mice until they performed at least one trial per minute before performing open-loop playback experiments.

## Widefield imaging for retinotopic mapping

We performed widefield epifluorescence imaging of jGCaMP7f[47] to generate a retinotopic map that was used to determine the two-photon imaging field of view for the PPC as previously described[48]. In brief, we excited jGCaMP7f with blue light (452–486 nm band-pass filtered, Thorlabs) and filtered green emitted light (505–545 nm band-pass filtered, Thorlabs) that was imaged with a CMOS camera (acA1920-155um, Basler) with a field of view covering the entire cranial window. Mice were anaesthetized under 1% isoflurane. Visual stimuli were coded with Psychtoolbox (MATLAB, MathWorks), presented on a 27 inch monitor, and consisted of a spherically corrected bar 12.5° in width moving at 10° s$^{-1}$ horizontally or vertically in either direction. The bar was patterned with a 3 Hz alternating black and white checkered pattern. Data were processed as previously described[49,50] using a temporal Fourier transform to extract responses to horizontal and vertical bar positions (since visual stimuli were presented periodically in time), creating horizontal and vertical response maps. The field sign was computed as the sine of the angle between the gradient of the horizontal and vertical maps. Fields of view for the PPC were centred approximately at (−1.7 (ML), −2 mm (AP) from bregma) as in previous studies[48,51].

## Two-photon imaging

**Microscope design.** We collected in vivo imaging data on a custom built two-photon scanning microscope with a ×16/0.8 NA water-immersion objective with a 3 mm working distance (Nikon) and a tunable femtosecond-pulsed laser (Chameleon Vision S, Coherent). The beam was scanned with a resonant and galvanometric mirror pair (Cambridge Technology) relayed with two scan lenses, scanning a 650 μm by 650 μm field of view at a resolution of 512 × 512 pixels at 30 frames per second for one plane (see the acquisition parameters specific to each experiment below). We imaged layer 2/3 at a depth of 100–250 μm below the pia. Green and red light were split with a 580 nm dichroic beamsplitter (FF580-FDi01-55x73, Semrock), and band-pass filtered (green, 500–550 nm, FF03-5525/50-50, Semrock; red, 604–679 nm, FF01-641/75-50, Semrock). Blue and green light were split with a 484 nm dichroic beamsplitter (FF484-FDi01-55x73, Semrock), and band-pass filtered (blue, 425–465 nm, FF01-445/40-50, Semrock; green, 500–550 nm, FF03-5525/50-50, Semrock). Emitted light was

detected with GaAsP photomultiplier tubes (Hamamatsu). The microscope was controlled with ScanImage 2018b (Vidrio). The mouse and ball set-up were placed onto a three-axis stage (Dover) to position the mouse and the field of view. We shielded light leak from the arena and other sources with a cutout black rubber balloon sealing the objective with the aluminium ring mounted onto the headplate.

**Data acquisition.** Imaging data were acquired using ScanImage. Virtual environment data were acquired using ViRMEn. ScanImage and ViRMEn data were synchronized by acquiring triggers on a Digidata analogue to digital converter (Axon Instruments), which was also used to collect raw ball velocities. ViRMEn data were downsampled to the imaging volume rate using the nearest point in time.

For each session, we collected a reference image at the surface of the brain to align across days along the anterior–posterior and medial–lateral axis. We collected a second reference image to align in depth, both across days and to compensate for axial drift over the course of the session. We collected continuous imaging data over the entire behavioural session, typically lasting 50 min.

For imaging experiments, we collected volumes of three planes spaced by 30 μm at 7.5 Hz (one frame blank for flyback), by scanning the objective with a piezo motor (Physik Instrumente). We collected imaging data from the same volume over days for each region in each mouse. jGCaMP7f[47] and tdTomato were imaged together during the behavioural session with 920 nm or 950 nm excitation (40–70 mW). After an imaging session, mTagBFP2[42] and jGCaMP7f were imaged with 850 nm excitation (60 mW).

For photostimulation experiments, we collected data from a single plane at 30 Hz. For each field of view (including different depths), we collected a paired dataset of imaging only on the first day and photostimulation on the second day. We imaged jGCaMP7f alone at 920 nm excitation (40 mW), to minimize excitation of ChRmine[52]. ChRmine-mScarlet[53] intensity was too weak to image under these conditions, so after the session we took a second image of jGCaMP7f and ChRmine-mScarlet with 980–1,000 nm excitation (40 mW), and a third image of jGCaMP7f and mTagBFP2 with 850 nm excitation (60 mW).

**Imaging data preprocessing.** Frames were motion-corrected as previously described[23], the code for which is available at GitHub (https://github.com/HarveyLab/Acquisition2P_class). Raw fluorescence from sources was extracted with Suite2p[54]. Fluorescence traces were baseline-subtracted, with the baseline estimated on a rolling basis (Suite2p baseline=maximin, win_baseline=60, sig_baseline=10), and deconvolved with OASIS[55] (decay constant initialized to 0.8 s and optimized for each source) as previously described[23].

For each channel, we computed the mean image after motion correction. All channels were registered to a common reference image using the green channel in the main imaging session. As the blue channel was dim, and sometimes contained bleed through from the green channel, we demixed the blue channel by estimating the baseline contribution from the green channel. We selected pixels with a green value that was greater than 50% of the maximum green value, and from these selected pixels in the lower 50th percentile of blue values, to select pixels with baseline blue values and extreme green values. After adding in zeros (20% of points) to estimate a line going through the origin, we performed a linear regression on these points, and subtracted this regression from the blue values. We performed the same process to demix green bleed through from the red channel in photostimulation experiments, where the red intensity was relatively low.

We used Fiji[44] to generate the images in each figure.

## Two-photon photostimulation

**Microscope design.** Influence mapping experiments were performed as the mouse performed the T-maze task with a delay on a microscope separate from the imaging experiments that was also equipped with a virtual reality set-up. The imaging path was as described in 'Two-photon imaging'. An independent photostimulation path with a spatial light modulator (SLM) in series with two galvos was used to excite ChRmine using a 1,060 nm laser (repetition rate, 2 MHz, Spark). The power was modulated using a Pockels cell (M350-105BK-02 DRY with 1,060 nm AR coatings for high power, Conoptics). A reflective SLM (HSP-1920-1064) was installed, and the beam was expanded to fill its short axis. Beam polarization was rotated using a half-wave plate (WPH10M-1064, Thorlabs) to maximize diffraction efficiency. The surface of the SLM was imaged onto a 3 mm galvo (6210H, Cambridge) using a telescope (ACT508-400-B and ACT508-100-B, Thorlabs), and the zero-order beam was blocked at the focus of the first lens in the telescope using a piece of aluminium foil glued to a coverglass. Galvo scanners were optically conjugated using a pair of scan lenses (SL50-2P2, Thorlabs) and imaged onto the back aperture of the objective (CFI75, Nikon) using a scan lens (55-S30-16T, Special Optics) and tube lens (MXA20696, Nikon). The imaging and stimulation paths were combined with a 1,000 nm dichroic filter and aligned in ScanImage while imaging a fluorescent pollen slide. To target multiple neurons for stimulation, we used the Gerchberg–Saxton algorithm to compute a phase mask and wrote this to the SLM. Spots were scanned in a spiral pattern as described below. We estimated the diffraction efficiency of the SLM by imaging a fluorescent slide with the SLM and compensated for this difference in efficiency using the weighted Gerchberg–Saxton algorithm.

**Photostimulation protocol.** Spots were manually defined targeting 4–10 *Sst44*+ChRmine+ cells, or an equivalent number of control sites that did not overlap with ChRmine+ cells. These spots were scanned in a spiral pattern with a 6.6 μm radius over 32 ms (one frame) to excite the entire cell. We used 3–5 mW average power per spot. We found that lower powers (1–2 mW) did not produce robust photostimulation with our virus and photostimulation parameters. We also found that, at the minimum power level that produced robust photostimulation, the effectiveness of photostimulation (the degree to which targeted cells were activated) tended to decrease past approximately 100 photostimulation trials. These parameters may represent a limited dynamic range within which this cell type can be photostimulated. We therefore limited our analyses to the first 200 trials (half of which were control trials), which represented the typical number of trials in a behavioural session. Photostimulation occurred on alternating frames (30 Hz frame rate, 15 Hz stimulation rate, 50% duty cycle) and lasted 1 s. Photostimulation was triggered as the mouse performed the task on each behavioural trial just before the T-junction. Sst44-cell-targeting (target) and control trials were randomly interleaved. Furthermore, on each trial we omitted one Sst44 cell from the photostimulation group (in other words, if five cells were chosen per target and control group, four were randomly selected out of that group to be photostimulated on each trial). We chose this design to also measure the influence on cells that were part of the targeted group; however, in the end, these cells were omitted because we selected for cells with a certain number of trials to omit low-confidence influence values.

Of the cells that we targeted, 78% were significantly activated relative to control trials (Wilcoxon signed-rank test, $P < 0.01$ after Bonferroni correction for 137 targets). Of the cells that were significantly activated, our stimulation success rate (change in deconvolved activity above 10% of each cell's 99th percentile of activation) was 75%.

For determining the resolution of photostimulation, we chose isolated ChRmine+jGCaMP7f+ cells and chose photostimulation targets directly over that cell, and at different distances. In this case, we did not apply a phase mask to the SLM and scanned a single spot at these different distances using the galvanometric mirrors.

## Slice electrophysiology

**Viral Injections.** We injected the Sst44-mTagBFP2 virus into mice at postnatal day 14–15 to allow for at least 10 days of viral expression

before the experiment. Virus was injected targeting the PPC at four sites on each side, centred on (−1.7 mm (ML), −2 mm (AP) from bregma) spaced by 0.4–0.5 mm, at 0.25 mm below the dura. Nine mice were used for slice electrophysiology. A separate set of three mice were used for cell fills.

**Acute slice preparation.** Coronal cortical slices were prepared from postnatal day 24–29. Mice were anaesthetized with isoflurane and transcardially perfused with ice-cold choline-based artificial cerebrospinal fluid (choline ACSF: 110 mM choline chloride, 25 mM NaHCO$_3$, 1.25 mM NaH$_2$PO$_4$, 2.5 mM KCl, 7 mM MgCl$_2$, 0.5 mM CaCl$_2$, 25 mM glucose, 11.6 mM sodium-L-ascorbate and 3.1 mM sodium pyruvate, 320–330 mOsm) equilibrated with 95% O$_2$/5% CO$_2$. After perfusion, the brain was rapidly dissected and blocked in ice-cold equilibrated choline ACSF. Tissue was then transferred to a cutting chamber containing ice-cold equilibrated choline ACSF and cut on a Leica VT1200S (300 μm thickness, 0.10 mm s$^{-1}$, 1 mm amplitude, 85 Hz). The slices were then collected in a holding chamber containing ACSF (127 mM NaCl, 25 mM NaHCO$_3$, 1.25 mM NaH$_2$PO$_4$, 2.5 mM KCl, 1 mM MgCl$_2$, 2 mM CaCl$_2$ and 10 mM glucose, 300–310 mOsm). The slices were recovered at 32 °C for 20 min and then maintained at room temperature (22 °C) for 20 min before the start of recordings. AAV infection was assessed by epifluorescence. We recorded from Sst44 cell pairs that were less than 100 μm apart, and less than 400 μm from the pia to target layer 2/3 cells. Experiments were performed within 6 h after cutting.

**Ex vivo slice electrophysiology.** For whole-cell current-clamp recordings, patch pipettes made with borosilicate glass with filament (Sutter BF150-86-7.5) with 3–6 MΩ resistance were filled with a K$^+$-based internal solution (142 mM K-gluconate, 4 mM KCl, 10 mM HEPES, 4 mM MgATP, 0.3 mM NaGTP, 10 mM Na$_2$-phosphocreatine, 1.1 mM EGTA, pH 7.2, 280 mOsm). Recordings were made on an upright Olympus BX51 W1 microscope with an infrared CCD camera (Dage-MTI IR-1000) and a ×60 water-immersion objective (Olympus Lumplan FI/IR 60Å-/0.90 NA). Neuronal tissue was visualized with infrared differential interference contrast. mTagBFP2-expressing Sst44 neurons were identified by epifluorescence driven by a light-emitting diode (Excelitas XCite LED120).

Connectivity measurements were made in current clamp. We injected a hyperpolarizing current into one cell (the driver cell) and measured the resulting change in the membrane voltage of the second cell (the follower cell). For each driver cell, we tuned the hyperpolarizing current (range of −30 to −180 pA) to achieve an approximately −30 mV (range of −19 to −37 mV) deflection in the driver cell. The current step lasted for 600 ms. In total, 20–40 sweeps were collected from each cell pair. This procedure was then repeated in the other direction, injecting current into the second cell and measuring from the first. We initially injected both positive and negative current steps, but then focused on injecting negative current steps to test specifically for gap junctions. Before and after current-clamp recordings to measure connectivity, we measured series resistances in voltage clamp, holding cells at −70 mV and applying a 200 ms −5 pA current step 10 times. If a connection was observed, slices were perfused with a cocktail of synaptic transmission blockers consisting of 10 μM NBQX (Tocris, 1044), 50 μM AP-5 (Tocris, 0106), 10 μM gabazine (Tocris, 1262) to test whether the connection was dependent on synaptic transmission. We perfused the slice with the cocktail for 10 min before measuring the connectivity.

**Cell fills.** In experiments separate from our paired patch recordings, we used a patch pipette loaded with internal solution containing 250 μM Alexa Fluor488 (Life Technologies, A10436) to label single Sst44 cells. After forming a seal and breaking through the cell membrane, cells were held for 15–30 min. Over several minutes, we then slowly retracted the pipette to detach it from the cell body. We then imaged the cell without fixation under a two-photon microscope (see the 'Two-photon imaging' section), taking a stack with 2–5 μm steps. Cell processes were traced by hand in Fiji over a maximum z-projection image with guidance from the full z-stack.

**Data acquisition.** We used an Axon Multiclamp700B to perform voltage and current clamp and low-pass filtering at 4 kHz. Data were sampled at 10 kHz with the Axon Digidata 1440A system. Both instruments were controlled using Clampex10.6 (Molecular Devices). The recorded traces were analysed with Stimfit0.15 and example traces were extracted using Clampfit10.6 (Molecular Devices). All statistical analysis were performed using Prism 9 (GraphPad).

**Analysis.** We measured the connectivity strength as the change in membrane potential in the follower cell (cell without current injection) divided by the change in membrane potential in the driver cell (cell with current injection). The change in membrane potential was defined as the average membrane potential during hyperpolarization (600 ms duration) minus the average membrane potential during a baseline period (60 to 10 ms before current pulse onset). Most cells did not spike at the baseline. For those cells that did spike, sweeps containing action potentials were excluded from analysis of that cell. Significantly connected cells were defined using the Wilcoxon signed-rank test ($P < 0.01$, adjusted post hoc using the Benjamini–Hochberg method). Data were collected from nine mice.

To compute the delay between the follower and driver cell, we selected connections for which the deflection in the membrane potential of the follower cell was greater than 3 mV. To compute the probability of a connection (Fig. 4e), we excluded pairs if either cell had a series resistance higher than 45 MΩ. In the reciprocal connectivity analysis (Fig. 4f), we excluded pairs with a series resistance that differed by more than 30%. For the analysis of synaptic blockers (Fig. 4g,h), we excluded pairs if the series resistance changed by more than 30% after adding synaptic blockers.

To measure the time delay of the connection between Sst44 cells, for each cell, we computed the $P$ value (one-sided Wilcoxon rank-sum test) at each timepoint by comparing a sliding 2 ms window to an equivalent window centred at 1.5 ms before the pulse onset, and computed the average time delay between the log[$P$] curves (between natural log[$P$] values of −5 and −10). Note that any analysis is limited by the signal to noise ratio, which can be significant when analysing small deviations in membrane voltage. We therefore chose recordings in which the deflection in the follower cell was greater than 3 mV. Owing to this noise limitation, the delays reported here should be interpreted as an upper bound.

## Targeting optogenetic inhibition to Sst44 cells during behaviour

**Characterization with extracellular electrophysiology.** We injected an AAV with stGtACR2[56] driven by the Sst44 enhancer in two C57Bl/6J male mice at 2–3 sites spaced anterior-posterior by 0.4 mm centred on (−1.7 mm (ML), −2.0 mm (AP) from bregma, 0.3 mm below the dura, 70 nl per site, 1 × 10$^{12}$ GC per ml in PBS). The injection pipette was inserted through a small craniotomy and was angled at 30° from the horizontal such that the injection site was under the bone. For recordings, we removed the bone above the injection sites. On a rig in which the mouse was head-fixed and awake, we inserted a 32-channel silicon probe coupled to an optic fibre (A1x32-Poly2-5mm-50s-177-OA32LP, Neuronexus) near the injection sites, and advanced the probe such that the optic fibre was touching the dura. After insertion, we added 2% agarose in PBS to stabilize the brain. Recordings were amplified using a headstage amplifier (RHD2132, Intan Technologies) and digitized at 20 kHz (512ch recording controller, Intan Technologies) operated using the Intan RHX Data Acquisition Software (v.3.1.0). We used Kilosort (v.2.5)[57] with the default parameters to detect spikes. We pooled spikes from all of the channels without sorting as a measure of overall circuit activity.

Unfortunately, we did not have a way to test the direct effect of stG-tACR2 activation on Sst44 cell spiking because we could not identify Sst44 cells using this method (with an excitatory opsin, one can identify time-locked, short-latency responding cells, but with an inhibitory opsin, one needs a high baseline firing rate to identify cells that are inhibited with a short latency). We therefore instead measured the effect of stGtACR2 activation in Sst44 cells on spiking in the entire population of neurons recorded on the multichannel electrode. As Sst cells are inhibitory, we expect that inhibiting these cells will disinhibit the circuit. As expected, we observed an increase in the population activity when we delivered blue light. This experiment indicates that the stG-tACR2 activation worked to some degree. However, as we were unable to measure Sst44 cell spiking directly, it remains unclear whether these cells were completely silenced or just experienced reduced spiking. Moreover, we did not measure the effect on Sst44 cell spiking during error-correction events when these cells are driven very strongly, and when it would be more difficult to fully inhibit these cells. In particular, as these cells have electrical coupling to one another, it may be hard to silence them, especially if the whole, gap-junction-coupled population is not sufficiently silenced through direct optogenetic silencing. For example, it remains possible that some Sst44 cells were not transduced with the virus or had low expression of stGtACR2. Thus, despite our efforts to develop and validate this approach, we remain uncertain of the extent of inhibition of Sst44 cell activity by stGtACR2, especially during behaviour.

**Optogenetic inhibition of Sst44 cells during behaviour.** We injected Sst44-stGtACR2 AAV bilaterally in the PPC (four sites spaced by 0.5 mm centred on 1.7 mm (ML), −2.0 mm (AP) from bregma, 0.3 mm below the dura, 70 nl per site, $1 \times 10^{12}$ GC per ml in PBS) in 6 C57Bl/6J male mice. We replaced the skull above the injection sites with two glass windows (one on each hemisphere) to allow light to enter the brain. After training the mice to perform the T-maze task in which the cue was omitted in the second half of the maze (delay task), we performed bilateral inhibition experiments during heading pulse perturbations as described in Fig. 6, and on interleaved control trials without a heading perturbation. We focused on the heading perturbation because the mouse must course-correct during these times, and because we expect Sst44 activity to be strong during these error corrections. We used a 470 nm laser (LRD-0470-PFR-00200, Laserglow Technologies) to deliver blue light and directed the laser beam (1 mm in diameter at the brain surface) using a galvanometric mirror pair (6210H, Cambridge Technology) as previously described[48] to either the PPC (1.7 mm (ML), −2 mm (AP) from bregma), or control sites on the dental cement, alternating each side at 40 Hz with a 50% duty cycle. We used twice the average power density as in our electrophysiology experiments to help to ensure that the opsin was activated, as we often observe dura growth under the window after significant periods of time, such as after months of training. On 50% of trials, we started photoinhibition halfway through the maze, which was when the delay period started, and was also when the heading pulse was triggered on a random subset of trials (30%). We terminated photoinhibition after 5 s or once the mouse reached the intertrial interval.

This experiment aims to test whether Sst44 cell activity is required for the mouse to execute a course correction. The outcome of these experiments is that we did not observe a difference in the mouse's behaviour on trials in which we targeted the blue laser light to the PPC compared with interleaved trials in which we targeted the laser to control sites. It is possible to interpret this negative result in several ways, some of which result to technical challenges and other to biological interpretations. A first possible interpretation is that we did not inhibit Sst44 cell spiking sufficiently (see more details above). A second possibility is that Sst44 cells in the PPC contribute to the execution of course corrections, but that other cells in other cortical or subcortical regions can also drive this behaviour. That is, there may be redundancy across brain areas and,

therefore, silencing only the PPC's Sst44 cells might not have silenced the entire relevant population. Relatedly, it is possible that the silencing of Sst44 cells was heterogeneous in the PPC itself. For example, Sst44 cells in deeper layers may not have been as strongly inhibited given that we expect there to be less light delivered to deeper layers. If not all Sst44 cells were adequately silenced, the remaining active population may be sufficient to carry out the behaviour. A third possibility presents an interesting biological interpretation: that Sst44 cells are not required to perform the error corrections after they have been learned and that instead these cells serve a function in learning to correctly navigate towards the reward. This latter possibility was not tested in our optogenetics experiments as we focused only on well-trained mice. Further experiments will help to distinguish between these possibilities.

### Activity analysis

Data analysis was performed using Python 3. Imaging and behaviour data from each session were imported into the anndata object[58]. Although designed for single-cell sequencing data, this data structure was convenient for imaging data. We used the 'X' central matrix to hold the session activity for each cell, the 'var' matrix to carry metadata for each cell (that is, channel values, cell type and so on) and the 'obs' matrix to carry time-varying information, including behavioural and task variables. The 'uns' dictionary carried metadata and other unstructured information.

**Data inclusion criteria.** Sources were classified as cells or non-cells with a three-layer convolutional neural network trained on manually labelled sources that has been previously described[28]. Moreover, we excluded sources for which raw fluorescence values exceeded 95% of the digitized dynamic range (to ensure values were not saturated and cells were not over-expressing jGCaMP7f), and sources that were near to the edge of the field of view.

For imaging experiments, we included sessions in which the mice performed the task with >85% correct trials. In imaging cohort 1 (7 mice), this resulted in 27 sessions for the PPC (17,254 cells total), and 24 sessions for the RSC (23,717 cells total). In imaging cohort 2 (8 mice), this resulted in 16 sessions for the PPC (7,940 cells total). For reward-omission experiments, this resulted in 6 sessions from 4 mice (2 mice from each imaging cohort) in the PPC (3,045 cells total). Total cells represent the total cell count across all sessions, not independently sampled cells, as we imaged activity from the same field of view for a given region in each mouse.

For photostimulation experiments, we included sessions in which mice performed the task with >80% correct trials. This resulted in 20 sessions across 4 mice in the PPC (6,793 cells total). These cells were independently sampled.

**Classifying cell types.** For each cell, we computed the background-subtracted intensity for each channel by taking the mean intensity within that cell's spatial mask and subtracting the mean intensity in a two-pixel outline around the mask. Cell types were called on a relative basis. Blue cells (*Sst44*⁺) were defined as having a background-subtracted intensity of greater than 15% of the maximum blue value, defined as the mean of the top three cells. Red cells (Sst⁻Cre⁺) were defined in the same way, except we also included cells that had a spatial correlation of >0.7 with the closest cell mask defined by Cellpose (v.0.0.2.8, default cyto model). For photostimulation experiments, we used a slightly more lenient threshold of 10% the maximum red value (defined again as the mean of the top 3 cells) and 0.6 for the spatial correlation with Cellpose masks, and then manually removed negative cells (-10%) as, in this case, we observed false-positives from sparse neuropil-expressing soma membrane-localized ChRmine-mScarlet.

**Activity normalization and smoothing.** Different cells expressed different levels of jGCaMP7f, resulting in different absolute changes

in fluorescence for each cell. We therefore normalized deconvolved activity by dividing by the 99th percentile of activity within the session for each cell. Unless indicated, all analyses were computed on these normalized activity values, including the mean activities for each cell type. In some cases, indicated in each analysis, we also smoothed each cell's activity over time with a 0.25 s Gaussian filter.

**Correlation analysis.** We computed the Pearson correlation between the smoothed activity of each cell, generating a correlation matrix. We then computed the average correlation between cells of different cell types (excluding the diagonal for self-correlations). In Extended Data Fig. 2b,c, instead of computing the correlation between individual cells, we computed the correlation between population means, to assess whether Sst44 cell activity was correlated to overall circuit activity.

**UMAP analysis.** UMAP projections[41] were computed on the smoothed activity of each cell. Note that, because the features here are activity measurements at different points in time, we cannot project cells from different sessions together using this method. UMAP parameters were as follows: Version=0.5.1, n_neighbors=10, min_dist=0.1, metric='euclidean'.

**Nearest-neighbour analysis.** We computed a nearest neighbour graph ($k = 10$) using the smoothed activity of each cell (smoothed deconvolved session activity, not the UMAP projection). We then computed for each cell the fraction of nearest neighbours that were $Sst44^+$ and averaged this fraction over cells from each cell type.

**Clustering analysis.** To assess what fraction of $Sst44^- Sst^+$ cells co-clustered with Sst44 cells, we clustered all cells on the basis of their activity using the Leiden community clustering algorithm (resolution=6, n_neighbors=5), selected the cluster that was most populated by Sst44 cells and examined how many of the $Sst44^- Sst^+$ neurons fell into that cluster. We performed this analysis for one PPC session from each mouse.

**Influence analysis.** We analysed cells that were >40 μm from the nearest photostimulation site. From these, we also removed cells that overlapped with pixels contiguous with target sites in the binarized image of ChRmine-mScarlet, to remove cells with ChRmine-expressing proximal dendrites that overlapped with a photostimulation site. We also removed cells that overlapped with pixels contiguous with target sites in the binarized photostimulation triggered d$F/F$ image on target relative to control trials, removing cells that overlapped with the dendrites of directly stimulated cells. Finally, we also chose cells with >20 trials from which to compute influence. Following previous work[23], we computed influence as the mean change in deconvolved activity on target trials minus the mean change in deconvolved activity on control trials, divided by the s.d. in deconvolved activity on control trials. We computed the mean change in deconvolved activity as the mean after photostimulation (0.5 to 1.0 s relative to the start of photostimulation, where we observed maximum influence) minus the mean before photostimulation (−1.0 to 0 s relative the start of photostimulation).

**Computing heading deviation.** To compute the mouse's heading deviation, we computed the difference between its heading at each timepoint and its heading at the same point along its mean smooth trajectory. To estimate the mean smooth trajectory, we selected the shortest 25% of trials for each world (black–left or white–right) in each session. From these trials, we computed the median position, and the circular mean heading of the mouse at binned distances from the reward zone. We used 15 bins that evenly tiled the log-transformed distance from the reward zone. We used the log-transformed distance because this approach produces more bins at the end of the maze near the T-intersection where there are large changes in heading. We linearly interpolated this coarse trajectory at a resolution of 0.75 cm and aligned each session timepoint to the nearest point along the interpolated trajectory to assign to each timepoint the heading under a smooth trajectory. In a small number of sessions, the mean smooth trajectory included bins where the mouse was facing away from the reward zone at the end of the T-arms (sometimes the mouse would turn too much and overshoot, even on its shortest trials). We assigned to these bins the last heading that was facing towards the reward zone.

**Triggered analyses.** We triggered either on activity or behavioural events by identifying the rise time of when a variable exceeded a threshold defined in each analysis. We omitted events that were separated by less than 5 s, before plotting each metric with a 5 s window before and after. We also omitted events for which the rise time fell within the intertrial interval. We smoothed the turning acceleration and mean activity signals with a 0.25 s Gaussian filter for triggering and selecting trials. Raw signals are plotted.

We pooled trials across sessions for these triggered analyses. Note that, although cells were not independent for trials from different sessions from the same mouse, the same is also true for trials within the same session. To compare more data, we therefore compiled trials across sessions, conceptually treating trials in the same and different sessions as equivalent.

When triggering on Sst44 cell activity, we triggered on the average activity of Sst44 neurons imaged in a given session (smoothed Sst44 cell activity < 0.4).

When triggering on entering the T-junction, we triggered on the position ($y = 225$ cm) at which the maze widens slightly ($y = 225$–229 cm) before entering the maze arms ($y = 229$–231 cm). When splitting these events on the basis of heading deviation (Fig. 5h), a large deviation was defined as >π/6 and a small deviation as <π/12 at +1.5 s after the trigger. When splitting high-deviation trials by turning acceleration (Fig. 5i), a strong correction was defined as >1 rad s$^{-2}$ in the opposite direction as the heading deviation, and a weak correction as <0.5 rad s$^{-2}$, at +1.5 s after the trigger.

When triggering on turning accelerations (Fig. 5j), we triggered on turning accelerations (>1 rad s$^{-2}$), with a high turning velocity (>0.5 rad s$^{-1}$) to match the turning profile across conditions, and split by large (>π/6) and small (<π/12) heading deviations at 0 s.

When triggering on heading perturbations (Fig. 6), strong corrections were defined as >1 rad s$^{-2}$ in the opposite direction as the perturbation, and weak corrections as <0.5 rad s$^{-2}$ at +2 s after the trigger.

When triggering on heading deviations, we used a 0.3 s delay to parse trials based on turning acceleration (we also used this method to define course-correction rates in Extended Data Fig. 5 and Extended Data Fig. 7j,k). We used this delay to account for the average delay in turning acceleration when triggering on heading deviation with the threshold of π/6 used in these analyses. Note that the heading deviation continues to increase after this point and, on average, peaks approximately with the peak in turning acceleration. We therefore do not use this delay when parsing triggered data in other analyses. For analyses without heading perturbations, we selected sessions with >85% correct trials to increase the likelihood that the mouse was attempting to follow a desired trajectory towards the reward zone. The heading perturbation was triggered at the halfway point in the maze ($y = 113$ cm). We triggered at the same position on control trials interleaved with heading perturbations.

## Statistics
Statistics are indicated in each figure and are all two-sided unless otherwise noted.

## Reporting summary
Further information on research design is available in the Nature Portfolio Reporting Summary linked to this article.

## Data availability

Sequencing data are available at the Gene Expression Omnibus (GSE232200). Other data are available on request.

## Code availability

Code is available on request.

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

**Acknowledgements** We thank G. Maimon, S. Jereb, L. Orefice, G. Fishell and the members of the Harvey laboratory for comments on the manuscript; D. Paul for advice on detecting gap junctions; and A. Emanuel and D. Aponte for help with extracellular electrophysiology. FACS was performed at the Harvard Systems Biology FACS Facility. Custom AAVs were produced by the Boston Children's Hospital Viral Core (NEI P30 grant 5P30EY012196). Confocal imaging was performed at the Harvard Neurobiology Imaging Facility (NINDS P30 Core Center grant NS072030). The Harvard Medical School Neurobiology Research Instrumentation Core and Machine Shop were used for the construction of rig components. Research reported in this publication was supported by a Lefler Fellowship to J.G., a Life Sciences Research Foundation Fellowship sponsored by Vertex Pharmaceuticals to J.G., a Harvard Brain Science Initiative Postdoc Pioneers Grant to J.G., a Long-Term Fellowship from the Human Frontier Science Program to L.T., a Brain Initiative Fellowship (NIMH, F32MH118698) to D.E.W., a NIH Director's Pioneer Award (DP1 MH125776) to C.D.H., NIH grants R01 NS089521 and R01 NS108410 to C.D.H., and NIH grants RF1 DA048787 and R01 NS028829 to M.E.G.

**Author contributions** J.G., C.D.H. and M.E.G. conceived the study. J.G. and C.A.B. performed ATAC experiments. C.A.B. performed RNA hybridization experiments. J.G., C.A.B., T.K. and J.S. trained and imaged mice. J.G. performed photostimulation experiments. L.T. and J.G. performed patch-clamp electrophysiology experiments. J.D. and J.G. performed single-cell filling experiments. C.A.B. and J.G. analysed RNA hybridization experiments. L.T. and J.G. analysed electrophysiology experiments. J.G. analysed all other experiments with input from C.D.H., C.A.B. and M.E.G. D.E.W. designed and helped to implement photostimulation instrumentation. S.H. shared the *Sst44* enhancer before publication and provided feedback on analyses. J.G. and C.D.H. wrote the paper with input from all of the authors.

**Competing interests** S.H. is a member of the scientific advisory board and M.E.G. is a consultant for Apertura Gene Therapy. The other authors declare no competing interests.

**Additional information**
**Correspondence and requests for materials** should be addressed to Jonathan Green or Christopher D. Harvey.

## ATAC-seq

**a** Leiden clustering

Pvalb

Sst

Lamp5

Scng

Vip

Sst gene accessibility
max: 60
Inhibitory cells

Pvalb gene accessibility
max: 82

Vip gene accessibility
max: 67

Lamp5 gene accessibility
max: 62

Sncg gene accessibility
max: 17

## RNAScope

**b**

[AAV9] Sst44-BFP

Sst44-NLS-mTagBFP2

[AAV9] Sst44-BFP
+Subtype probe

Calb2 probe

Hpse probe

Chodl probe

Crh probe

[AAV9] Sst44-BFP
+Subtype probe
+Sst probe

Sst probe

Sst (b)

100 µm

**c** ● Sst44+ nuclei  ● Sst+ cell
● Sst+ and Calb2+ cell

● Sst+ and Hpse+ cell

L1
L2/3
L4
L5
L6

200 µm

● Sst+ and Chodl+ cell

● Sst+ and Crh+ cell

L1
L2/3
L4
L5
L6

**d**

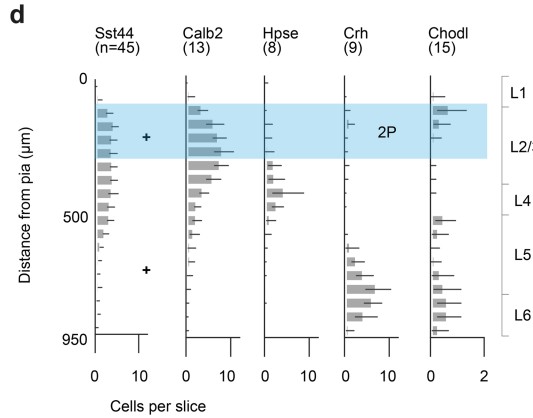

Sst44 (n=45), Calb2 (13), Hpse (8), Crh (9), Chodl (15)

Distance from pia (µm)

0
L1
L2/3
2P
+
500
L4
L5
+
L6
950

Cells per slice
0   10   0   10   0   10   0   10   0   2

**Extended Data Fig. 1 | Molecular characterization of Sst44 cells. a**, UMAP projection of the genomic accessibility of 10,375 inhibitory cortical neurons from posterior cortex, showing Leiden clustering as different colours and the accessibility of gene markers for the 5 major inhibitory cell classes. Data pooled from two mice. **b**, Example images of native Sst44 enhancer-driven mTagBFP2 fluorescence and RNAScope for Sst and Sst subtype marker genes. **c**, Spatial distribution of Sst44 enhancer-labelled cells and Sst subtypes based on *in situ* RNA labelling. **d**, Depth distribution of Sst44 cells and Sst subtypes across slices. + denotes injection depths. Blue region highlights 2P imaging depth in subsequent figures.

**Correlation between population mean activity of cell types**

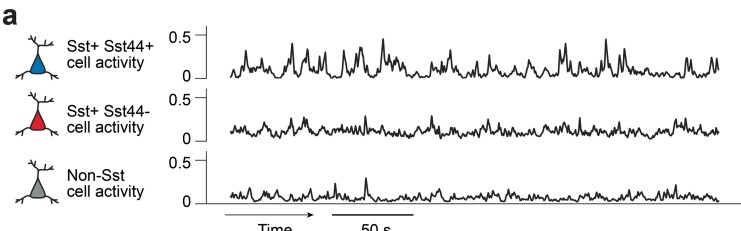

**a**

Sst+ Sst44+ cell activity

Sst+ Sst44- cell activity

Non-Sst cell activity

Time  50 s

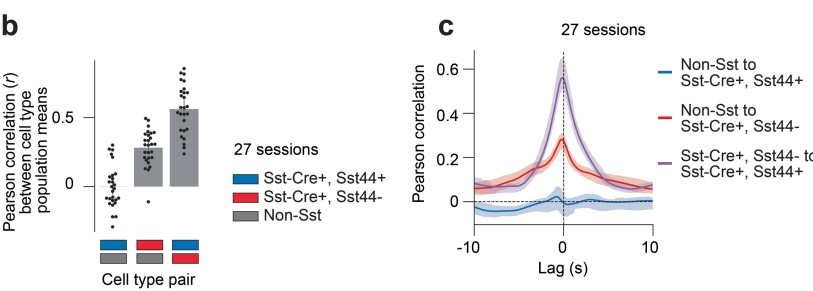

**b** Pearson correlation (*r*) between cell type population means

27 sessions
■ Sst-Cre+, Sst44+
■ Sst-Cre+, Sst44-
■ Non-Sst

Cell type pair

**c** 27 sessions

Pearson correlation

— Non-Sst to Sst-Cre+, Sst44+
— Non-Sst to Sst-Cre+, Sst44-
— Sst-Cre+, Sst44- to Sst-Cre+, Sst44+

Lag (s)

**Average correlation between single neuron activities from each cell type**

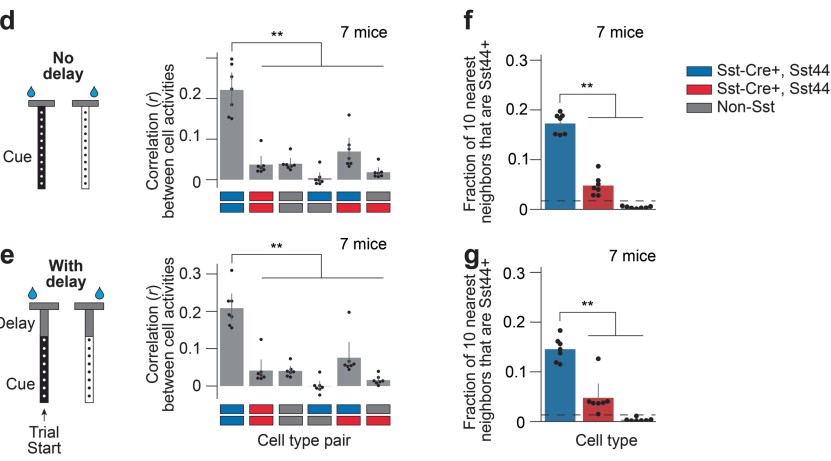

**d**

**No delay**

Cue

Correlation (*r*) between cell activities

** 7 mice

**e**

**With delay**

Delay

Cue

Trial Start

Correlation (*r*) between cell activities

** 7 mice

Cell type pair

**f** 7 mice

Fraction of 10 nearest neighbors that are Sst44+

**

■ Sst-Cre+, Sst44+
■ Sst-Cre+, Sst44-
■ Non-Sst

**g** 7 mice

Fraction of 10 nearest neighbors that are Sst44+

**

Cell type

**Extended Data Fig. 2 | Sst44 cells are correlated with each other, but not with overall circuit activity. a–c**, Sst44 cell activity is weakly correlated with overall circuit activity. **a**, Sample trace of mean population activity of Sst44+, Sst44-/Sst+ and non-Sst cells in PPC. **b–c**, Pearson correlation between population means of each cell type across sessions (**b**), and at different time lags (**c**). **d–g**, Similar correlation and clustering statistics between mazes that contain or do not contain a delay between the visual cue and T-intersection. **d–e**, Pearson correlation between cells of each cell type. **f–g**, Fraction of 10 nearest neighbours in activity space that are Sst44+. ** p < 0.01, Kolmogorov-Smirnov test. Mean and bootstrapped 95% confidence intervals are shown. Activity was smoothed with a 0.25 s gaussian filter for these analyses.

## Influence mapping using *in vivo* optogenetics during task performance

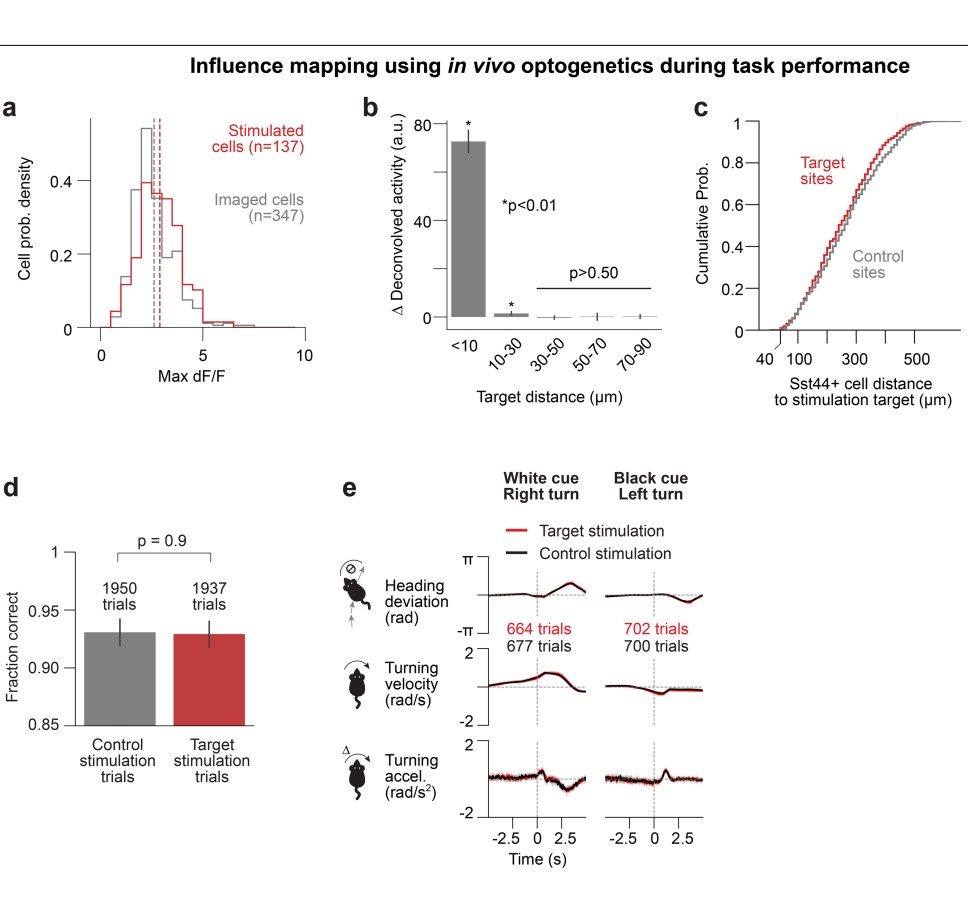

## Ex vivo patch clamp electrophysiology

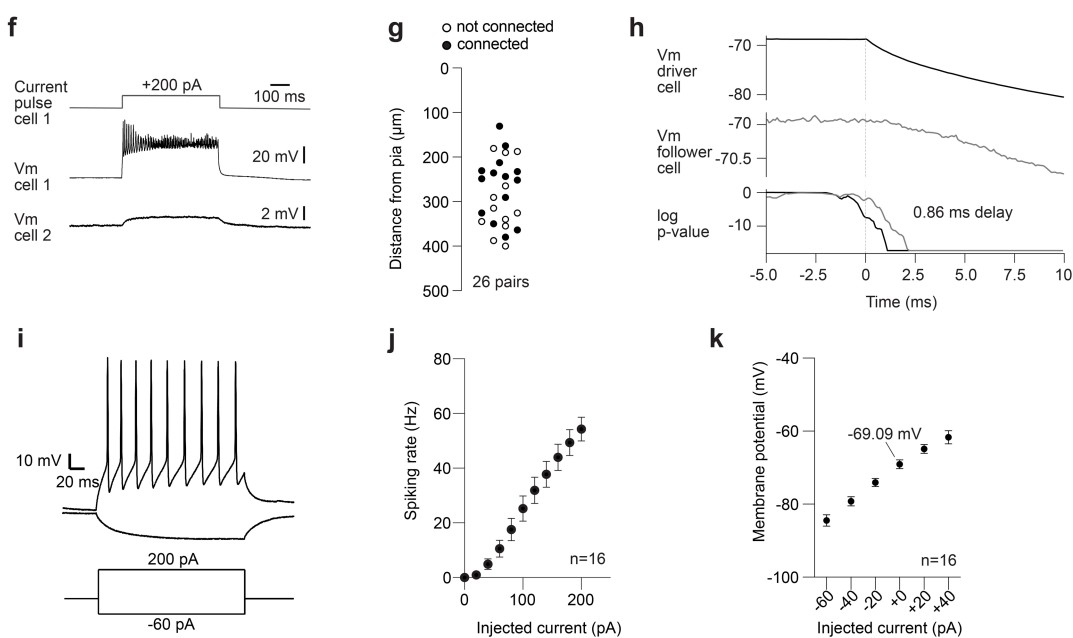

**Extended Data Fig. 3** | See next page for caption.

**Extended Data Fig. 3 | Characterization of *in vivo* influence mapping and slice electrophysiology experiments. a**, Distribution of peak dF/F of Sst44 cells during photostimulation sessions (n = 20) and during paired sessions from the same field of view without photostimulation (n = 20). Dashed lines indicate the mean of each distribution. 4 mice. **b**, Change in deconvolved cell activity as a function of the photostimulation target distance. p-values evaluated using the Wilcoxon signed rank test vs zero across trials. Number of trials: < 10 μm: 600, 10–30 μm: 600, 30–50 μm: 800, 50–70 μm: 350, 70–90 μm: 300. 12 isolated cells, 6 mice. Mean and bootstrapped 95% confidence intervals are shown. **c**, Distribution of target and control sites from influenced (non-stimulated, >40 μm from nearest target) Sst44+ cells. 141 target sites and 141 control sites total. The total number of target sites is slightly larger than the number of stimulated Sst44 cells (n = 137) because some photostimulated cells were not detected by the cell detection algorithm if they were not successfully stimulated. **d**–**e**, Stimulating 4–10 Sst44 cells in PPC does not change the mouse's choice or turning behaviour. **d**, Fraction correct on control and target trials. p-value evaluated with Wilcoxon rank-sum test across trials. **e**, Turning behaviour triggered on photostimulation on target and control trials. Mean and bootstrapped 95% confidence intervals are shown. 4 mice, 20 sessions. **f**–**k**, Characterization of Sst44 cell electrophysiology. **f**, Mean membrane voltage from an example connected pair of Sst44 cells shown with a positive current pulse. **g**, Distribution of distance of cell pairs from pia. **h**, Example trace showing how we computed the delay in the membrane voltage deflection between the driver and follower cell. For each cell, we computed the p-value (Wilcoxon rank-sum test, one-sided) at each time point by comparing a sliding 2 ms window to an equivalent window centred at 1.5 ms before the pulse onset and computed the average time delay (horizontal distance between the two log(p) curves) between natural log(p) values of −5 and −10. Natural log is plotted. See Methods for more details. **i**–**k**, Characterization of intrinsic excitability. **i**, Example trace showing membrane voltage in response to a positive and negative current pulse. **j**, Spiking rate as a function of injected current. **k**, Membrane potential as a function of injected current. Mean and s.e.m. are shown in **j**–**k**.

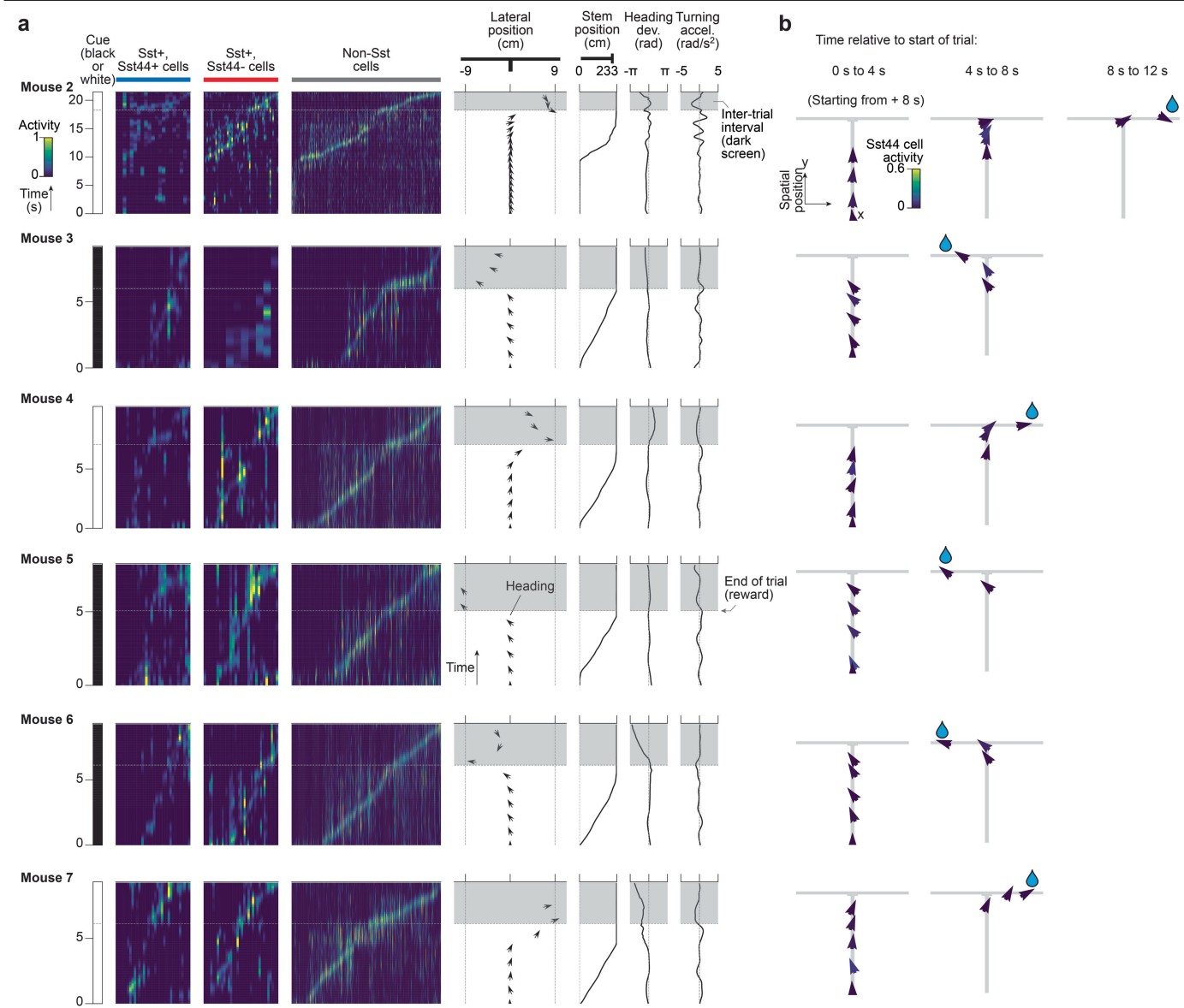

**Extended Data Fig. 4 | Smooth navigation behaviour during example trials with low Sst44 cell activity for each mouse. a**, Activity and behaviour over time during trials with low Sst44 cell activity. Same examples as in Supplementary Video 1 (mouse 1 shown in Fig. 5b). Each column in the activity heatmaps represents the activity of one cell over time. Cells sorted based on peak activity along the maze. Smoothed activity is plotted. Black arrows indicate the lateral position and heading of the mouse over time (pointing up is toward the end of the maze). **b**, Activity as a function of the mouse's trajectory. X and y length scales are not proportional.

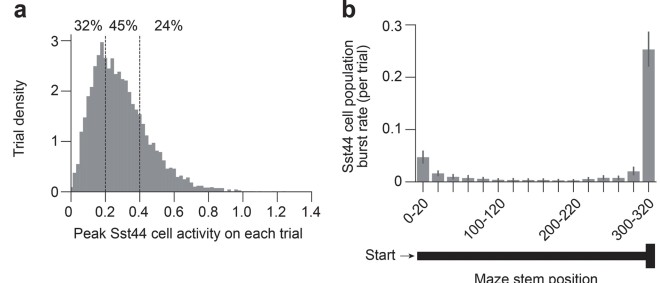 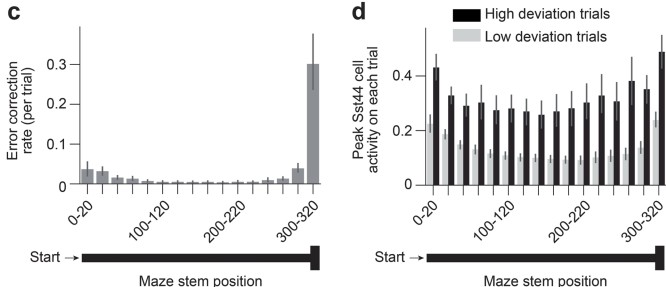

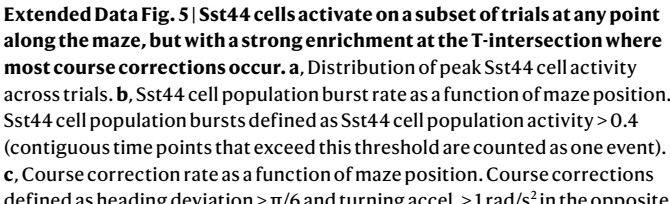

**Extended Data Fig. 5 | Sst44 cells activate on a subset of trials at any point along the maze, but with a strong enrichment at the T-intersection where most course corrections occur. a**, Distribution of peak Sst44 cell activity across trials. **b**, Sst44 cell population burst rate as a function of maze position. Sst44 cell population bursts defined as Sst44 cell population activity > 0.4 (contiguous time points that exceed this threshold are counted as one event). **c**, Course correction rate as a function of maze position. Course corrections defined as heading deviation > π/6 and turning accel. > 1 rad/s² in the opposite

direction, delayed by +0.3 s (to account for the average delay in the mouse's reaction – see Methods). **d**, Peak Sst44 cell activity (after smoothing) for each spatial bin in each trial, splitting based on whether there was a high (> π/2) or low (< π/4) heading deviation, showing that Sst44 cell activity is strongly modulated by heading deviation at any point along the maze. Wilcoxon rank-sum test across sessions, high vs low deviation, p < 1e-6 for each spatial bin. Mean and bootstrapped 95% confidence intervals are shown. 7 mice, 27 sessions.

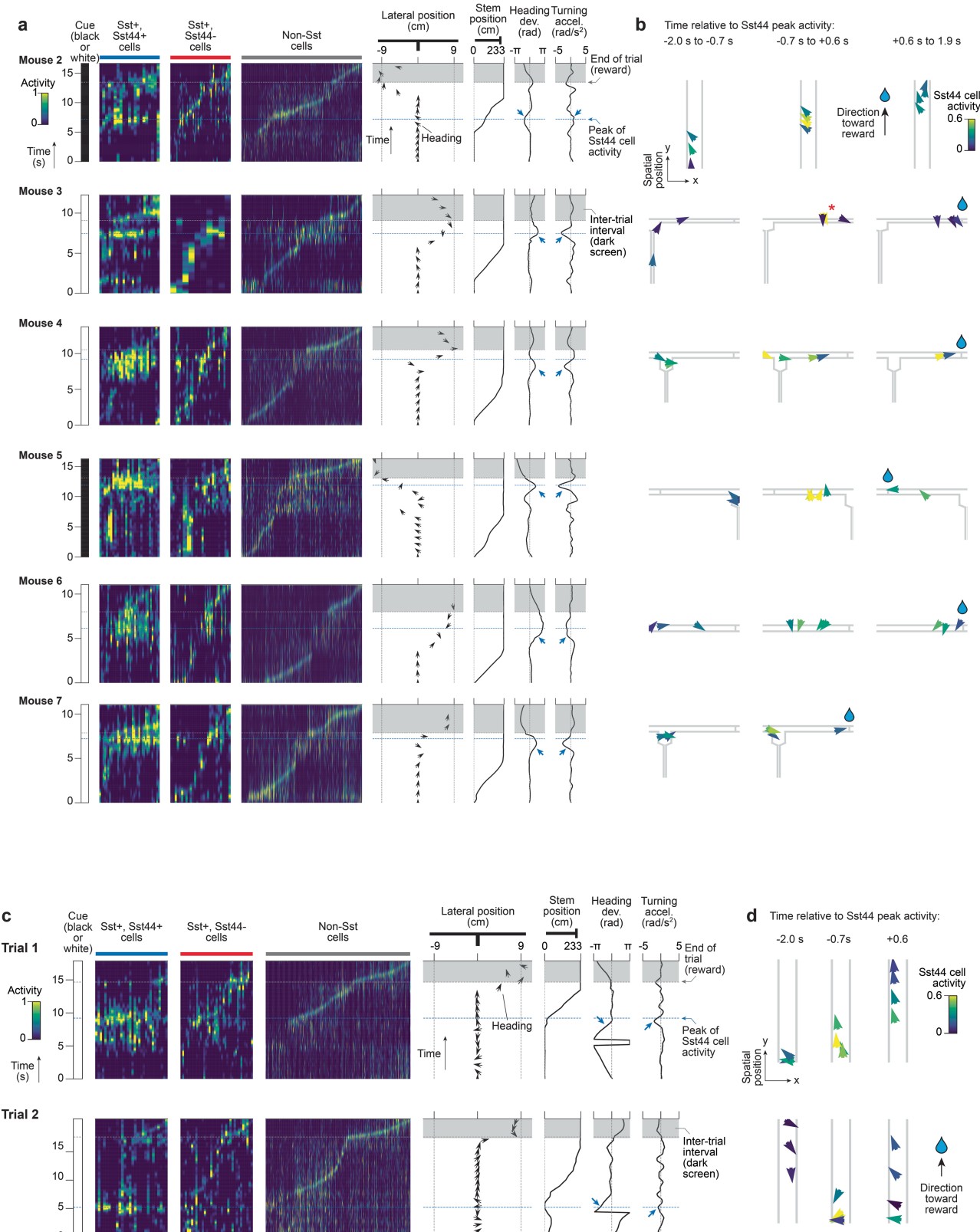

**Extended Data Fig. 6 | Course correction behaviour during example trials with high Sst44 cell activity for each mouse. a**, Activity (smoothed) and behaviour over time during trials with high Sst44 cell activity for each mouse. Same examples as in Supplementary Video 2 (mouse 1 shown in Fig. 5c). Each column in the activity heatmaps represents the activity of one cell over time. Cells sorted based on activity along the maze. Black arrows indicate the lateral position and heading of the mouse over time (pointing up is toward the end of the maze). **b**, Activity as a function of the mouse's trajectory surrounding the peak in Sst44 cell activity (t = 0 is the Sst44 cell activity peak). X and y length scales are not proportional. * highlights arrow with high Sst44 cell activity. **c–d**, Same as **a–b**, for example peaks of Sst44 cell activity during course correction events where turning acceleration does not oppose heading deviation. Same examples as in Supplementary Video 3.

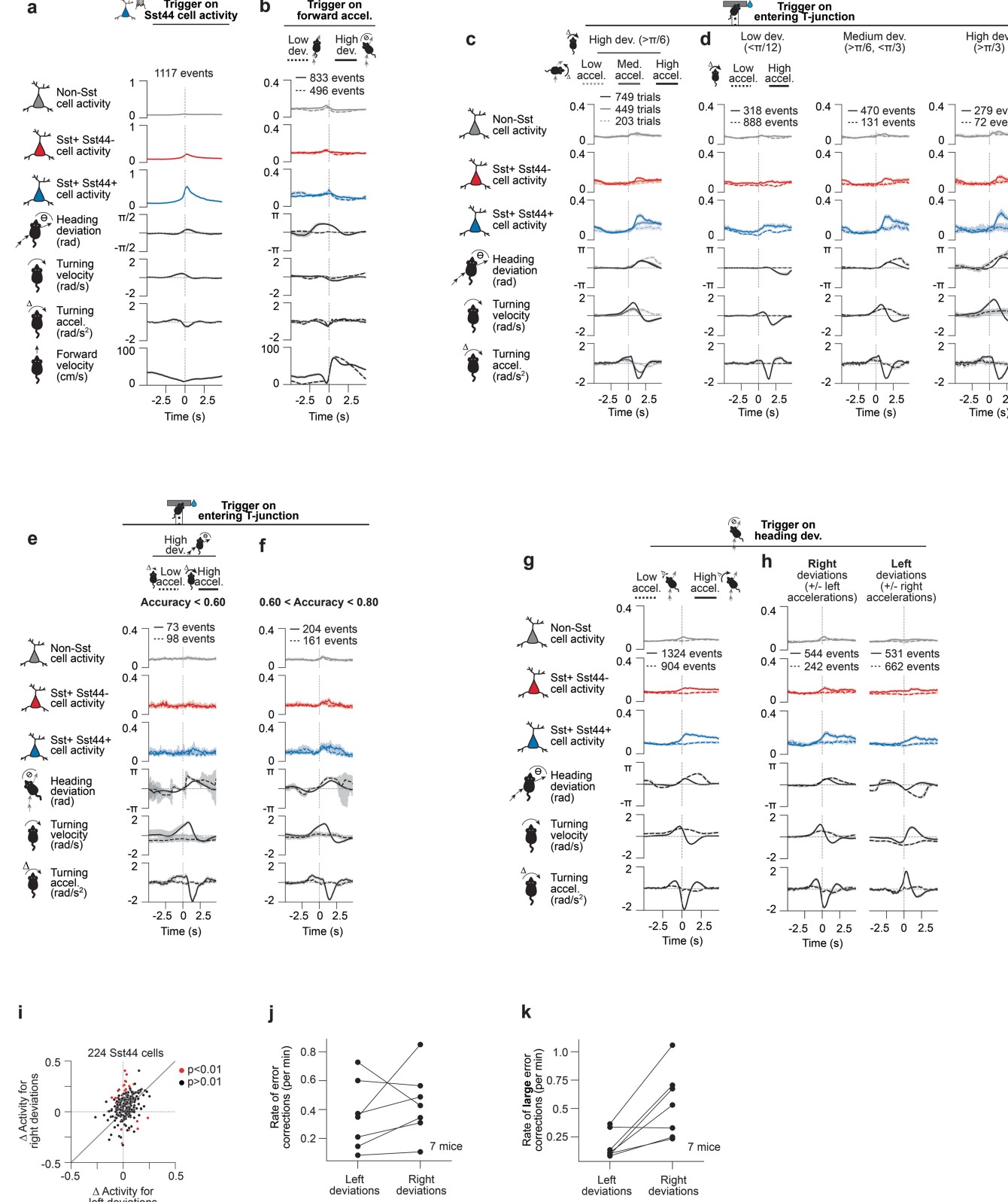

**Extended Data Fig. 7** | See next page for caption.

**Extended Data Fig. 7 | Characterization of activity during forward movements, errors and corrections of varying magnitudes, error corrections during training, and corrections in different directions.**
**a**–**b**, Slow down and speed up in forward velocity before and after Sst44 cell activity, but weak contribution of forward velocity to Sst44 cell activity. 7 mice, 27 sessions. **a**, Activity and behaviour, including forward velocity, averaged over large bursts of Sst44 cell activity (smoothed Sst44 cell activity > 0.4).
**b**, Activity and behaviour during sharp increases in forward running (forward accel. > 70 cm/s$^2$), split by high (> $\pi/6$) and low (< $\pi/12$) heading deviations. Left and right deviations were pooled after inverting behaviour for left deviations.
**c**–**d**, Sst44 cell activity as a function of turning acceleration and heading deviation magnitude. 7 mice, 27 sessions. **c**, Activity and behaviour as the mouse entered the T-junction with a large (>$\pi/6$) heading deviation, split based on whether the mouse corrected with a low (< 0.5 rad/s$^2$), medium (>0.5, <1 rad/s$^2$), or high (>1, <2 rad/s$^2$) turning acceleration in the opposite direction. Left and right deviations were pooled after inverting behaviour for left deviations. Wilcoxon rank-sum test across trials, Sst44 cell activity high vs medium accel.: p = 6e-11, medium vs low accel.: p = 6e-5. **d**, Activity and behaviour as the mouse entered the T-junction with a small (<$\pi/12$), medium (>$\pi/6$, <$\pi/3$) or large (>$\pi/3$) heading deviation, split based on whether the mouse corrected with a high (>1, <2 rad/s$^2$) or low (<0.5 rad/s$^2$) turning acceleration in the opposite direction. High turning accelerations were capped at 2 rad/s$^2$ to better compare across conditions. Left and right turning accelerations were pooled after inverting behaviour for right accelerations. Wilcoxon rank-sum test across trials, Sst44 cell activity solid line, low vs medium or high deviation: p < 1e-8, medium vs high deviation: p = 0.64. Activity averaged over 0.5 to 2.5 s relative to −1 to 0 s was used for statistical analyses for **c**–**d**. **e**–**f**, Sst44 cell activity during training. **e**, Activity during training for sessions with low accuracy (< 0.6). At this stage of training, a landmark indicates the location of the reward. Activity and behaviour as the mouse entered the T-junction with a large (>$\pi/6$) heading deviation, split based on whether the mouse corrected with a low (< 0.5 rad/s$^2$), or high (>1 rad/s$^2$)

turning acceleration in the opposite direction. Left and right deviations were pooled after inverting behaviour for left deviations. 3 mice, 4 sessions.
**f**, Same as **e**, for sessions with intermediate accuracy (> 0.6, < 0.8). 5 mice, 7 sessions. Wilcoxon rank-sum test across trials, Sst44 cell activity solid vs dashed line: p = 0.5 (**e**), p = 7e-3 (**f**), solid line **f** vs **e**: p = 0.02. Activity averaged over 0.5 to 2.5 s relative to −1 to 0 s was used for statistical analyses for **e**–**f**.
**g**–**k**, Sst44 cell activity is present during leftward and rightward course corrections, even though course corrections occurred more often in response to right deviations. 7 mice, 27 sessions. **g**, Activity and behaviour during heading deviations (> $\pi/6$) at any point in the maze, split by whether the mouse corrected with a strong (>1 rad/s$^2$) or a weak (< 0.5 rad/s$^2$) opposing turning acceleration delayed by +0.3 s (Methods). Left and right deviations were pooled after inverting behaviour for left deviations. **h**, Same as **g**, showing left and right deviations separately. In this analysis we additionally capped the heading deviation at $\pi/3$ and the turning acceleration at 2 rad/s$^2$ to better compare activity between left and right deviations. **i**, Mean change in activity (0 to 3 s versus −2 to −1 s in **h**) in single Sst44 cells in response to corrections for left and right deviations. We selected errors and corrections within the same range as in **h**, to better compare activity between left and right corrections. We note that this analysis has more noise because (1) we are measuring from single cells, and (2) we restricted the analysis to one session (to sample independent cells), which will have a limited number of deviations to analyse. Cells that are significantly more active for either left or right deviations are highlighted in red (p < 0.01, Wilcoxon rank-sum test across turning events, corrected using Benjamini-Hochberg method). **j**, Rate of course corrections for left and right deviations. Course corrections defined as in **g**. **k**, Rate of large course corrections (heading deviation > $\pi/3$, turning accel. > 1 rad/s$^2$ in the opposite direction), showing a strong bias toward large course corrections for right deviations, mirroring the slight bias in activity in **h**. Mean and bootstrapped 95% confidence intervals are shown.

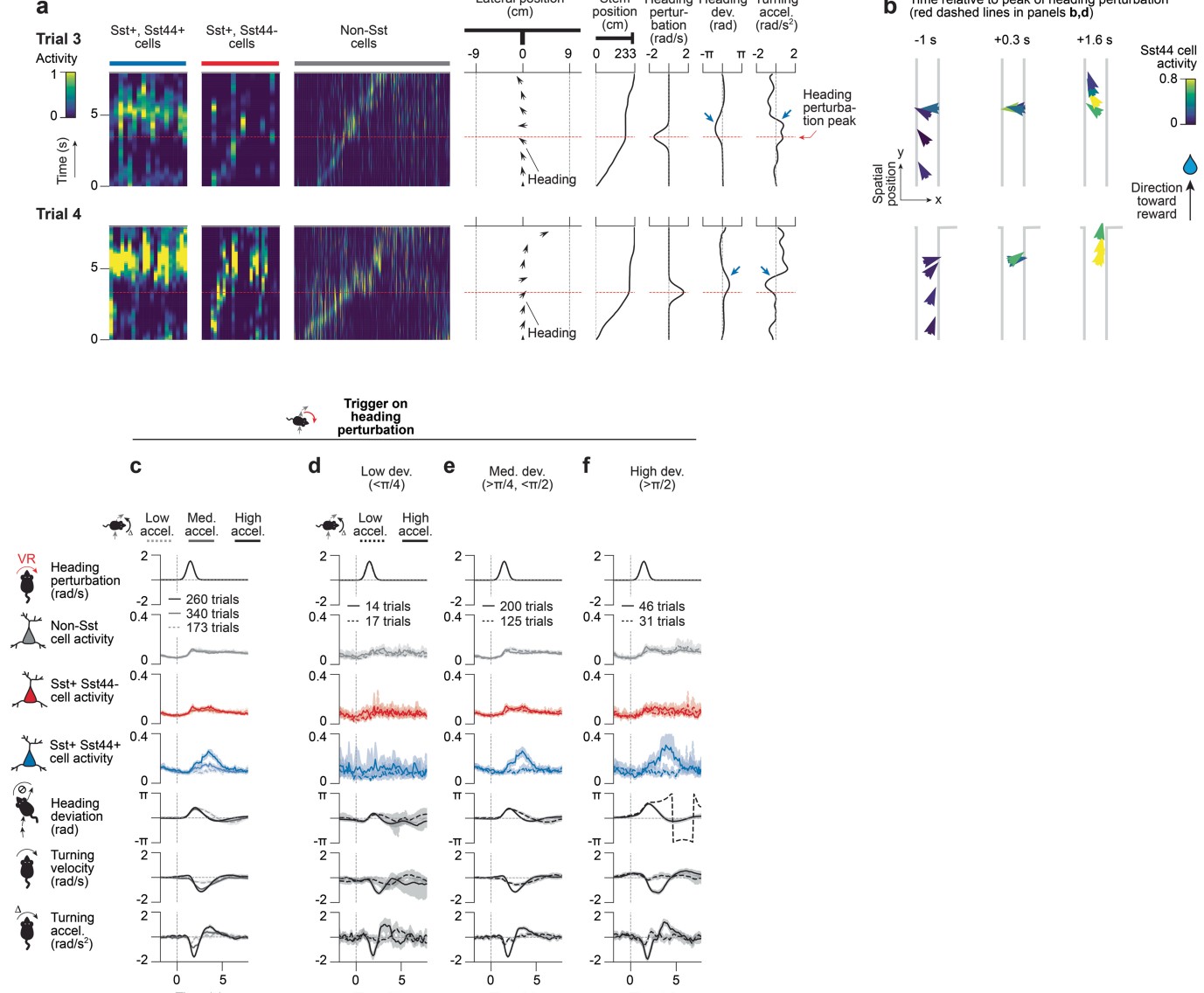

**Extended Data Fig. 8 | Examples and quantitative characterization of Sst44 cell activity during heading perturbations. a–b,** Additional examples of Sst44 cell activity during corrections for experimentally induced heading deviations. **a,** Activity (smoothed) and behaviour over time during heading perturbations. Same examples as in Supplementary Video 4. Each column in the activity heatmaps represents the activity of one cell over time. Cells sorted based on activity along the maze. Black arrows indicate the lateral position and heading of the mouse over time (pointing up is toward the end of the maze). Trial clipped to highlight heading perturbation event. **b,** Activity as a function of the mouse's trajectory relative to the heading perturbation (t = 0). X and y length scales are not proportional. **c–f,** Sst44 cell activity as a function of turning acceleration and heading deviation magnitude during heading perturbations. 8 mice, 16 sessions. **c,** Mean activity and behaviour during heading perturbations, split by whether the mouse corrected strongly (turning accel. > 1 rad/s² in the opposite direction, +2 s after the heading perturbation

was triggered – trigger indicated by the grey dashed line at 0 s), moderately (turning accel. > 0.5 rad/s², < 1 rad/s²) or weakly (< 0.5 rad/s²). Left and right heading perturbations were pooled after inverting behaviour for leftward perturbations. Wilcoxon rank-sum test across trials, Sst44 cell activity strong vs moderate accel.: p = 4e-7, moderate vs weak accel.: p = 1e-3. **d–f,** Mean activity and behaviour during heading perturbations, splitting by low (**d,** <π/4), moderate (**e,** >π/4, <π/2) and high (**f,** >π/2) heading deviations, as well as strong (>1 rad/s²) and weak (<0.5 rad/s²) turning accelerations in the opposite direction. Wilcoxon rank-sum test across trials, Sst44 cell activity high vs moderate or low heading deviation: p < 0.01, moderate vs low: p = 0.06. Activity averaged over 2.5 to 4.5 s relative to 0 to 1 s was used for statistical tests. Turning velocities and accelerations refer to ball movements, not virtual movements that also include the heading perturbation. Mean and bootstrapped 95% confidence intervals are shown.

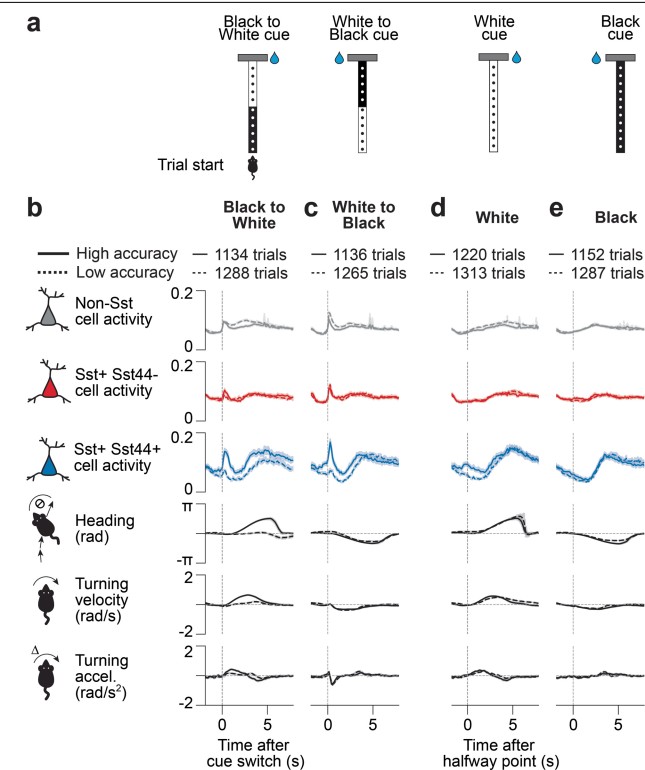

**Extended Data Fig. 9 | Sst44 cell activity increases after learning to adjust to a within-trial change in cue and reward location. a**, On 50% of trials, we changed the cue at the halfway point from black to white or white to black, along with the reward location, such that the mouse was rewarded based on the second cue. The change in cue was visible as the mouse approached it. **b–e**, Activity and behaviour as the mouse passed the halfway point during cue switch and control trials, split based on behavioural performance (low accuracy < 80% correct, high accuracy > 80% correct). Wilcoxon rank-sum test across trials, Sst44 cell activity solid vs dashed line, activity averaged over 0 to 1 s relative to −1 to 0 s: p < 1e-10 (**b–c**), p = 2e-4 (**d**), p = 0.997 (**e**). Wilcoxon signed-rank test across trials, activity on high accuracy trials relative to mean activity on low accuracy trials, averaged over 0 to 1 s, Sst44 vs other cell types: p < 1e-10 (**b–d**), Sst44 vs Sst: p = 2e-4 (**e**), Sst44 vs Non-Sst: p = 0.7 (**e**). Mean and bootstrapped 95% confidence intervals are shown. 7 mice, 45 sessions.

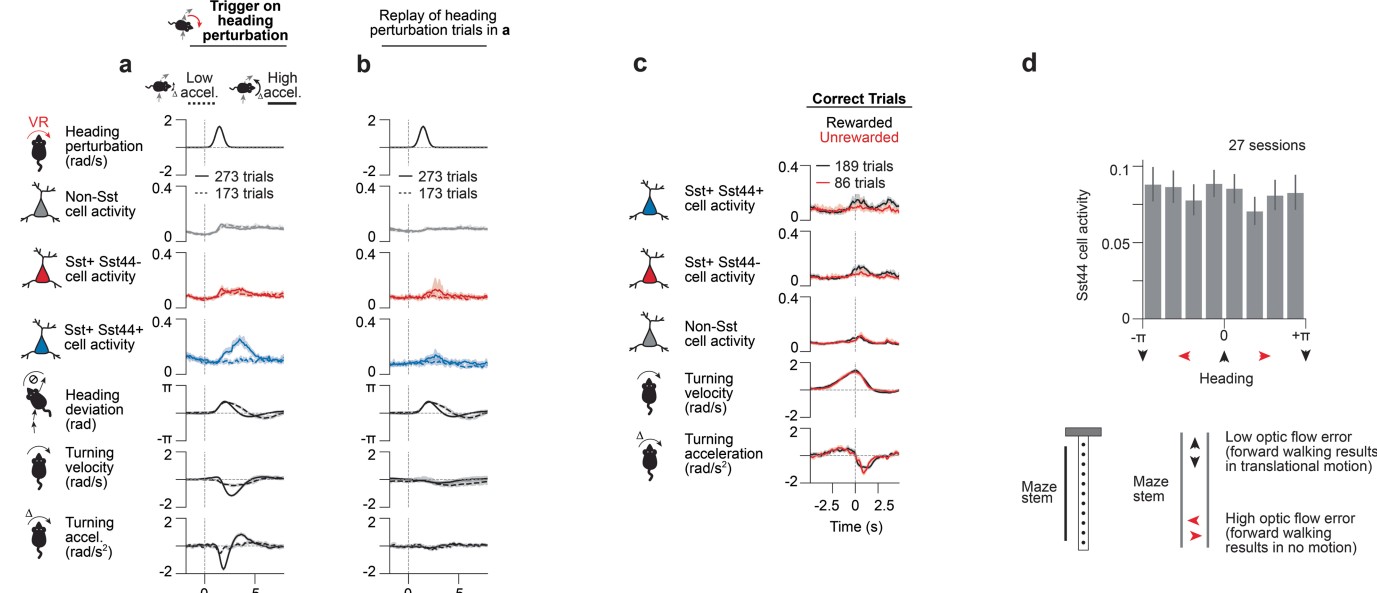

**Extended Data Fig. 10 | Sst44 cell activity during visual playback, reward omission and unexpected optic flow. a–b**, Visual playback of heading perturbations does not induce strong Sst44 cell activity in PPC. **a**, Fig. 6g reproduced for comparison. 8 mice, 16 sessions. **b**, Sst44 cell activity in response to visual playback of the same heading perturbation trials shown in **a**. 5 mice, 16 sessions. Mice in **b** were trained to run on a virtual linear track before being presented with the visual playback. Wilcoxon rank-sum test across trials, Sst44 cell activity solid line panel **a** vs **b**: p = 4e-13, solid vs dashed line panel **b**: p = 0.3. Activity averaged over 2.5 to 4.5 s relative to 0 to 1 s was used for statistical tests. Turning velocities and accelerations refer to ball movements, not virtual movements that also include the heading perturbation. **c**, Low contribution of reward expectation error to Sst44 cell activity. We omitted rewards on 20% of correct trials and added a reward on 20% of incorrect trials. Incorrect trials are not shown because there were too few trials after selecting for similar turning velocity. Activity and turning behaviour split by whether the mouse was

rewarded or not. We selected trials where turning velocity was similar to the unrewarded trial mean (cosine similarity > 0.8, using the same window as in the plot) in order to minimize contributions from behavioural differences. Heading deviation is not shown because the screen is dark when the trial ends and the reward is delivered. 4 mice, 6 sessions. **d**, Sst44 cell activity is inconsistent with a response to an error in expected visual flow. Sst44 cell activity as a function of heading within the maze stem. Within the stem, the walls are always oriented north-south, meaning that running north or south is equivalent to running parallel to the wall, and running east or west is equivalent to running directly into the wall. When running into the wall, visual flow is expected but not received, which should generate an error in the expected visual flow. Samples in each bin were matched for pitch ball velocity to control for expected visual flow. 7 mice, 27 sessions. Mean and bootstrapped 95% confidence intervals are shown.

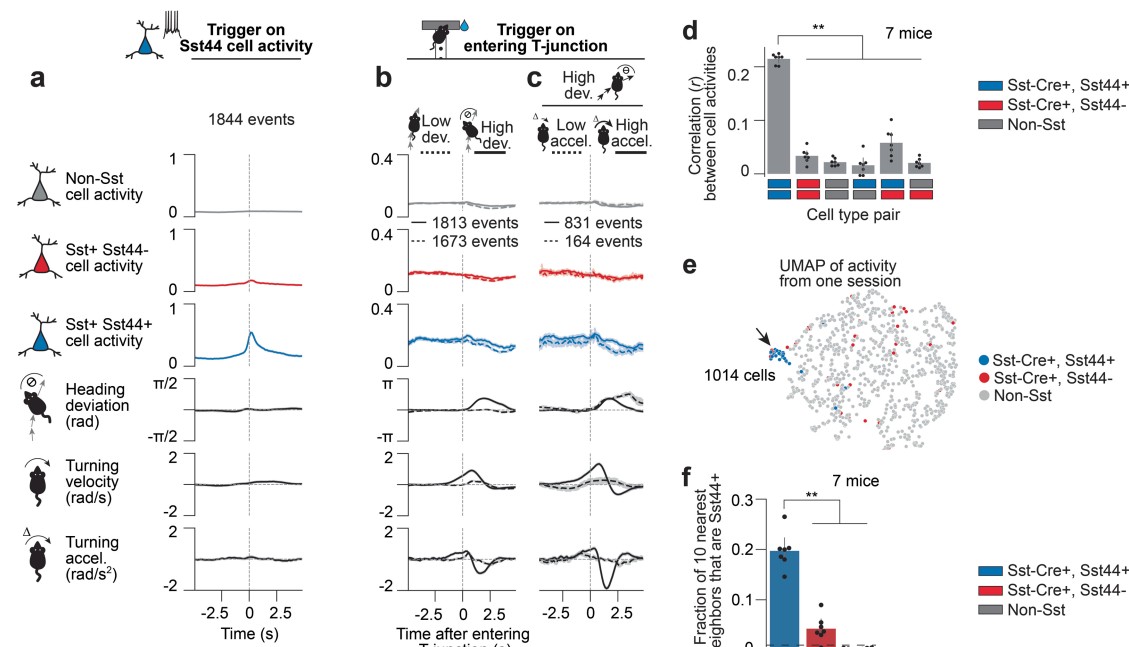

**Extended Data Fig. 11 | Sst44 cells in retrosplenial cortex do not activate during corrections for heading deviations. a**, Activity in RSC and behaviour averaged over large bursts of Sst44 cell activity (smoothed Sst44 cell activity > 0.4). **b**, Activity and behaviour as the mouse entered the T-junction, split based on whether this was followed by a large (> π/6) or a small (< π/12) heading deviation. Left and right deviations were pooled after inverting behaviour for left deviations. Selection criteria were evaluated at +1.5 s after entering the T-junction. **c**, Same as **b**, splitting trials with a high deviation based on whether the mouse corrected strongly (turning acceleration > 1 rad/s² in the opposite direction) or weakly (< 0.50 rad/s²). **d**, Pearson correlation between cells of each cell type. Kolmogorov-Smirnov test, Sst44+/Sst44+ pair vs other cell type pairs, p < 0.01. **e**, UMAP projection of each cell's activity from a sample session, showing clustering of Sst44 neurons. **f**, Fraction of 10 nearest neighbours in activity space that are Sst44+. Dashed line: mean after shuffling cell type identities. ** p < 0.01, Kolmogorov-Smirnov test. These data for RSC were collected on interleaved PPC sessions from the same mice as in Fig. 5. In **d**–**f**, activity was smoothed with a 0.25 s gaussian filter. 7 mice, 23 sessions. Mean and bootstrapped 95% confidence intervals are shown.

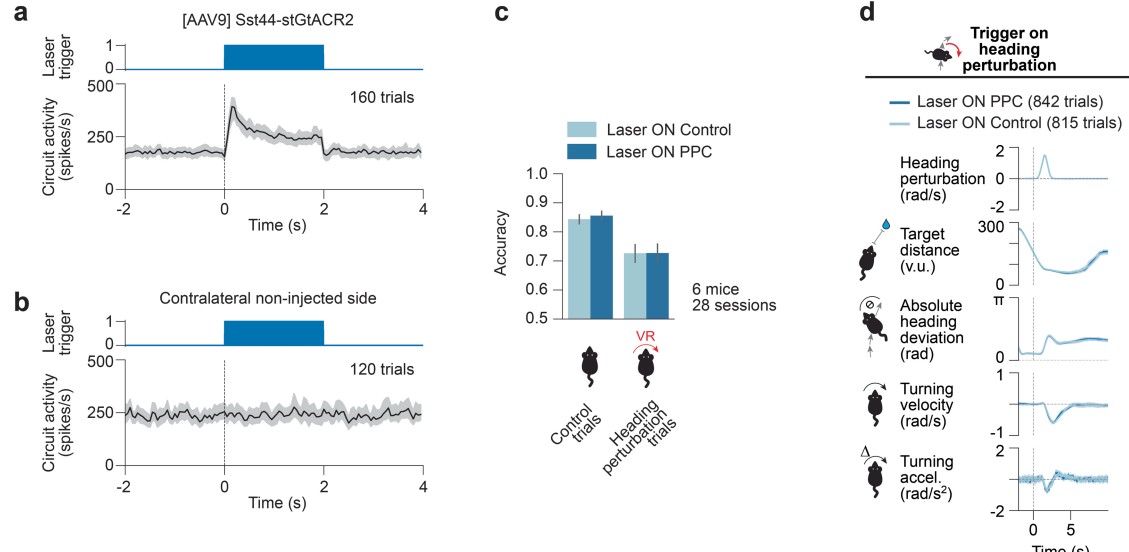

**Extended Data Fig. 12 | Optogenetic inhibition of Sst44 cells in posterior parietal cortex during navigation and heading perturbations.** This experiment aims to test the involvement of Sst44 cells in driving moment-to-moment corrections in well trained mice. We targeted optogenetic silencing to Sst44 cells and looked for changes in behaviour during course corrections. We tested the efficacy of our optogenetic inhibition approach in a separate set of experiments, not in the context of behaviour, by looking for an indirect effect of optogenetic inhibition of Sst44 cells on circuit activity. Note that we did not directly measure the suppression of activity in Sst44 cells nor did we measure their suppression during error correction events when they are expected to receive strong inputs, including through electrical connections from gap junction coupling. Therefore, we cannot exclude that Sst44 cells spiked during error corrections in our behavioural experiments. Although we did not observe a behavioural effect from targeting optogenetic inhibition to Sst44 cells, we caution that this may be due to technical reasons, rather than reflecting the lack of involvement of Sst44 cells in driving moment-to-moment course corrections. We thus encourage readers to interpret these experiments cautiously. See Methods for further details. **a**, We expressed stGtACR2 under the control of the Sst44 enhancer in PPC. Extracellular recording of circuit activity (pooling spikes across all probe units) in PPC during photostimulation (470 nm, 40 Hz, 50% duty cycle, 0.7 mW average power through a 200 μm diameter optic fibre). 8 probe insertions, 20 trials per probe insertion, 2 mice. **b**, Same as **a**, for the non-injected contralateral side. 6 probe insertions, 20 trials per probe insertion, 2 mice. **c**, T-maze accuracy during trials in which Sst44 cells in PPC were inhibited (Laser ON PPC) vs control trials in which the laser was directed to control sites (Laser ON Control). In heading perturbation trials, we added a heading perturbation as in Fig. 6. Control trials without a heading perturbation were interleaved. Light stimulation was as in **a**–**b**, except we illuminated a 1 mm diameter spot centred on PPC in each hemisphere to fully inhibit this area, and doubled the average power density to help compensate for the thickening of the dura under the window over time during training. **d**, Mouse's behaviour over time during heading perturbations. Left and right perturbations were pooled after inverting behaviour for left perturbations. Mean and bootstrapped 95% confidence intervals across trials are shown.

# Reporting Summary

## Statistics

For all statistical analyses, confirm that the following items are present in the figure legend, table legend, main text, or Methods section.

| n/a | Confirmed | |
|---|---|---|
| ☐ | ☒ | The exact sample size (*n*) for each experimental group/condition, given as a discrete number and unit of measurement |
| ☐ | ☒ | A statement on whether measurements were taken from distinct samples or whether the same sample was measured repeatedly |
| ☐ | ☒ | The statistical test(s) used AND whether they are one- or two-sided *Only common tests should be described solely by name; describe more complex techniques in the Methods section.* |
| ☒ | ☐ | A description of all covariates tested |
| ☐ | ☒ | A description of any assumptions or corrections, such as tests of normality and adjustment for multiple comparisons |
| ☐ | ☒ | A full description of the statistical parameters including central tendency (e.g. means) or other basic estimates (e.g. regression coefficient) AND variation (e.g. standard deviation) or associated estimates of uncertainty (e.g. confidence intervals) |
| ☐ | ☒ | For null hypothesis testing, the test statistic (e.g. *F*, *t*, *r*) with confidence intervals, effect sizes, degrees of freedom and *P* value noted *Give P values as exact values whenever suitable.* |
| ☒ | ☐ | For Bayesian analysis, information on the choice of priors and Markov chain Monte Carlo settings |
| ☒ | ☐ | For hierarchical and complex designs, identification of the appropriate level for tests and full reporting of outcomes |
| ☐ | ☒ | Estimates of effect sizes (e.g. Cohen's *d*, Pearson's *r*), indicating how they were calculated |

*Our web collection on statistics for biologists contains articles on many of the points above.*

## Software and code

Policy information about availability of computer code

| Data collection | ScanImage (Vidrio), AxoScope (Axon Instruments), MATLAB (Mathworks), Olympus Acquisition Software |
|---|---|
| Data analysis | Python, R, MATLAB (Mathworks), Fiji |

For manuscripts utilizing custom algorithms or software that are central to the research but not yet described in published literature, software must be made available to editors and reviewers. We strongly encourage code deposition in a community repository (e.g. GitHub). See the Nature Portfolio guidelines for submitting code & software for further information.

## Data

Policy information about availability of data

All manuscripts must include a data availability statement. This statement should provide the following information, where applicable:
- Accession codes, unique identifiers, or web links for publicly available datasets
- A description of any restrictions on data availability
- For clinical datasets or third party data, please ensure that the statement adheres to our policy

Genomic sequencing data will be deposited before publication. Other data are available upon request.

# Field-specific reporting

Please select the one below that is the best fit for your research. If you are not sure, read the appropriate sections before making your selection.

☒ Life sciences  ☐ Behavioural & social sciences  ☐ Ecological, evolutionary & environmental sciences

For a reference copy of the document with all sections, see nature.com/documents/nr-reporting-summary-flat.pdf

# Life sciences study design

All studies must disclose on these points even when the disclosure is negative.

| | |
|---|---|
| Sample size | Sample sizes were chosen based on previous similar experiments. |
| Data exclusions | Data inclusion criteria are indicated in the manuscript. |
| Replication | We observed similar results in preliminary experiments, and in multiple cohorts of mice presented in the manuscript. |
| Randomization | Comparisons were made within the same animals or samples. Trial types were randomly interleaved. |
| Blinding | Blinding not relevant because comparisons were made within animals or samples. |

# Reporting for specific materials, systems and methods

We require information from authors about some types of materials, experimental systems and methods used in many studies. Here, indicate whether each material, system or method listed is relevant to your study. If you are not sure if a list item applies to your research, read the appropriate section before selecting a response.

## Materials & experimental systems

| n/a | Involved in the study |
|---|---|
| ☒ | ☐ Antibodies |
| ☒ | ☐ Eukaryotic cell lines |
| ☒ | ☐ Palaeontology and archaeology |
| ☐ | ☒ Animals and other organisms |
| ☒ | ☐ Human research participants |
| ☒ | ☐ Clinical data |
| ☒ | ☐ Dual use research of concern |

## Methods

| n/a | Involved in the study |
|---|---|
| ☒ | ☐ ChIP-seq |
| ☐ | ☒ Flow cytometry |
| ☒ | ☐ MRI-based neuroimaging |

## Animals and other organisms

Policy information about studies involving animals; ARRIVE guidelines recommended for reporting animal research

| | |
|---|---|
| Laboratory animals | Mice were used in this study. The following strains were used: Gad2-Cre (Jax 010802), Sun1-GFP (Jax 021039), C57Bl/6J (Jax 000664), Sst-Cre (Jax 013044) and Ai14 (Jax 007914). Sex indicated in each experiment. Aged 8 weeks to 72 weeks. |
| Wild animals | The study did not involve wild animals. |
| Field-collected samples | The study did not involve samples collected from the field. |
| Ethics oversight | All experimental procedures were approved by the Harvard Medical School Institutional Animal Care and Use Committee. |

Note that full information on the approval of the study protocol must also be provided in the manuscript.

# Flow Cytometry

## Plots

Confirm that:

☐ The axis labels state the marker and fluorochrome used (e.g. CD4-FITC).

☐ The axis scales are clearly visible. Include numbers along axes only for bottom left plot of group (a 'group' is an analysis of identical markers).

☐ All plots are contour plots with outliers or pseudocolor plots.

☐ A numerical value for number of cells or percentage (with statistics) is provided.

## Methodology

| | |
|---|---|
| Sample preparation | *Describe the sample preparation, detailing the biological source of the cells and any tissue processing steps used.* |
| Instrument | *Identify the instrument used for data collection, specifying make and model number.* |
| Software | *Describe the software used to collect and analyze the flow cytometry data. For custom code that has been deposited into a community repository, provide accession details.* |
| Cell population abundance | *Describe the abundance of the relevant cell populations within post-sort fractions, providing details on the purity of the samples and how it was determined.* |
| Gating strategy | *Describe the gating strategy used for all relevant experiments, specifying the preliminary FSC/SSC gates of the starting cell population, indicating where boundaries between "positive" and "negative" staining cell populations are defined.* |

☐ Tick this box to confirm that a figure exemplifying the gating strategy is provided in the Supplementary Information.

