## [Peer Review File · Nature]

Manuscript Title: A cell type specific error correction signal in posterior parietal cortex

Reviewer Comments & Author Rebuttals

Reviewer Reports on the Initial Version:

Referees' comments:

Referee #1 (Remarks to the Author):

Using a highly exciting new method of labeling a defined subset of somatostatin neurons (Chodl+, Hpse+, Calb+; Sst44) by targeting enhancers inserted into an AAV, the authors perform 2P calcium imaging in PPC while animals navigate a visually cued T-maze in virtual reality. Activity of all neurons in PPC were recorded and SST+ and Sst44+ neurons were each labeled with different fluorophores. Sst44+ neurons activate more synchronously than other neuron types. Stimulation of Sst44+ neurons at the decision-point of the t-maze excited other Sst44+ neurons and inhibited other neurons. Stimulation of SST-Cre neurons did not have the same excitatory effect. There were no changes in behavior caused by stimulation. Sst44+ neurons were activated during course corrections on the maze, firing at high heading deviation from its typical course and high turning acceleration to correct its path. Introducing drift into the VR artificially and forcing the mouse perform course corrections also activates Sst44+ neurons. Sst44 neurons were not however activated during error trials, so they did not signal unrewarded trials.

Major comments:

This is a potentially very exciting and highly novel finding, showing self-excitation among a precisely defined subtype of interneurons and activation by course correction. The experiments are performed carefully and the analysis is elegant. The photostimulation experiments in particular go a long way for demonstrating self-excitation rather than excitation by common inputs. Further experiments to flesh out the story would greatly enhance the paper.

rrrr

1. The major weakness of the paper is there is no causal role for these neurons in the VR-based behavior or in any other navigation-based behavior. This makes it difficult to understand the role these neurons play. What is the impact of more large-scale excitation/inhibition of these neurons?
2. Does stimulation of Sst44+ neurons during baseline periods outside of T-maze behavior also result in activation of other Sst44+ neurons or does it require that the animal be performing the behavior or running?
3. What is the mechanism of the self-excitation of Sst44+ neurons? It would seem characterizing this mechanism would be essential for presenting a full story. There is speculation in the discussion but it would be important to elucidate this mechanism.
4. Is the downstream effect of activation Sst44+ neurons compared with Sst-Cre neurons? Do they target different populations?
5. Do Sst44+ neurons receive different inputs than Sst-Cre neurons? Where does the error signal come from?
6. What is the activity of the Sst44+ neurons during early stages of training when animals make frequent navigational errors?

rrrrrr

Referee #2 (Remarks to the Author):

This paper uses genomic enhancers, optogenetics and imaging to identify the functional role of a

cortical interneuron sub-type in behavior. First, the authors target SST+interneurons, using the SST44 enhancer and interestingly, show that increased SST44 accessibility correlates with increased genomic accessibility of Arc and Fos. Next, they show that the activity of individual SST44 neurons in the PPC is highly correlated with the activity of other SST33 neurons, suggesting that this neuronal cell type forms a specialized subnetwork. This idea is further supported by the targeted activation of individual SST neurons, in which photostimulation of small groups of SST44 neurons led to a net excitatory effect on other SST44 neurons, even when the non-targeted neurons were several hundred micrometers away from the closest photo-stimulated target. Perhaps interestingly, targeted activation of SST44 neurons did not influence the animal's behavior. Finally, the authors investigate the activity of SST44 neurons during a goal-directed navigation task and report that SST44 neurons are most active when mice correct for deviations in their heading direction (i.e. bring current heading back towards average heading at a particular location). This is also observed when mice correct experimenter induced heading deviations. The authors thus conclude that SST44 neurons signal 'error correction'. While much of their analysis supports this idea, there are several shortcomings to the paper.

1. Perhaps most critically, it appears as if the SST44 neurons are most strongly activated when mice vigorously correct for heading deviations (i.e. a large rotational acceleration). For example, Figure 5D and extended data Figure 9, appears to show that when mice correct their heading with lower acceleration the heading deviation is corrected almost as quickly as the high acceleration case, but SST44 neurons show extremely low levels of activity. Thus, even though the error is corrected on similar timescales, SST44 neurons are much less active – which would complicate the interpretation that provide a true general navigational 'error signal'.

2. The authors note that increased SST44 accessibility correlates with increased genomic accessibility of Arc and Fos, and speculate that SST44 could therefore label recently active Calb2+ and Hpsc+ cells. However, it is unclear if increased Arc/Fos accessibility correlates with recent activity. In addition, if this were indeed the case, this could be problematic as it would mean that expression driven by the SST44 enhancer would depend on the activity of neurons in the injected region around the time of injection.

3. It is interesting to see that activation of SST44 neurons in RSC is also synchronous but not driven by corrective movements. However, the absence of this signal in RSC isn't sufficient to say that the function is specific to PPC. Repeating the experiment in V1, for example, which plays a role in course corrections would strengthen this claim however.

4. It wasn't clear to me if all SST44 neurons are active independent of the deviation of direction, or if there is selectivity for right or left corrective movements in some cells. Other work, for example, has shown that excitatory neurons (e.g. in motor cortex) are preferentially active either during left or right turns in a similar task (Heindorf, Keller, Arber, Neuron 2018).

5. Related to point 4, in ED9, the authors show that average SST44 activity split by right vs left deviations. It is hard to say from the plotted lines, but it looks like corrections to right deviations lead to stronger activity.

6. There are several places where figure legends lack important details. For example: Fig 1A: Colors, Fig 4E: average of how many cells in each group?, Ext Fig 6: burst rate is not defined.

7. Details in the Methods appear incomplete.

- Classifying cell types with cellpose: what was the version of cellpose, segmentation model used?
- UMAP both for ATAC seq embedding and activity embedding should also include details such as version number, distance function used etc.
- Photostimulation: what was the stimulation frequency, duration of spiral, duty cycle?
- Influence mapping: Unclear if this was done during the task or in the dark?
- For triggering on SST44 activity, it is unclear if activity of all co-recorded SST44 neurons is

averaged, and the triggers calculated on that average.

8. For imaging experiments, authors note that they sample same field of view multiple times, except for the photostimulation experiments. In many of the analyses it is unclear if the data presented is averaged across cells, i.e. treating each cell from each session as independent, even though the authors state themselves that that is not the case. The authors should clarify and demonstrate that their conclusions hold even when considering that cells from multiple sessions of the same animal. For example by first averaging within animals, and then averaging between animals.

9. Along similar lines, in many instances, number of animals and total number of sessions is listed, but it is unclear how the sessions are distributed among the animals.

Referee #3 (Remarks to the Author):

Green and colleagues identify a subset of somatostatin positive neurons (Sst44+) in the mouse association cortex. In a series of elegant experiments, the authors develop a method to specifically target, observe, and perturb this population to show that Sst44+ neurons tend to fire synchronously (often near the end of the virtual track), inhibit neighboring neurons, and are most strongly activated when mice exhibit course corrections on a cued navigation task. As with other work from the Harvey laboratory, the experiments are carefully conducted with numerous controls and the manuscript is well written. Accordingly, I have very few technical comments overall. That said, I think there are more general issues with interpretation that may require rethinking base claims or adding additional experiments in order for this work to merit publication (especially in such an impactful journal).

Main comments.

The authors make a strong case for the impact of this work purely based on their cell-type-specific targeting of a sub-class of Sst neurons (which I mostly agree with). While they conduct photostimulation experiments to map the functional circuits centered on these neurons there is little other rationalization for why, at an anatomical, physiological, or circuit level, this population should play a significant role in signaling and/or mediating error corrections in navigation. While the experiments specifically examining the Sst44+ population are certainly impressive, there is a bit of a 'black box' feel to the main findings concerning these cells. It would be useful to know more about how these neurons project within PPC (or beyond). This anatomy is especially important considering the lack of a similar role of Sst44+ neurons in RSC. Moreover, I think the authors should more concretely consider how this small subset of cells might engage other neurons in the area. A model could provide insight into how the Sst44+ population can alter processing dynamics given error.

The key finding is that Sst44+ neurons are synchronously activated when mice exhibit course correction behaviors to reorient during virtual navigation. Critically, the authors show by omitting the reward that Sst44+ cells are not signaling error generally but instead navigation related errors. To me, a course correction felt kind of arbitrarily defined and potentially highly dependent upon the experimental configuration. Many of the example course corrections possess a very specific movement profile (high angular acceleration/velocity) that could relate to controlling the inertia of the floating ball (suggesting that the current interpretation could be dependent upon the experimental configuration). Because of this, course correction is strongly conflated with a specific action state related to reorienting in virtual space. Given that PPC neurons are sensitive to changes to velocity and acceleration (Whitlock et al., 2012), it would be interesting to see these Sst44+ fire for an error that is decoupled from a reorienting response (e.g. if the animal were to overshoot or undershoot a rewarded area on a linear track, start the trial early or late, or if the cues were swapped mid trial or their contingencies reversed). Similarly, controls showing that Sst44+ cells

are similarly active when the animal course corrects but the movement of the floating platform is altered are critical (or its relationship to visual feedback as in the comment below).

Analyses excluding a role of virtual angular velocity (5h) and visual flow information on activation of Sst44+ neurons were not convincing in my opinion. The authors should look at activation of this population during a passive version of the task, a version with no optic flow (e.g. all black walls), and a version in which movement and visual feedback are decoupled.

Minor comments.

The co-activity based UMAP projections and quantification in Figure 2i-l are not especially compelling in my opinion as there are multiple other Non-Sst clusters of similar quality as well as Non-Sst cells embedded within the Sst44+ cluster in 2i. It is unclear what these plots add beyond 2g and 2h.

3f - What is the failure rate of target stimulation? It appears that many stims do not yield a calcium transient. Do failed stims have any temporal relationship to stimulation of other Sst44+ cells within the sweep?

In figure 4e-i, Sst44- and Sst44+ cells appear to have correlated activity patterns while the influence mapping experiments in Figure 3 indicate that they should be anticorrelated. This is a bit confusing and I'm curious if the authors have any thoughts about this? Is it possible that synchrony among these Sst populations is state dependent?

Is there more Sst44+ activity on delay maze trials wherein there are presumably more errors? What about in sessions prior to reaching the 85% criteria?

Referee #4 (Remarks to the Author):

In this study, Green et al. report that a subset of SST+ cells in the PPC (identified via the enhancer Sst44+) tend to fire synchronously with other Sst44+ cells, and excite each other. In a navigation task, they report a cell-type specific error correction signal, one in which that these Sst44+ neurons in PPC (but not RSC), but not other SST or non-SST cells are activated during error correction. There are additional experiments and analyses that would be required to support the hypothesis that Sst44+ cells are truly a unique population with specialized coding properties during navigation for reward.

Detailed comments:

1) Fig 1: The authors should report the identity of the fraction (~10%) of Sst44+ cells that are not SST+, as this may have implications for their subsequent results. Are they another subclass of inhibitory neurons? In addition, what fraction of the total SST neurons counted in L2/3 are Sst44+?

2) Fig 2d: Authors should validate in the PPC that the TdTomato signals from the Sst-Cre line are indeed only labeling SST+ neurons to ensure no leak in the viral expression. Similarly to the point above, what kind of neurons are Sst44+ cells that are not SST-Cre+?

3) Fig 3f: . The manuscript would benefit from a clearer example of traces to validate the stimulation, one in which the reader can see the time window of stimulation and the subsequent Ca transient in optically targeted and non-targeted neurons. In its current form it is very difficult to tell the differential effects of target vs. control optical simulations

4) Fig 3h-i: it is difficult to reconcile the relationship between Sst44+ and Sst44-, as well as Sst44+ and non-SST+ cells shown here, vs. the correlations shown in Fig 2 and Extended Fig 10. Specifically, here it seems that Sst44+ stimulation leads to inhibition of non-SST and Sst44- cells, but the correlations between Sst44+ and Sst44- populations are positive in Fig 2 (and Extended Fig 10).

5) A central component of this study is the synchronization/net excitatory effect Sst44+ cells have

with one another. This is a highly impactful finding that should be explored in more depth. In order to achieve a better understanding of this, the authors should attempt to describe this connection more rigorously with another method such as slice electrophysiology.

6) The authors state "Remarkably, the synchronous, self-reinforcing, and course correction activity was specific to Sst44 cells and not present in other neighboring Sst cells." However the data, as presented in figures 4 and 5 don't entirely support this claim. From the data in Figs 4F-I and 5G it seems that the Sst44- and non-SST cells also correlate with heading deviation and turning acceleration, albeit the effects are larger in Sst44+ population. This can also be seen in the heat maps where a significant portion of activity in the heat maps of Sst44- cells occurs in the same location of the maze as the synchronous activation of the Sst44+ cells This raises the question of how specialized are the Sst44+ cells in this response to course correction? More detailed analysis of the single neuron and population response properties are warranted, particularly comparing the highly correction selective SST+ Sst44- cells to the 44+ population as well as the response properties of Sst44- neurons in this task.

7) Related to the above point, in Fig 4e, the example trace from Sst44+ neurons is not entirely in support of the author's central point, as there are some small increases in Sst44+ neuron activity that do not correspond to any turn correction, and some turn corrections that do not recruit Sst4+ cells.

8) Fig 4g/i: A statistical comparison between Sst44+ and Sst44- activity across these conditions is required.

9) Fig 4g/i: what does activity look like in low heading deviation at T junction, but split up into low vs. high acceleration? ie/ same as 4h, but looking at low heading deviation trials.

While unilateral stimulation did not cause any behavioral changes, the impact of this study would be increased with bilateral optical modulation of these cells to link them directly with behavior.

10) Are there task periods where non-SST cells fire while Sst44+ do not? This data could provide a more wholistic understanding of the circuit in this behavior.

11) It would help if there was a discussion of the potential roles of Sst44+ in RSC, and why they do not seem to be activated in error correction.

12) Also in the discussion, a couple lines on basic circuit mechanism on what happens during course correction trials, and the role of Sst44+ neurons in PPC as it relates to the larger neural circuit that drives behavior?

Author Rebuttals to Initial Comments:

Response to all reviewers

We thank the reviewers for providing constructive feedback and helpful suggestions on how to improve the manuscript. We are happy that the reviewers found the work to be of interest. We have worked hard to incorporate the reviewers' suggestions by performing an extensive set of new experiments, analyses, and text revisions. We feel our manuscript is greatly improved as a result of the feedback and suggestions, and hope that our paper will now be considered appropriate for publication in *Nature*.

In brief, we have:

1. Performed new patch clamp electrophysiology on pairs of Sst44 cells. These experiments were challenging because they involved simultaneously patching two identified Sst44 cells, which are a small fraction of all neurons. We find that Sst44 cells are electrically coupled by gap junctions, validating the excitatory interactions we observed with optogenetics *in vivo*. Further, these new results provide a mechanism for the excitatory interactions among an inhibitory neuron population, as well as for the synchronous activity of Sst44 cells during task performance. These data are presented in a new main figure.
2. Performed new experiments in which we imaged Sst44 cell activity during a version of the T-maze task in which the visual cue is switched partway through the maze, indicating a change in reward location. Consistent with an error correction signal, we observed Sst44 cell activity when this cue switch occurred, as mice had to change their trajectory toward a new reward site. This Sst44 cell activity around the cue switch only occurred after mice had learned the cue-switch task, as expected for an error correction signal. This experiment provides a second experimental test of the error correction hypothesis.
3. Performed new experiments in which we imaged Sst44 cell activity during training, before the mice achieved high performance on the task. An error correction implies that the mouse knows the location of the reward, and therefore we expect to only observe an error correction signal once the mouse has learned the location of the reward. Consistent with this hypothesis, and consistent with the cue switch experiments above, we observed little Sst44 cell activity before mice had learned the location of the reward site, and increasing activity once the mice began to learn the location of the reward.
4. Performed new experiments in which we imaged the activity of Sst44 neurons during the open-loop playback of traversals of the virtual maze, including playback of heading perturbation trials. These experiments allowed us to compare Sst44 cell activity in response to the same visual stimulus, with and without a corrective action. During playback, we find little Sst44 cell activity for the same events that led to Sst44 cell activity when the mice were in closed loop, ruling out that Sst44 cells respond mostly to visual flow.
5. Performed new control analyses to firmly rule out the possibility that Sst44 cells in PPC respond generally to the act of turning or to visual flow. Specifically, we identified events with similar turning velocity and acceleration and divided them into those events with and without a heading deviation. We show that, without a heading deviation, Sst44 cells do not activate strongly, even during vigorous turning. These new analyses show that Sst44 cells do not activate generally during all turning events, even when these events are matched in magnitude. These experiments also show that Sst44 cells do not respond generally to visual flow, since the angular velocity of the virtual environment is also matched in these analyses. Instead, Sst44 cells activate specifically during course corrections. This conclusion is also corroborated by our new experiments involving early training sessions, the cue-switch task, and open-loop playback.
6. Performed new experiments to fill individual Sst44 cells and traced their arbors. We find that Sst44 cells have arbors biased in the direction of layer 1, consistent with the morphology of Martinotti cells. These morphological data provide a deeper characterization of the Sst44 cell type.
7. Performed optogenetic inhibition of Sst44 cells during the task to measure behavioral effects. These experiments involved developing a new virus, validating an optogenetic inhibition scheme, and new behavioral experiments. Unfortunately, these experiments presented technical challenges for interpretation and did not yield a clear biological result. A larger study will be needed to fully characterize the causal role of these cells. We therefore did not include these experiments in the paper, but the results are presented below.

8. Performed RNAScope for Arc expression after task performance to test the possibility of activity inducibility for the Sst44 enhancer. We did not observe substantial Arc induction specifically in Sst44 cells in well trained mice, suggesting that expression from the Sst44 enhancer is not likely to be activity dependent. We have revised our conclusions regarding this point.
9. Performed RNAScope to confirm that the Sst-Cre line is specific for Sst cells in PPC.
10. Analyzed Sst44 cell activity with respect to the magnitude of the mouse's heading deviation and its corrective action. We find that Sst44 cells respond in a graded manner, especially to turning, helping to explain the variability in Sst44 cell activity during individual course corrections.
11. Analyzed the selectivity of individual Sst44 cells to corrections for left or right heading deviations. We find that most Sst44 cells respond to both, although some Sst44 cells respond more to one side versus the other.

We address each specific comment separately below.

Referee #1

Using a highly exciting new method of labeling a defined subset of somatostatin neurons (Chodl+, Hpse+, Calb+; Sst44) by targeting enhancers inserted into an AAV, the authors perform 2P calcium imaging in PPC while animals navigate a visually cued T-maze in virtual reality. Activity of all neurons in PPC were recorded and SST+ and Sst44+ neurons were each labeled with different fluorophores. Sst44+ neurons activate more synchronously than other neuron types. Stimulation of Sst44+ neurons at the decision-point of the t-maze excited other Sst44+ neurons and inhibited other neurons. Stimulation of SST-Cre neurons did not have the same excitatory effect. There were no changes in behavior caused by stimulation. Sst44+ neurons were activated during course corrections on the maze, firing at high heading deviation from its typical course and high turning acceleration to correct its path. Introducing drift into the VR artificially and forcing the mouse perform course corrections also activates Sst44+ neurons. Sst44 neurons were not however activated during error trials, so they did not signal unrewarded trials.

Major comments:

This is a potentially very exciting and highly novel finding, showing self-excitation among a precisely defined subtype of interneurons and activation by course correction. The experiments are performed carefully and the analysis is elegant. The photostimulation experiments in particular go a long way for demonstrating self-excitation rather than excitation by common inputs. Further experiments to flesh out the story would greatly enhance the paper.

1. The major weakness of the paper is there is no causal role for these neurons in the VR-based behavior or in any other navigation-based behavior. This makes it difficult to understand the role these neurons play. What is the impact of more large-scale excitation/inhibition of these neurons?

Thank you for this suggestion. We propose two hypotheses for the function of the error correction signal carried by Sst44 cells in PPC, which could be tested with this experiment. These hypotheses are presented in the Discussion section of the paper. The first hypothesis is that this signal acts as a “command” signal that contributes to executing corrections on a moment-by-moment basis. The second hypothesis is that this signal acts as a “teaching” signal that provides feedback to other cells in PPC and helps the circuit to learn the correct actions over time. This “teaching” signal would be important during the process of task learning. These possibilities are not mutually exclusive, and they each require a dedicated set of time-intensive experiments to test. Given the challenges to testing both hypotheses in a single experiment, we focused on experiments designed specifically to test the first hypothesis – that the error correction signal in Sst44 cells acts as a “command” signal that drives course corrections on a moment-by-moment basis.

We first considered performing strong and large-scale optogenetic activation of Sst44 neurons to test for a behavioral effect. When reasoning through the possible outcomes, however, we concluded that this type of experiment may be challenging to interpret. Large-scale activation of an inhibitory neuron type, especially

in a non-physiological manner as can be created with optogenetic stimulation, may lead to an overall shutdown of the neighboring excitatory neurons. In contrast, in natural cases, the effects of Sst44 cell activation may have a more tuned and specific effect on the circuit. We concluded that, if we only measured behavioral performance, it would be difficult to disambiguate these two possibilities, leading to an unclear interpretation. These experiments would require simultaneous calcium imaging, photostimulation, and behavior, along with careful calibration of the optogenetic stimulation, to ensure that our photostimulation was not simply shutting down the surrounding network. We were uncertain if it would be possible to achieve an appropriate stimulation regime with large-scale photostimulation approaches.

Instead, we considered that optogenetic inhibition experiments may be easier to interpret and thus attempted to perform optogenetic inhibition of Sst44 cells in PPC. These experiments presented multiple technical challenges and required the development of new reagents and the validation of these reagents.

We made an AAV containing the Sst44 enhancer driving soma-targeted GtACR2 (Sst44-stGtACR2), a blue light-activated inhibitory opsin. We first tested its efficacy using extracellular electrophysiology in awake head-restrained mice. Unfortunately, we did not have a way to test the direct effect of stGtACR2 activation on Sst44 cell spiking because we could not identify Sst44 cells with *in vivo* electrophysiology (with an excitatory opsin, one can identify time-locked, short latency responding cells, but with an inhibitory opsin, one needs a high baseline firing rate to identify cells that are inhibited with a short latency). We therefore instead measured the effect of stGtACR2 activation in Sst44 cells on spiking in the entire population of neurons recorded on a multi-channel electrode. Since Sst cells are inhibitory, we expect that inhibiting these cells will disinhibit the circuit. As expected, we observed an increase in the population activity when we delivered blue light. We observed no effect on the contralateral, uninjected side. This experiment indicates that the stGtACR2 activation did in fact work to some degree. However, because we were unable to measure Sst44 cell spiking directly, it remains unclear if these cells were completely silenced or just experienced reduced spiking. Also, because Sst44 cells are electrically coupled by gap junctions, as revealed in our new experiments, it is possible that the self-excitation through gap junctions may be challenging to suppress entirely with an inhibitory opsin, especially during large bursts of activity. Therefore, despite extensive efforts to develop and validate this approach, we remain uncertain of the extent of inhibition of Sst44 cell activity by stGtACR2. Notably, we considered chemogenetic approaches, like DREADDs, but these approaches lack the temporal resolution to test the specific hypothesis of Sst44 cells being involved in navigation course corrections. Specifically, if we observed an effect on behavioral performance with DREADDs, it would be difficult to determine if this effect was due to a deficit in course corrections or some other aspect of neural function required for the behavioral task. For this reason, we thought interpretation with chemogenetic approaches may be challenging and thus did not pursue this direction. Instead, we felt that interpretation would be easier with optogenetics that allowed us to inhibit Sst44 cells precisely at the time that course corrections were needed and not at other times when different neural processes were involved.

Despite these limitations with validating the optogenetic effect on Sst44 cell spiking, we nevertheless performed experiments in which we aimed to optogenetically inhibit Sst44 cells in PPC during behavior. Specifically, we injected the Sst44-stGtACR2 AAV bilaterally in PPC. We replaced the skull above the injection sites with two glass windows (one on each hemisphere) to allow light to enter the brain. After training mice to perform the T-maze task as described in Figure 2b, we performed bilateral inhibition experiments during the heading pulse perturbation described in Figure 6. We focused on the heading perturbation times because the mouse must course correct during these times, and because we expect Sst44 activity to be strong during these error corrections (Figure 6). We used a 470 nm laser to deliver blue light and directed the laser beam (1 mm in diameter at the brain surface) using a galvanometric mirror pair to either PPC (1.7 mm ML, -2 mm AP from bregma), or control sites on the dental cement, alternating each side at 40 Hz. We started photoinhibition when the heading pulse was triggered and terminated photoinhibition after 5 s or once the mouse reached the inter-trial interval.

This experiment aims to test if Sst44 cell activity is required for the mouse to execute a course correction. The outcome of these experiments is that we did not observe a difference in the mouse's behavior on trials

where we targeted the blue laser light to PPC compared to interleaved trials where we targeted the laser to control sites (Referee 1 Figure 1), both in terms of task accuracy (Referee 1 Figure 1a) and in terms of the mouse's running trajectory over time during the trial (Referee 1 Figure 1b).

It is possible to interpret this negative result in several ways, some of which result to technical challenges and other to biological interpretations.

A first possible interpretation is that we did not inhibit the Sst44 cell spiking sufficiently. In our electrophysiology validation experiments, we only observed circuit disinhibition with the strongest light intensities we could deliver with our optrode setup (38 mW/mm²), indicating that the efficacy of the Sst44-stGtACR2 virus is weak. In general, the Sst44 enhancer is weaker than strong promoters such as CAG or hSyn, which are more commonly used in optogenetics. Related to this possibility, and as mentioned above, we did not measure Sst44 cell spiking directly. It is therefore possible that we did not sufficiently reduce Sst44 cell spiking. In particular, it may be challenging to optogenetically counteract a large, gap junction-mediated self-excitation in the Sst44 cell population (see new experiments in Figure 4). Thus, even with our best efforts and even with stGtACR2 activation, Sst44 cells may have still spiked.

A second possibility is that Sst44 cells in PPC contribute to the execution of the course correction, but that other cells in other cortical or subcortical regions can also drive this behavior. That is, there may be redundancy across brain areas and thus silencing only the PPC's Sst44 cells might not have silenced the entire relevant population. Relatedly, it is possible that the silencing of Sst44 cells was heterogeneous in PPC itself. For example, not all Sst44 cells were necessarily transduced with the virus. Also, Sst44 cells in deeper layers may not have been as strongly inhibited given that we expect there to be less light delivered to deeper layers. If not all Sst44 cells were adequately silenced, the remaining active population may be sufficient to carry out the behavior.

A third possibility presents an interesting biological interpretation and is currently the interpretation we favor. If we interpret the optogenetics results as true, putting aside technical caveats and concerns, then we could conclude that Sst44 cells are not required to perform the error corrections after they have been learned. Instead, these cells may serve a different function. A leading candidate for an error-related signal is a function involved as a teaching signal that contributes to the learning of the corrective actions over time. Thus, Sst44 cells may not be involved directly in the execution of course corrections but rather may be critical as a teaching signal to guide learning of appropriate navigational trajectories to goal locations. This latter possibility was not tested in our optogenetics experiments as we focused only on well-trained mice. Testing this possibility would involve a different set of experiments.

Performing more experiments to explore the role of Sst44 cells in behavior is of high interest to us. However, after making this major effort on this first set of experiments, we realize that a great deal of more work is needed. First, significantly more work is needed to calibrate the conditions (if any) under which we can be confident that we are strongly inhibiting Sst44 cells, while preventing over-excitation of the circuit over long periods of disinhibition. Second, to test the hypothesis about a role in learning instead of the execution of course corrections, we would need to develop a suitable learning paradigm. Learning experiments present challenges because of significant variability from mouse to mouse. In particular, there is large variability in the speed with which mice learn our virtual T-maze task. Also, the silencing would need to be performed in one cohort and compared to a separate cohort, which, combined with the large variability between mice, would be a major undertaking given the time-consuming and complex nature of these behavioral experiments. The experiments mentioned here have already taken months of work from a postdoc and a senior research assistant. Extending these experiments as outlined here would take an even more substantial effort.

Given the difficulty of interpreting these results, we opt to not include these experiments in the paper but have discussed some of these possibilities in the discussion of our revised manuscript.

Referee 1 Figure 1. Behavioral effect of Sst44 cell inhibition in PPC.

We inhibited Sst44 cells as we perturbed the mouse's trajectory to determine if these cells are required for the mouse to perform error corrections. **a**, Accuracy in entering the rewarded arm during trials in which Sst44 cells in PPC were inhibited (Laser ON PPC) vs control trials in which the laser was directed to control sites (Laser ON Control). **b**, Mouse's behavior over time during heading perturbations. Left and right perturbations were pooled after inverting behavior for left perturbations. Line and bar plots show the mean and bootstrapped 95% confidence intervals across trials.

2. Does stimulation of Sst44+ neurons during baseline periods outside of T-maze behavior also result in activation of other Sst44+ neurons or does it require that the animal be performing the behavior or running?

Thank you for bringing up this interesting question. In new experiments, we performed patch clamp electrophysiology experiments on pairs of Sst44 neurons in slices made from mice that were not trained on the task (Figure 4). Consistent with our *in vivo* experiments during behavior, we observed that Sst44 cell neighbors are positively coupled through gap junctions (Figure 4, see our response to question 3 below). These experiments demonstrate that Sst44 cells excite each other, not only outside of the task, but also in an untrained animal.

3. What is the mechanism of the self-excitation of Sst44+ neurons? It would seem characterizing this mechanism would be essential for presenting a full story. There is speculation in the discussion but it would be important to elucidate this mechanism.

We agree that understanding the mechanism of the self-excitation is an exciting direction. Because the reviewer considered this direction essential for fleshing out the story, we have made a major effort to perform a set of new and difficult experiments. In these experiments, we performed patch clamp electrophysiology in brain slices from pairs of Sst44 cells. These experiments are challenging because only a small fraction of neurons is Sst44 cells, and we have to patch a pair of these rare cells simultaneously. We aimed to test whether these cells are electrically coupled by gap junctions and strikingly found that about 50% of pairs of Sst44 cells are connected through gap junctions (Figure 4). First, we show that a hyperpolarizing current in one cell hyperpolarizes the second cell in about half of pairs (Figure 4b-e). This protocol is standard in the literature to measure gap junction coupling¹. Second, we show that this connection is rapid, with a latency of approximately 1 ms (Figure 4c-d). Third, we show that this connection is reciprocal, where a hyperpolarizing current injection in cell 1 causes a hyperpolarization of cell 2, and a hyperpolarizing current injection in cell 2 likewise causes a hyperpolarization of cell 1 (Figure 4f). Fourth, we show that this connection is not affected by blockers of excitatory and inhibitory synaptic transmission

(AMPA, NMDA, and GABA receptor blockers), ruling out any possible polysynaptic connections that anyway would likely be longer latency (Figure 4g-h). These experiments therefore provide strong evidence that Sst44 cells are connected by gap junctions. This electrical coupling is consistent with and provides a mechanism for the self-excitation we observed in our *in vivo* optogenetic experiments during behavior. We feel this is an important finding and reveals a surprising and significant mechanism for self-excitation amongst an inhibitory neuron population. We thank the reviewer for encouraging us to pursue this direction.

4. Is the downstream effect of activation Sst44+ neurons compared with Sst-Cre neurons? Do they target different populations?

We agree it is an important direction to understand how Sst44 cells and other Sst cells impact the circuit in different ways. Our photostimulation experiments focused on stimulating Sst44 cells, and these experiments were challenging because they required combining imaging, photostimulation, and high accuracy behavioral performance. Although separately these components are less challenging, combining them simultaneously with a behavior task that is difficult to learn required a year of work from a postdoc and senior research assistant. In addition, it is unclear which downstream cell types would be differentially affected, which would require us to test multiple different options. While we agree this is an interesting direction, we feel that the scope of this experiment is better suited for a separate study to investigate the connectivity of these cells with the rest of the circuit in more depth.

Nevertheless, we were inspired by the reviewer's comment to identify where Sst44 neurons project within PPC, to better understand their downstream targets. We filled individual Sst44 neurons with a patch pipette (Figure 1j). These data showed that Sst44 neurons project to layer 1, where Sst neurons are known to modulate spine density on the apical dendrites of excitatory neurons². These data therefore show that the downstream targets of Sst44 neurons include layer 1. We also highlight this point in the Discussion (line 409).

5. Do Sst44+ neurons receive different inputs than Sst-Cre neurons? Where does the error signal come from?

We agree that understanding how the error correction signal in Sst44 neurons is computed based on the inputs to these cells is a major question. We propose one possibility based on our data – that the excitatory interactions among Sst44 cells serve a key role in computing this signal. For example, if Sst44 neurons receive inputs from excitatory neurons that each respond in a different way during error corrections, the gap junctions among Sst44 cells may serve to average these inputs, and, in combination with the non-linearity in spike generation, could create an activity signal that is specific to error corrections, even though individually, each input excitatory neuron is not specific to error corrections. Because the excitatory interactions are specific among Sst44 neurons, this mechanism could explain why this signal is specific to Sst44 neurons because, as a population, these cells are able to amplify this signal, whereas the other Sst neurons cannot. We emphasize that this is a hypothesis that remains to be experimentally tested. We highlight this possible mechanism in the Discussion (line 456).

To specifically identify the inputs to these cells, we would have to study inputs from multiple candidate cell types and inputs from both within PPC and from other brain regions. Investigating each of these inputs would require characterization of their activity and potentially manipulations to each of these inputs. While this is an exciting direction, it is broad in scope and exploratory. We therefore feel it would be better suited for a dedicated study to investigate this topic in depth.

6. What is the activity of the Sst44+ neurons during early stages of training when animals make frequent navigational errors?

Thank you for this excellent question. In new experiments, we imaged activity during training, before mice reached high performance (Extended Data Figure 11). An error correction signal implies that the mouse knows where the reward is located, which only occurs once the mice have learned the task. We therefore expect to observe high Sst44 cell activity only once the mice have learned the task, and low Sst44 cell activity during training, before the mice have learned the location of the reward, as measured by their accuracy in turning into the correct T-maze arm. Indeed, we found little Sst44 cell activity during early stages of learning when the mice performed with low accuracy. Specifically, in early training sessions, during times in which mice made turns that would put them on the path toward a reward (i.e., events with behavioral actions similar to error corrections), we did not observe synchronous Sst44 cell activity events. On the other hand, Sst44 cell activity was intermediate on sessions with intermediate behavioral performance. This experiment therefore indicates that Sst44 cells are active during corrective turns only once the mice have learned the location of the reward, and is therefore consistent with Sst44 cells carrying an error correction signal. In addition, the lower Sst44 cell activity early in training indicates that Sst44 cells do not respond generally during the act of turning or to visual flow and instead are specific to error corrections.

In a second set of new experiments, we imaged activity as mice learned to navigate a version of the T-maze in which the visual cue is switched partway through the maze (Extended Data Figure 17). For example, the cue might be switched from a white cue to a black cue. As a result, the location of the reward also changes, such as from the right T-arm to the left T-arm. In this task, mice have to update their trajectories partway through the maze when the cue and reward locations are changed. We observed an increase in Sst44 cell activity at the time of the cue switch, providing additional evidence that they activate during course corrections. Notably, this increase in Sst44 cell activity at the cue switch was only present after mice had learned this cue switch task. This result is consistent with our imaging experiments during training (above) where we also observed little Sst44 cell activity before mice learned the task. In addition, this result is consistent with a course correction signal because the mouse is only expected to make course corrections after it knows that the reward location has switched (i.e., after it has learned the task).

In addition to these new experiments, we have added new text to the Discussion that includes possible mechanisms by which Sst44 cells could be involved in learning (line 409).

Referee #2

This paper uses genomic enhancers, optogenetics and imaging to identify the functional role of a cortical interneuron sub-type in behavior. First, the authors target SST+interneurons, using the SST44 enhancer and interestingly, show that increased SST44 accessibility correlates with increased genomic accessibility of Arc and Fos. Next, they show that the activity of individual SST44 neurons in the PPC is highly correlated with the activity of other SST33 neurons, suggesting that this neuronal cell type forms a specialized subnetwork. This idea is further supported by the targeted activation of individual SST neurons, in which photostimulation of small groups of SST44 neurons led to a net excitatory effect on other SST44 neurons, even when the non-targeted neurons were several hundred micrometers away from the closest photo-stimulated target. Perhaps interestingly, targeted activation of SST44 neurons did not influence the animal's behavior. Finally, the authors investigate the activity of SST44 neurons during a goal-directed navigation task and report that SST44 neurons are most active when mice correct for deviations in their heading direction (i.e. bring current heading back towards average heading at a particular location). This is also observed when mice correct experimenter induced heading deviations. The authors thus conclude that SST44 neurons signal 'error correction'. While much of their analysis supports this idea, there are several shortcomings to the paper.

1. Perhaps most critically, it appears as if the SST44 neurons are most strongly activated when mice vigorously

correct for heading deviations (i.e. a large rotational acceleration). For example, Figure 5D and extended data Figure 9, appears to show that when mice correct their heading with lower acceleration the heading deviation is corrected almost as quickly as the high acceleration case, but SST44 neurons show extremely low levels of activity. Thus, even though the error is corrected on similar timescales, SST44 neurons are much less active – which would complicate the interpretation that provide a true general navigational ‘error signal’.

Thank you for raising this point, which has encouraged us to do more detailed analyses of our data. These new analyses reveal that Sst44 cell activity is related to the strength of both the corrective action (turning acceleration) and the error (heading deviation) (Extended Data Figure 16). Specifically, by selecting trials where the corrective action was moderate (in between the weak and strong turning accelerations shown in Figure 6), we observed a moderate Sst44 cell response (Extended Data Figure 16a). In addition, we considered in more detail the effects of the heading perturbations that we applied in virtual reality (Figure 6). We realized that the same heading perturbation can result in different heading deviations depending on the mouse’s starting heading. For example, if the mouse started with low heading deviation, then the perturbation will result in a large heading deviation. Alternatively, if the mouse started with a large heading deviation, the perturbation may push the mouse back to a low heading deviation. If we select trials where the heading deviation after the perturbation was low, we observed little activity in Sst44 cells (Extended Data Figure 16b). On the other hand, Sst44 cell activity was stronger for moderate and high heading deviations (Extended Data Figure 16c-d).

We agree with the reviewer that the weakest corrections do not produce noticeable Sst44 cell activity, and therefore that Sst44 cells do not respond to *all* error corrections. Instead, our new analyses indicate that Sst44 cells respond in a graded manner with the largest responses to those course corrections with a stronger error or corrective action. We feel these new analyses add significantly to our understanding of the variability in Sst44 cell activity that the reviewer noted.

We added a dedicated section in the Results (line 303) to make this point.

In addition, we performed a similar analysis for course corrections due to self-induced, rather than experimenter-induced, heading deviations and found similar results (line 224-238, Extended Data Figure 10).

2. The authors note that increased SST44 accessibility correlates with increased genomic accessibility of Arc and Fos, and speculate that SST44 could therefor label recently active Calb2+ and Hpse+ cells. However, it is unclear if increased Arc/Fos accessibility correlates with recent activity. In addition, if this were indeed the case, this could be problematic as it would mean that expression driven by the SST44 enhancer would depend on the activity of neurons in the injected region around the time of injection.

We agree with the reviewer that our previous notes about the relationship between expression from the Sst44 enhancer and activity were somewhat speculative. To test if the relationship between activity-dependent genes and the Sst44 enhancer is relevant to our experiments, we performed new experiments in which we compared Arc expression (which correlated with Sst44 enhancer ATAC accessibility) in Sst44+/Sst+ cells and Sst44-/Sst+ cells one hour after mice performed the T-maze task. We observed no difference in Arc expression in Sst44+/Sst+ cells and Sst44-/Sst+ cells (Referee 2 Figure 1). We therefore conclude that our AAV is unlikely to label cells based on recent activity. These new results have caused us to revise our thinking about the activity-dependence of the Sst44 enhancer. We have therefore removed the activity-dependent analysis of the ATAC-seq dataset from the paper since they do not appear to be relevant. We note that the AAV construct we used was designed with elements to stabilize expression levels and was not intended to measure transient changes in expression.

Referee 2 Figure 1. Sst44 cells do not express more Arc RNA

3. It is interesting to see that activation of SST44 neurons in RSC is also synchronous but not driven by corrective movements. However, the absence of this signal in RSC isn't sufficient to say that the function is specific to PPC. Repeating the experiment in V1, for example, which plays a role in course corrections would strengthen this claim however.

We agree that it is possible that the error correction activity in Sst44 cells may be present in other areas, such as V1. We find it interesting that the Sst44 cell signal for course corrections is not present in RSC because RSC is also involved in navigation. We simply wanted to make the point that this signal is not present in all areas, even though the Sst44 cells still show synchronous activity in another area. Further experiments will be required to show if Sst44 cell activity is indeed distinct in each area. We have clarified our thinking on this point by revising the text in the Results (line 367) and the Discussion (line 467). Specifically, we clarify that our results do not prove that this signal is specific to PPC, and we also expand on our thinking about how Sst44 cells may function in other parts of cortex.

4. It wasn't clear to me if all SST44 neurons are active independent of the deviation of direction, or if is there a selectivity for right or left corrective movements in some cells. Other work, for example, has shown that excitatory neurons (e.g. in motor cortex) are preferentially active either during left or right turns in a similar task (Heindorf, Keller, Arber, Neuron 2018).

We performed new analyses to address this point. We measured the change in activity for single Sst44 cells during left and right course corrections (Extended Data Figure 12c). Many Sst44 cells responded to both left and right course corrections, with some more selective for one or the other. We added these findings in the Results (line 266).

5. Related to point 4, in ED9, the authors show that average SST44 activity split by right vs left deviations. It is hard to say from the plotted lines, but it looks like corrections to right deviations lead to stronger activity.

We performed new analyses to examine this point more carefully. Because we noted above in response to the reviewer's point #1 that the magnitude of the Sst44 cell response is related to the magnitude of both the corrective action and the heading deviation (Extended Data Figure 10, Extended Data Figure 16), we limited the range of heading deviation and turning acceleration for left and right corrections (Extended Data Figure 12b, see legend for details). This allowed us to more directly compare the magnitude of Sst44 cell activity on corrections in the different directions. This new analysis showed that the response to right and left deviations is mostly similar, although with a mild rightward bias, as suggested by the reviewer. We observed the same mild rightward bias in single Sst44 cells (Extended Data Figure 12c, see comment above). As we noted in the original manuscript, the mice also had a behavioral bias toward making large rightward errors (Extended Data Figure 12e), although this bias was not present if we included smaller errors (Extended Data Figure 12d). However, because the activity bias is not very strong and because our data do not allow us to indicate the cause of the bias, we have not emphasized this point in the paper.

6. There are several places where figure legends lack important details. For example: Fig 1A: Colors, Fig 4E [now Figure 5f]: average of how many cells in each group?, Ext Fig 6 [Now Ext Fig 7]: burst rate is not defined.

We added these details in the respective figure legends:

Figure 1a: *“Leiden clustering of cell types shown as different colors.”*

Figure 5f: *“Average of 706, 50 and 30 cells for non-Sst, Sst+/Sst44-, Sst+/Sst44+, respectively.”*

Figure 6f: *“Average of 410, 26 and 21 cells for non-Sst, Sst+/Sst44-, Sst+/Sst44+, respectively.”*

Extended Data Figure 7b: *“Sst44 cell population burst rate as a function of maze position. Sst44 cell population bursts defined as Sst44 cell population activity > 0.4 (contiguous time points that exceed this threshold are counted as one event).”*

7. Details in the Methods appear incomplete.

We apologize for these missing details. We have added them, as outlined below.

- Classifying cell types with cellpose: what was the version of cellpose, segmentation model used?

We added this information in the Methods:

“Red cells (Sst-Cre+) were defined in the same way, except we also included cells that had a spatial correlation > 0.7 with the closest cell mask defined by Cellpose (version 0.0.2.8, default cyto model).”

- UMAP both for ATAC seq embedding and activity embedding should also include details such as version number, distance function used etc.

For the activity embedding we added the following line in the Methods:

“UMAP parameters: version=0.5.1, n_neighbors=10, min_dist=0.1, metric=‘euclidean’.”

For the ATAC-seq embeddings, we added the following in the Methods:

“[...] visualized cell type clusters with UMAP³ (default parameters within SnapATAC).”

SnapATAC version is also included earlier in this section:

“We used SnapATAC⁴ (version 1.0.0) [...]”

- Photostimulation: what was the stimulation frequency, duration of spiral, duty cycle?

We added this information in the Methods:

“These spots were scanned in a spiral pattern with a 6.6 μm radius over 32 ms (one frame) to excite the entire cell.”

“Photostimulation occurred on alternating frames (30 Hz frame rate, 15 Hz stimulation rate, 50% duty cycle) and lasted 1 second.”

- Influence mapping: Unclear if this was done during the task or in the dark?

We clarify that we performed influence mapping during the task in the Methods (line 1303).

To further clarify, we added “during navigation” in the title of Figure 3, added a diagram in Figure 3a, and also state this in the Figure 3 legend.

- For triggering on SST44 activity, it is unclear if activity of all co-recorded SST44 neurons is averaged, and the triggers calculated on that average.

We apologize that this was not clear. We first averaged co-recorded Sst44 neurons in a given session and then triggered on that average. We added a sentence clarifying this point in the Methods (line 1542).

8. For imaging experiments, authors note that they sample same field of view multiple times, except for the photostimulation experiments. In many of the analyses it is unclear if the data presented is averaged across cells, i.e. treating each cell from each session as independent, even though the authors state themselves that that is not the case. The authors should clarify and demonstrate that their conclusions hold even when considering that cells from multiple sessions of the same animal. For example by first averaging within animals, and then averaging between animals.

As the reviewer notes, the cells across sessions from the same mouse are not necessarily different. However, this is also true for cells across trials within the same session. Since our aim in Figures 5 and 6 was to compare activity across trials, and not across cells, we therefore pooled trials across sessions to compare more data, conceptually treating trials in the same and different sessions as equivalent. Note that we did not do this for Figure 2, where we measure cell type statistics for each mouse and not each session. We clarify this point in the Methods (line 1537).

As suggested by the reviewer, we performed an additional check to make sure our conclusions hold for independent cells. We repeated our main analysis from Figure 5 using one session from each mouse, in which case cells from each session will be independent. The results, as plotted below, are highly similar to those reported with the analysis in the paper.

Referee 2 Figure 2

Figure 5h,i analysis repeated using one session from each mouse. **h**, Activity and behavior as the mouse entered the T-junction, split based on whether this was followed by a large ($> \pi/6$) or a small ($< \pi/12$) heading deviation. Left and right deviations were pooled after inverting behavior for left deviations. Selection criteria were evaluated at +1.5 s after entering the T-junction. **i**, Same as **h**, splitting trials with a high deviation based on whether the mouse corrected strongly (turning acceleration $> 1 \text{ rad/s}^2$ in the opposite direction) or weakly ($< 0.5 \text{ rad/s}^2$). Wilcoxon rank-sum test across trials, Sst44 cell activity solid vs dashed line: $p < 1e-4$. Wilcoxon signed-rank test across trials, Sst44 cell activity vs other cell types: $p < 1e-10$.

9. Along similar lines, in many instances, number of animals and total number of sessions is listed, but it is unclear how the sessions are distributed among the animals.

We added this information in the Methods (line 1136).

Referee #3

Green and colleagues identify a subset of somatostatin positive neurons (Sst44+) in the mouse association cortex. In a series of elegant experiments, the authors develop a method to specifically target, observe, and perturb this population to show that Sst44+ neurons tend to fire synchronously (often near the end of the virtual track), inhibit neighboring neurons, and are most strongly activated when mice exhibit course corrections on a cued navigation task. As with other work from the Harvey laboratory, the experiments are carefully conducted with numerous controls and the manuscript is well written. Accordingly, I have very few technical comments overall. That said, I think there are more general issues with interpretation that may require rethinking base claims or adding additional experiments in order for this work to merit publication (especially in such an impactful journal).

Main comments.

The authors make a strong case for the impact of this work purely based on their cell-type-specific targeting of a sub-class of Sst neurons (which I mostly agree with). While they conduct photostimulation experiments to map the functional circuits centered on these neurons there is little other rationalization for why, at an anatomical, physiological, or circuit level, this population should play a significant role in signaling and/or mediating error corrections in navigation. While the experiments specifically examining the Sst44+ population are certainly impressive, there is a bit of a ‘black box’ feel to the main findings concerning these cells. It would be useful to know more about how these neurons project within PPC (or beyond). This anatomy is especially important considering the lack of a similar role of Sst44+ neurons in RSC. Moreover, I think the authors should more concretely consider how this small subset of cells might engage other neurons in the area. A model could provide insight into how the Sst44+ population can alter processing dynamics given error.

We thank the reviewer for the feedback and suggestions on ways to improve the paper. In our opinion, our paper makes two major contributions to the field. First, we assign a precise cognitive signal to a molecular cell type in cortex. There are relatively few examples of this in the literature, especially when considering “cognitive” functions and sub-classes of cell types in the cerebral cortex. Second, we have identified an error correction signal for course corrections during navigation. To our knowledge, a navigation error correction signal has not been identified previously, and thus our findings open new avenues for understanding how error signals can guide the execution and learning of navigation behaviors. We feel that the identification of a new error correction signal with its assignment to a molecular cell type is an important advance for the field. We agree with the reviewer that an important next question is to investigate the mechanisms by which this cell type signals error corrections and mediates effects on the circuit. Related to this question is why this sub-class of Sst neurons may be particularly well suited for this function. As we considered these questions and ways to address them, we realized that experiments to fully flesh out these mechanisms will necessarily be extensive and exploratory in nature and that such experiments would be necessary prerequisites for building a mechanistic model. While, in our opinion, it will require a dedicated follow up study to fully understand the mechanisms of action of these cells, we agree with the reviewer that providing more insight into how this population of cells may be acting is helpful.

We have therefore added new experiments and analyses that aim to provide a better understanding of how these cells could contribute to error corrections. We feel these new experiments and analyses add significantly to our understanding of Sst44 neurons. In addition, we have added new discussion material that incorporates previous work from the literature on the function of Sst neurons. We feel these new

analyses and conceptual text additions add toward the understanding of why this population is a good one for signaling error corrections.

First, as suggested by the reviewer, we performed morphological reconstructions of individual Sst44 cells in PPC by filling cells with a dye through a patch pipette (Figure 1j). We found that Sst44 cells send processes to or towards layer 1, both from layer 2/3 and from layer 4. These projection patterns are consistent with those reported for Calb2+ and Hpse+ Sst cells by other work⁵, and help to further characterize the cells targeted by the Sst44 enhancer. Importantly, these data are consistent with the expectation that cortical interneurons do not project outside of their local cortical area. This local connectivity could explain why we observed activity related to course corrections in Sst44 cells in PPC but not in Sst44 cells in RSC. We highlight this point in the Discussion (line 467).

Second, as suggested by the reviewer, it is important to consider how Sst44 neurons might engage other neurons in the area. Given that Sst44 neurons in layer 2/3 (whose physiology we characterized in this study) project within layer 2/3 as well as to layer 1, and given that Sst cells in general are known to target dendrites, we expect that Sst44 cells inhibit the dendrites of excitatory pyramidal cells within these layers. Moreover, layer 1 is known to receive feedback connections from other cortical areas, and Sst44 neurons might target layer 1 dendrites to modulate or gate signals from these feedback connections. At present, we do not know which feedback connections would be affected, making it challenging to develop a well constrained model. However, we present several general possibilities. Modulating these feedback connections could serve a role in regulating synaptic plasticity between local circuits and feedback connections. Consistent with this hypothesis, Sst cells are known to regulate the spine density of apical pyramidal cell dendrites in layer 1 in motor cortex during learning². This spine density regulation is also required for learning². The layer 1-projecting Sst44 neurons in PPC might therefore play a similar role in learning to navigate toward the reward. Another possibility is for Sst44 cells to serve a “command” signal role, for example by gating feedback connections that are appropriate for executing the correct action. We highlight these possibilities in the Discussion (line 409).

Regarding the question of why an error correction signal would be present in this particular cell type, we note that, among the different classes of inhibitory neurons, Sst neurons are known to target dendrites, and are therefore poised to modulate synaptic connections. Thus, if this error correction signal functions as a learning signal, it would make sense that it is present in a cell type that is poised to use this learning signal to modify synapses. In contrast, parvalbumin neurons, another class of inhibitory neurons, target cell bodies, and might be involved in regulating neuronal activity rather than synaptic plasticity.

In addition to these new anatomical data, we performed a new mechanistic characterization of the excitatory interaction among Sst44 neurons. While we reported in the original manuscript that Sst44 neurons excite one another, it was not clear how this happens for an inhibitory neuron population. We therefore sought to characterize in more detail the circuit mechanism that underlies the synchronous activity of Sst44 cells, which we had shown is a major feature of their activity. In addition to our original *in vivo* photostimulation experiments, which showed that Sst44 cells excite each other, but not other Sst cells, we have now performed slice electrophysiology experiments on pairs of Sst44 neurons. These experiments show that Sst44 cells are gap junction coupled (Figure 4) and thus identify a circuit mechanism for how Sst44 cells excite each other even though they release inhibitory neurotransmitters. The excitatory interactions among Sst44 cells in turn help to explain their synchronous activity, a major feature of the error correction signal. These new results therefore help us to understand why this cell type is well suited for generating the error correction signal.

Together, we hope these new experiments, analyses and text additions help to provide a better understanding of these cells. In particular, our new experiments with paired patch clamp electrophysiology reveal a way in which the synchronous activity could arise. Our new anatomical tracing experiments indicate why these cells might have an area-specific function, and point to these cells being involved in modulating feedback connections. Further, we provide new discussion text, building from the literature on Sst cells, to indicate how these cells may mechanistically contribute to the circuit and why Sst Martinotti

cells may be a good cell type for an error signal. We feel our mechanistic understanding of these cells has improved through these revisions, and we thank the reviewer for encouraging us to make additions in this direction.

The key finding is that Sst44+ neurons are synchronously activated when mice exhibit course correction behaviors to reorient during virtual navigation. Critically, the authors show by omitting the reward that Sst44+ cells are not signaling error generally but instead navigation related errors. To me, a course correction felt kind of arbitrarily defined and potentially highly dependent upon the experimental configuration. Many of the example course corrections possess a very specific movement profile (high angular acceleration/velocity) that could relate to controlling the inertia of the floating ball (suggesting that the current interpretation could be dependent upon the experimental configuration). Because of this, course correction is strongly conflated with a specific action state related to reorienting in virtual space. Given that PPC neurons are sensitive to changes to velocity and acceleration (Whitlock et al., 2012), it would be interesting to see these Sst44+ fire for an error that is decoupled from a reorienting response (e.g. if the animal were to overshoot or undershoot a rewarded area on a linear track, start the trial early or late, or if the cues were swapped mid trial or their contingencies reversed). Similarly, controls showing that Sst44+ cells are similarly active when the animal course corrects but the movement of the floating platform is altered are critical (or its relationship to visual feedback as in the comment below).

We agree that it is critical to separate neural activity related to course corrections from neural activity related to the act of turning or reorientations more generally. We have now performed new analyses and two new experiments that rule out the possibility that Sst44 cells respond to the act of turning alone. These new experiments and analyses provide additional lines of evidence to support our main conclusion that Sst44 cells activate during course corrections.

First, we performed a series of new analyses. Importantly, not all turning events are associated with course corrections. For example, the mouse turns into the T-maze arms as a normal part of its trajectory. Thus, it is possible to dissociate turning accelerations and course corrections. To show that the Sst44 cell activity is not due only to reorientations or turning acceleration generally, we performed a new analysis where we compared turning events with the same turning acceleration and velocity profile, under conditions of a high or low heading deviation (Figure 5j). If Sst44 cell activity mostly reflects a response to reorientation, we would expect to see the same activity for similar turns in the cases with low and high heading deviation. However, we observed significantly higher Sst44 cell activity when these turning events occur during a large heading deviation (Figure 5j, solid line), and little Sst44 cell activity when these turning events occur during a low heading deviation (Figure 5j, dashed line, also compare the behavior and activity in this trace to the solid line in Figure 5i, where the turning profile is very similar, but the Sst44 cell activity is very different). We conclude that because these turning events are matched in terms of turning velocity and acceleration, this analysis strongly indicates that Sst44 cells are not responding more generally to a “reorientation response”.

In addition, we added a new analysis that compares Sst44 cell activity when the mouse enters the T-junction with strong or weak accelerations, but in both cases with low heading deviations. Similar to the analysis above, we observed low Sst44 cell activity even on trials with high turning, consistent with their activity not reflecting a general turning or reorientation response (Extended Data Figure 10b).

We highlight these new analyses in the Results (line 232).

In addition, we performed new analyses of our heading perturbation experiments. In these experiments, depending on the heading of the mouse at the time of the heading perturbation, the result of the heading perturbation could be a low, moderate, or high heading deviation. For example, if the mouse has low heading deviation before the heading perturbation, it will end up with high heading deviation afterwards. Instead, if the mouse has a large heading deviation before the heading perturbation, it is possible that the heading perturbation pushes the mouse back toward the correct trajectory, resulting in a low heading deviation after the perturbation. We parsed the data based on whether, at the end of the heading perturbation,

the mouse's heading deviation was low, moderate, or high (Extended Data Figure 16b-d). We observed that when the heading deviation was low (Extended Data Figure 16b), there was little Sst44 cell activity, even when we selected for trials with strong turning (compare solid lines in Extended Data Figure 16b-d). These results further confirm that Sst44 cells do not respond solely to turning or reorientation events. These results are highlighted in the text (line 303).

We also performed two new experiments that help to rule out the possibility that Sst44 cells respond to turning alone. In a first experiment, we imaged activity in PPC during sessions early in training, before the mice reached high performance on the task (Extended Data Figure 11). A course correction toward a reward site implies that the mouse knows the reward location. If Sst44 cells signal a course correction, then we expect Sst44 cell activity to be low before the mice learn the task, and to increase with the learning of the task. Consistently, we found little activity in Sst44 cells during large turning events toward the reward site early in learning and increasing activity as mice got better at the task. If Sst44 cells responded mostly to turning, then we would expect Sst44 cell activity to be similar at all stages of learning of the task. Since we did not observe Sst44 cell activity during turning at early stages, we conclude that Sst44 cells do not respond mostly to turning. Instead, our results are consistent with the conclusion that Sst44 cells signal an error correction.

In a second experiment, we performed the experiment suggested by the reviewer. We switched the cue and the location of the reward partway through a trial, requiring the mouse to update its trajectory partway through a trial (Extended Data Figure 17). For example, we switched from a white cue to a black cue partway through a trial, indicating a switch in the reward location from the right T-arm to the left T-arm. We observed an increase in Sst44 cell activity time-locked to the cue switch, consistent with the expectations for a course correction signal. Importantly, this increase in Sst44 cell activity at the cue switch was only present after mice had learned the task, providing further evidence that Sst44 cells only respond once mice have learned the location of the reward. This new experiment, as suggested by the reviewer, provides a complementary line of evidence to support the conclusion that Sst44 activate during a course correction and not during turning alone. It also shows that these course correction signals can occur in different contexts, further supporting the idea of a course correction signal as opposed to a turning signal.

Taken together, these new experiments and analyses demonstrate that Sst44 neurons in PPC do not generally activate during reorientations, but rather have activity that is specific to course corrections, including course corrections under different contexts.

Analyses excluding a role of virtual angular velocity (5h [now 6h]) and visual flow information on activation of Sst44+ neurons were not convincing in my opinion. The authors should look at activation of this population during a passive version of the task, a version with no optic flow (e.g. all black walls), and a version in which movement and visual feedback are decoupled.

We thank the reviewer for suggesting these experiments to strengthen our conclusion that Sst44 cells are not responding mostly to visual flow or movement. As suggested by the reviewer, we performed new experiments where we played back in open-loop the visual scene from the heading perturbation experiments that were originally performed in closed-loop. In this way, we could ask whether the Sst44 cell activity that we observed in the heading perturbation trials was due solely to the visual input, since the visual input was matched across the open loop and closed loop cases. We observed little Sst44 cell activity when the heading perturbation trials were played back in open loop (Extended Data Figure 18b), showing that these cells do not respond mostly to the visual stimulus, including visual flow. We report these new experiments in the Results (line 340).

As we describe in our response to the question above, our new analyses in Figure 5j also show that turning events with matched virtual angular velocity induced very little Sst44 cell activity under conditions of low heading deviation (Figure 5j dashed line). These turns have the same visual flow to those in the cases of

high deviations, but Sst44 cells are only active during the turns with high heading deviations. This analysis thus further indicates that Sst44 cells do not respond generally to visual flow.

Therefore, based on analyses and experiments with matched visual inputs, we conclude that Sst44 cells do not respond mostly to visual stimuli or visual flow. Instead, these experiments and analyses are consistent with the conclusion that Sst44 cells respond specifically during course corrections.

Minor comments.

The co-activity based UMAP projections and quantification in Figure 2i-l are not especially compelling in my opinion as there are multiple other Non-Sst clusters of similar quality as well as Non-Sst cells embedded within the Sst44+ cluster in 2i. It is unclear what these plots add beyond 2g and 2h.

The aim of Figure 2h-i (note modified figure panels) was to provide an additional analysis and visualization of the point that Sst44 cells activate in a different pattern than other Sst and non-Sst cells. While these analyses make a similar point to Figure 2g, we feel they are still helpful in visualizing this important finding about synchronous and specialized activity in Sst44 cells. Specifically, these plots highlight that Sst44 neurons cluster together in terms of their activity, whereas Sst44-/Sst+ neurons are distributed among other cell types.

We agree that there are multiple non-Sst clusters. Our goal was not to conclude that Sst44 cells form the only cluster. Rather, our point is simply to show that Sst44 cells do cluster in terms of their activity. Many neuronal subtypes exist within the population of non-Sst cells, and we might expect that other activity clusters within this category will form if other neuronal subtypes also display unique activity patterns. However, the existence of other activity clusters within non-Sst cells is separate from the point that Sst44 cells label their own activity cluster.

It is true that there are non-Sst cells within the Sst44+ cluster in Figure 2h. This result could arise in multiple ways. First, it is expected that not all Sst and Sst44 cells are labeled by the transgenic line and the Sst44 enhancer virus. Notably, we did find some clusters that only have Sst and Sst44 cells, as in Extended Data Figure 21e. Second, the ability to separate out different clusters depends on multiple factors, including the number of datapoints. As we have observed with single cell ATAC-seq, more cells will generally provide greater resolution when clustering cell types, and we expect the same to be true for clustering cells based on activity. Given the small number of cells sampled per cell type, including Sst44 cells, this factor is likely to be limiting. Third, it is of course possible that some non-Sst cells have a similar pattern as Sst44 cells, but that these cells are relatively rare. We agree that this overall point is important to note and now highlight it in the Results (line 121).

3f - What is the failure rate of target stimulation? It appears that many stims do not yield a calcium transient. Do failed stims have any temporal relationship to stimulation of other Sst44+ cells within the sweep?

We apologize that our example trace for photostimulation may have given the wrong impression and was not very informative, as another reviewer pointed out as well. We added a new zoomed in example trace in Figure 3f that better highlights spontaneous and stimulated activity in targeted and non-targeted cells. In most cases, the stimulation was effective at evoking a calcium transient.

We have now performed analyses on the success rate of stimulation. We find that, of the cells we targeted, 78% were significantly activated relative to control trials. Of the cells that were significantly activated, our stimulation success rate on individual trials was 75%. We now include this information in the Methods (line 1342).

We observed a weak correlation (Pearson $R=0.15$) in the photostimulation-induced change in activity across co-stimulated cells. Because this correlation is weak, we are not sure of how to interpret it and thus decided not to report it.

In figure 4e-i, Sst44- and Sst44+ cells appear to have correlated activity patterns while the influence mapping experiments in Figure 3 indicate that they should be anticorrelated. This is a bit confusing and I'm curious if the authors have any thoughts about this? Is it possible that synchrony among these Sst populations is state dependent?

We agree this is an important point to address. First, while it is possible that the interactions among Sst subtypes is state dependent, we measured the interactions among these cell types in the same task condition where these activity correlations were measured. The state of the network was therefore well controlled between our imaging and photostimulation experiments, and we therefore conclude that state-dependence is unlikely to explain the difference between these experiments.

In our view, the most likely interpretation of this difference is that these two subtypes are receiving weakly correlated inputs. Even if two cells are strongly correlated, this correlation does not necessarily mean that the two cells are strongly connected – they may in addition receive similar inputs. Indeed, influence mapping is a method that can identify interactions between cells without contributions from common inputs, whereas activity correlations include both cell-cell connections and common inputs. We therefore interpret the positive activity correlation among Sst44 and Sst44-/Sst+ cells to be due to these cell types receiving similar signals from their presynaptic partners. We also highlight that although the influence of Sst44+ cells on Sst44-/Sst+ cells is trending negative, this effect is not significant (see Figure 3 legend), and we interpret this result as there being little-to-no interaction between these subtypes.

We agree that this point is important to highlight. We have added new text in the Results (line 157).

Is there more Sst44+ activity on delay maze trials wherein there are presumably more errors? What about in sessions prior to reaching the 85% criteria?

We compared the number of course corrections and Sst44 bursts between sessions with and without a delay (Referee 3 Figure 1 below). In both the sessions with and without a delay, the mouse performed with high accuracy. There was little difference between sessions with and without a delay, for both Sst44 burst rate (Referee 3 Figure 1a) and error correction rate (Referee 3 Figure 1b). We did not include this figure in the paper because there was no difference to highlight.

We performed new experiments to look at sessions earlier in training (Extended Data Figure 11). We found that early in training there was less activity in Sst44 cells. At intermediate levels of behavioral performance, we observed intermediate levels of Sst44 cell activity. These results are consistent with our conclusion about a course correction signal. Early in training, the mouse does not know where the reward is located on each trial and thus cannot execute course corrections toward the reward site. These course corrections only become possible as the mouse starts to learn where the reward is located on each trial. Thus, we expect the course correction activity to track the behavioral performance of mice over time. This is indeed what we observe for Sst44 cell activity.

Referee 3 Figure 1

a, Sst44 cell population burst rate as a function of maze position, comparing mazes with and without delay. Sst44 cell population bursts defined as Sst44 cell population activity > 0.4 (contiguous time points that exceed this threshold are counted as one event). **b**, Course correction rate as a function of maze position, comparing mazes with and without delay. Course corrections defined as heading deviation $> \pi/6$ and turning accel. $> 1 \text{ rad/s}^2$ in the opposite direction, delayed by $+0.3 \text{ s}$ (to account for the average delay in the mouse's reaction – see Methods). Bar plots show the mean and bootstrapped 95% confidence intervals. Deconvolved activity was normalized by the 99th percentile for each neuron before smoothing with a 0.25 s gaussian filter. 7 mice, 27 sessions.

Referee #4

In this study, Green et al. report that a subset of SST+ cells in the PPC (identified via the enhancer Sst44+) tend to fire synchronously with other Sst44+ cells, and excite each other. In a navigation task, they report a cell-type specific error correction signal, one in which that these Sst44+ neurons in PPC (but not RSC), but not other SST or non-SST cells are activated during error correction. There are additional experiments and analyses that would be required to support the hypothesis that Sst44+ cells are truly a unique population with specialized coding properties during navigation for reward.

Detailed comments:

1) Fig 1: The authors should report the identity of the fraction (~10%) of Sst44+ cells that are not SST+, as this may have implications for their subsequent results. Are they another subclass of inhibitory neurons? In addition, what fraction of the total SST neurons counted in L2/3 are Sst44+?

Thank you for raising this important point. Based on an initial characterization of the Sst44 enhancer by Hrvatin et al. 2019⁷, the non-Sst cells that are labeled by the Sst44 enhancer are mostly parvalbumin (Pvalb) inhibitory neurons, as indicated by the single cell RNA sequencing screen (Hrvatin et al. 2019 Figure 2j) and colocalization with the Pvalb-Cre line (Hrvatin et al. 2019 Figure 3h).

To increase the specificity of the cells we analyzed, we only considered Sst44 cells that are also Sst-Cre+ for all imaging and photostimulation experiments. This means that the small Pvalb+ population labeled by the Sst44 enhancer are not included in our results. We now report the identity of Sst44 neurons that are non-Sst and emphasize that we focused on the Sst44 cells that are also Sst+ (line 104).

We find that the fraction of Sst neurons that are Sst44+ in L2/3 is $44 \pm 4\%$. We now report this number in the Results (line 108).

2) Fig 2d: Authors should validate in the PPC that the TdTomato signals from the Sst-Cre line are indeed only labeling SST+ neurons to ensure no leak in the viral expression. Similarly to the point above, what kind of neurons are Sst44+ cells that are not SST-Cre+?

We performed new RNAScope experiments to measure Sst RNA expression in mice carrying the Sst-Cre and tdTomato reporter alleles. We verified that the Sst-Cre line is specific for Sst cells in PPC ($90 \pm 3\%$, 4 mice) and now report this number in the Results (line 106).

We also highlight that we validated the specificity of the Sst44 enhancer using the Sst RNA probe (90% specificity, Figure 1g), providing an assessment of its specificity for Sst independent of the Sst-Cre line.

We address the point about Sst44 cells that are non-Sst in our response to the reviewer's comment #1 above.

3) Fig 3f: . The manuscript would benefit from a clearer example of traces to validate the stimulation, one in which the reader can see the time window of stimulation and the subsequent Ca transient in optically targeted and non-targeted neurons. In its current form it is very difficult to tell the differential effects of target vs. control optical simulations

We agree and appreciate the suggestion. We replaced the example trace with a new one that better highlights the calcium transient, displays the stimulation window, and includes non-targeted neurons in Figure 3f.

4) Fig 3h-i: it is difficult to reconcile the relationship between Sst44+ and Sst44-, as well as Sst44+ and non-SST+ cells shown here, vs. the correlations shown in Fig 2 and Extended Fig 10. Specifically, here it seems that Sst44+ stimulation leads to inhibition of non-SST and Sst44- cells, but the correlations between Sst44+ and Sst44- populations are positive in Fig 2 (and Extended Fig 10).

We agree this is an important point to address. The most likely interpretation of this difference is that these two subtypes receive weakly correlated inputs. Even if two cells are strongly correlated, this correlation does not necessarily mean that the two cells are strongly connected – they may in addition receive similar inputs. The activity correlation measure cannot separate the contribution of shared inputs and connectivity between cells. Indeed, this is the reason we performed influence mapping because it allows us to measure the contribution of connectivity between cells without the effects of shared inputs. We therefore interpret the positive correlation among Sst44 and Sst44-/Sst+ cells to be likely due to these cell types receiving similar signals from their presynaptic partners. We also highlight that although the influence of Sst44+ cells on Sst44-/Sst+ cells is trending negative, this effect is not significant (see Figure 3 legend), and we interpret this result as there being little to no interaction between these subtypes.

We agree that this point is important to highlight. We have now added new text about this point in the Results (line 157).

5) A central component of this study is the synchronization/net excitatory effect Sst44+ cells have with one another. This is a highly impactful finding that should be explored in more depth. In order to achieve a better understanding of this, the authors should attempt to describe this connection more rigorously with another method such as slice electrophysiology.

We thank the reviewer for the positive feedback about this result and for the suggestion to perform further investigations with slice electrophysiology. We have taken the reviewer's suggestion and performed an extensive set of new experiments.

In new experiments using patch clamp electrophysiology in brain slices, we show that pairs of Sst44 cells are connected through gap junctions (Figure 4). We patched two Sst44 cells simultaneously, which was challenging because this is a rare cell type. First, we show that a hyperpolarizing current in one cell hyperpolarizes the second cell in about half of pairs (Figure 4b-e). This protocol is standard in the literature to measure gap junction coupling¹. Second, we show that this connection is rapid with a latency of approximately 1 ms (Figure 4c-d). Third, we show that this connection is reciprocal, as expected for electrical coupling (Figure 4f). Fourth, we show that this connection is unaffected by blockers of excitatory and inhibitory synaptic transmission (Figure 4g-h). These experiments provide strong evidence that Sst44 cells are coupled via gap junctions. Thus, these experiments are consistent with our *in vivo* optogenetic experiments during behavior, and they provide a mechanism for the self-excitation among Sst44 cells.

6) The authors state “Remarkably, the synchronous, self-reinforcing, and course correction activity was specific to Sst44 cells and not present in other neighboring Sst cells.” However the data, as presented in figures 4 and 5 don't entirely support this claim. From the data in Figs 4F-I and 5G it seems that the Sst44- and non-SST cells also correlate with heading deviation and turning acceleration, albeit the effects are larger in Sst44+ population. This can also be seen in the heat maps where a significant portion of activity in the heat maps of Sst44- cells occurs in the same location of the maze as the synchronous activation of the Sst44+ cells. This raises the question of how specialized are the Sst44+ cells in this response to course correction? More detailed analysis of the single neuron and population response properties are warranted, particularly comparing the highly correction selective SST+ Sst44- cells to the 44+ population as well as the response properties of Sst44- neurons in this task.

We agree this is an important point. We have performed a new analysis to address it. Given that viral transduction is variable from cell to cell, it is very likely that not all true Sst44 cells will be labeled by the Sst44 enhancer virus, and that these cells will therefore sometimes show up in the Sst44-/Sst-Cre+ population. We therefore expect a minority of the Sst44-/Sst+ population to respond similarly to Sst44 cells, which will be evident in the population means, as for example when we measure activity in response to course corrections. In addition, it may be the case that true Sst44-/Sst+ cells also have some degree of similarity with Sst44+/Sst+ cells.

To address this point, we performed a new analysis on individual Sst44-/Sst+ neurons. We clustered all cells based on their activity using the Leiden community clustering algorithm⁸, selected the cluster that was populated by Sst44 cells, and asked how many of the Sst44-/Sst+ neurons fall into this cluster. As expected from the sample UMAP in Figure 2h, a minority (24%) of Sst44-/Sst+ cells co-clustered with Sst44 cells. This finding indicates that although some Sst44-/Sst+ cells have activity similar to Sst44 cells, the majority of other Sst cells have activity that is different from Sst44 cells. We feel this new analysis is helpful in understanding the specialization of Sst44 cells. We report this analysis in the Results (line 121).

In addition, we agree with the reviewer's assessment of our wording and revised the text accordingly. For example, we revised the sentence quoted above as:

“Notably, this course correction activity was strongest in Sst44 cells and much weaker in other Sst neurons or non-Sst cells [...].”

We also modified other sections of the text and have toned down the wording used to describe the Sst44 cell activity in relation to other cell types.

We also emphasize multiple other results that highlight the point that Sst44 cells represent a functionally distinct subtype of Sst cells. First, in terms of their activity, we show that Sst44 cells are much more correlated with each other than other Sst or non-Sst cells (Figure 2g). In a second, related point, we show that Sst44 cells tend to cluster together, in terms of their activity patterns, much more often than with other Sst or non-Sst cells (Figure 2h-i). Third, in terms of their connectivity, we show that Sst44 cells on average excite each other, but not other Sst cells, and inhibit non-Sst cells (Figure 3). These analyses show clear

distinctions between Sst44 cells and other Sst and non-Sst cells in terms of their activity and in terms of their connectivity. Taken together, these results strongly indicate that Sst44 cells represent a functionally distinct subtype of Sst cells.

We did not compare the activity of Sst44 cells with Sst44-/Sst+ cells that are similar to Sst44 cells since their activities will be similar simply based on how they were selected. We agree that the general point is important to highlight though, and we have made revisions throughout the manuscript to better address it, as highlighted above.

7) Related to the above point, in Fig 4e, the example trace from Sst44+ neurons is not entirely in support of the author's central point, as there are some small increases in Sst44+ neuron activity that do not correspond to any turn correction, and some turn corrections that do not recruit Sst4+ cells.

We thank the reviewer for pointing out the variability in Sst44 cell activity, which has led us to investigate if we can understand this variability in more depth.

First, we re-examined the example from Figure 5 (formerly Figure 4). We now highlight each bump of Sst44 cell activity in Figure 5f (see below) and have now added blue arrows that are aligned in time to highlight the heading deviation and turning acceleration in each case. In Figure 5 in the manuscript, we highlight activity bumps #2, 4, 6, which are the largest ones that occur while the mouse is not in the dark. Activity bump #3 is in the dark (during the inter-trial interval), but nevertheless follows the same pattern in terms of heading deviation and turning acceleration. Activity bumps #1 and #5, which are smaller in magnitude, occur with a smaller heading deviation, but nevertheless follow the same pattern. Thus, each of these activity bumps follows a similar pattern in terms of an increase in heading deviation with an opposing turning acceleration, only with different magnitudes.

We therefore looked at what underlies the variability in the magnitude of Sst44 cell activity. As the reviewer pointed out, not all corrective movements, especially ones of smaller magnitude, are associated with a burst of Sst44 cell activity. We therefore performed a new analysis in which we analyzed Sst44 activity as a function of the magnitude of the corrections, both in terms of the magnitude of the turning acceleration and the magnitude of the heading deviation. We find that Sst44 cell activity is larger for corrections with larger turning acceleration and for corrections with larger heading deviation. These results are now reported in Extended Data Figure 10 (for corrections for self-induced errors) and Extended Data Figure 16 (for corrections for experimentally-induced errors). Thus, small corrections with small deviations and small turning are associated with less activity, sometimes with none at all (for example, Extended Data Figure 16a dashed line, and Extended Data Figure 16b solid line). We have provided expanded text in the Results that highlights the factors underlying the variability in Sst44 cell activity (line 224-238, 303).

Beyond these effects related to course correction magnitude, we agree that there is still significant variability from one event to the next that we do not entirely understand. We think that the characterization we have provided in this initial study is the first step and that further studies with different experiments will help to reveal additional sources of variability and to further characterize these cells.

Referee 4 Figure 1

8) Fig 4g/i [now Figure 5h/j]: A statistical comparison between Sst44+ and Sst44- activity across these conditions is required.

Thank you for this comment. The statistics for this comparison are now reported in the legend of Figure 5 (“other cell types” includes separate comparisons between Sst44+/Sst+ and Sst44-/Sst+ cells, as well Sst44+/Sst+ cells and non-Sst cells):

“Wilcoxon signed-rank test across trials, Sst44 cell activity vs other cell types: $p < 1e-8$ (h-j).”

9) Fig 4g/I [now Figure 5h/j]: what does activity look like in low heading deviation at T junction, but split up into low vs. high acceleration? ie/ same as 4h, but looking at low heading deviation trials.

Thank you for this suggestion. We performed this analysis and added it in Extended Data Figure 10b. We find low Sst44 cell activity on trials with low heading deviation both for trials with low and high acceleration. In addition, we directly compare this analysis to high deviation trials (Extended Data Figure 10b, right panels). The results of this analysis are consistent with our conclusion that the combination of heading deviation and corrective turning drive the largest increases in Sst44 cell activity. We now report these findings in the Results (line 232).

While unilateral stimulation did not cause any behavioral changes, the impact of this study would be increased with bilateral optical modulation of these cells to link them directly with behavior.

Thank you for this suggestion. Please see our response to Reviewer 1 Comment 1, which provides a response to the same question.

10) Are there task periods where non-SST cells fire while Sst44+ do not? This data could provide a more holistic understanding of the circuit in this behavior.

We found that Sst44 cell activity is low during trials where the mouse navigated smoothly into the reward zone (Extended Data Figure 6, Extended Data Figure 7d “low heading deviation trials”). In contrast, non-Sst cells are active on these trials as sequences of neural activity that tile the entire trial. Therefore, one distinction is the smooth trajectory trials that lack a course correction. In addition, on trials with a course correction, Sst44 cells are active synchronously at the time of the course correction, but not at other times in the trial. In contrast, the sequences of activity in non-Sst cells span the entire trial, including periods outside of when Sst44 cells are active. Thus, there are multiple ways to identify periods in which non-Sst cells are highly active but Sst44 cells are not. We now highlight this point in the Results (line 274).

11) It would help if there was a discussion of the potential roles of Sst44+ in RSC, and why they do not seem to be activated in error correction.

We agree that this is an important point for the discussion. Because Sst44 cells are local interneurons and receive local inputs, we expect their activity to be different in each area and to reflect the function of the area in which they reside. We currently do not have a good understanding of the activity-behavior relationship of Sst44 cells in RSC and did not want to propose a specific role without any evidence. However, we have now added a general hypothesis that Sst44 cells in each cortical area carry an error-related signal that is tailored to each area’s function. For example, Sst44 cells in RSC may carry an error-related signal in relation to the navigational functions of RSC. This point is highlighted in the Discussion (line 467).

12) Also in the discussion, a couple lines on basic circuit mechanism on what happens during course correction trials, and the role of Sst44+ neurons in PPC as it relates to the larger neural circuit that drives behavior?

We added sections in the Discussion on a basic circuit mechanism of what may happen during course corrections. Specifically, gap junctions among Sst44 cells could help to promote the synchronous activity in this cell type, which is a defining feature of the error correction signal in Sst44 cells. In addition, these gap junctions could contribute to computing the error correction signal, for example by averaging the inputs to the Sst44 cell population. We also propose two potential functions for how this error correction signal is used in PPC, taking into account that layer 2/3 Sst44 cells project within layer 2/3 and to layer 1, which is a significant source of feedback projections from other cortical areas. In the first potential function, the error correction signal could act as a “learning” signal, in which it helps to “teach” which navigational actions lead to a reward to other cell types in PPC. In this case, Sst44 cells could modulate synaptic plasticity between excitatory neurons in layer 2/3 as well as between these neurons and feedback projections from other areas, in such a way that the excitatory neuron population over time learns a pattern of activity that is suitable for navigating towards the reward. A second potential function is for the error correction signal to serve as a “command” signal by driving the corrective action. Sst44 cells could perform this role by gating inputs to excitatory pyramidal cells that are appropriate for the corrective action by inhibiting inputs that are related to incorrect actions. These gated inputs could include feedback projections to layer 1. The error correction signal that we observe in PPC was not present in RSC, meaning that a role in learning and driving corrective actions may be specific to PPC, whereas other areas, such as RSC may perform other functions that are necessary for navigation (such as learning locations of rewards, etc.). We have expanded our text regarding all these points throughout the Discussion (line 409-483).

References

1. Hu, H. & Agmon, A. Properties of precise firing synchrony between synaptically coupled cortical interneurons depend on their mode of coupling. *J. Neurophysiol.* **114**, 624–637 (2015).
2. Chen, S. X., Kim, A. N., Peters, A. J. & Komiyama, T. Subtype-specific plasticity of inhibitory circuits in motor cortex during motor learning. *Nat. Neurosci.* **18**, 1109–1115 (2015).
3. McInnes, L., Healy, J. & Melville, J. UMAP: Uniform Manifold Approximation and Projection for Dimension Reduction. *ArXiv180203426 Cs Stat* (2020).
4. Fang, R. *et al.* Comprehensive analysis of single cell ATAC-seq data with SnapATAC. *Nat. Commun.* **12**, 1337 (2021).
5. Gouwens, N. W. *et al.* Integrated Morphoelectric and Transcriptomic Classification of Cortical GABAergic Cells. *Cell* **183**, 935-953.e19 (2020).
6. Gouwens, N. W. *et al.* Integrated Morphoelectric and Transcriptomic Classification of Cortical GABAergic Cells. *Cell* **183**, 935-953.e19 (2020).
7. Hrvatin, S. *et al.* A scalable platform for the development of cell-type-specific viral drivers. *eLife* **8**, e48089 (2019).
8. Traag, V. A., Waltman, L. & van Eck, N. J. From Louvain to Leiden: guaranteeing well-connected communities. *Sci. Rep.* **9**, 5233 (2019).
9. Nitz, D. A. Tracking Route Progression in the Posterior Parietal Cortex. *Neuron* **49**, 747–756 (2006).
10. Harvey, C. D., Coen, P. & Tank, D. W. Choice-specific sequences in parietal cortex during a virtual-navigation decision task. *Nature* **484**, 62–68 (2012).
11. Whitlock, J. R. Navigating actions through the rodent parietal cortex. *Front. Hum. Neurosci.* **8**, 293 (2014).
12. Andersen, R. A. & Buneo, C. A. Intentional maps in posterior parietal cortex. *Annu. Rev. Neurosci.* **25**, 189–220 (2002).

Reviewer Reports on the First Revision:

Referee #1 (Remarks to the Author):

The authors have performed a large number of new experiments to address my concerns and the concerns of other reviewers. The analysis has also been performed in a more detailed way and the findings are more convincing.

In particular I appreciate the discovery of gap junctions between these SST4+ positive neurons as a potential mechanism for their synchronization. One question that remains for me is whether the proportion and strength of gap junctions between SST+ neurons in general is much lower. Can the authors discuss what has been found in the literature?

As far as the causal experiments, the authors have performed multiple experiments to address this issue but as they say the results are difficult to interpret. I do think the experiments that they propose for future studies are excellent, but given the large effort that they will take to perform are best relegated to a different paper.

I don't have any further concerns and feel that the paper makes a major contribution for understanding interneuron function in the cortex.

Referee #2 (Remarks to the Author):

The authors have done an exceptional job of responding to the reviewer comments. The revised manuscript offers substantially more insight regarding the role of cortical interneurons in behavior and now provides a more complete story. I believe this will add key new insights regarding how cell-types in the cortex support behavior. Moreover, it provides a much needed bridge between genetically and functionally defined cell types in the context of behavior.

Referee #3 (Remarks to the Author):

I applaud the author's considerable efforts to address my (and other reviewers) comments, including the additional experiments exploring the anatomy and connectivity of Sst44+ cells, the role of visual flow, and new analyses disambiguating movement related activity and error correction. It is my opinion that the alterations to the manuscript have added clarity. I continue to believe that the scientific question is of great interest and that the experimental design is superb. That said, I believe there are remaining issues that should be addressed in rebuttal or a revised manuscript.

Given differences in the size of Sst44+, Sst44-, and non-Sst populations one thing I found myself wondering while going through the manuscript this time was what peri-event plots (e.g. figure e5g-j, etc.) and corresponding statistics would look like if the groups were matched for total number of cells. Have the authors looked into this for some of the main findings in the manuscript?

The inactivation experiments presented in response to reviewer 1 comment 1 and echoed by reviewer 4 are critical in my opinion. I personally think the corresponding figures should be included in the manuscript since they make it clear that, although Sst44+ neurons have activity correlated with error corrections, they are not required for such behavior. Otherwise, I think readers may get the incorrect impression that these cells are mediating error corrections. I think this is especially important given the substantial discussion of a role for Sst44+ cells in providing a

learning signal vs. commanding corrections in the discussion.

Are there examples of Sst44+ activation when the mouse recovers from making a more extreme navigational error such as navigating to the end of an incorrect reward arm? If the animal recovers by navigating from the incorrect to correct reward arm there would potentially exist an entire period in which the error correction primarily involves linear movement rather than the distinct angular velocity and accelerations that are primarily observed. It would be interesting to see the activity of Sst44+ neurons under such conditions.

Minor points.

Figure 1f - marking approximate layer depths might be useful here.

Figure 2g,3g,3h, etc. - I think indications of significance would improve readability.

Extended data figure 3c - what bar plot?

I'm still struggling with some of the logic pertaining to separating coding for movement versus error correction. For example, on L228 (and other related analyses) the authors report that "Activity was intermediate for deviations accompanied by an intermediate correction (Extended Data Figure 10a). Thus, Sst44 cells are activated during course corrections rather than heading deviations alone." - An intermediate correction is still a course correction, no? If the activity is related to correction and not movement associated with the correction shouldn't it be similar regardless of the heading deviation required to correct?

The new experiments pertaining to Figure 4 and Ext. Figure 5 are really well executed with super interesting results relating to the connectivity of the Sst44+ network. I have only one minor question which is why did the authors elect to show the delays of only a subset of cell pairs in figure 4d?

Referee #4 (Remarks to the Author):

The authors did an excellent and very thorough job in addressing the comments, and the manuscript was substantially improved in revision. Multiple reviewers raised the lack of demonstration of a causal relationship between Sst44 cells and behavior, which was addressed but kept out of the current revision. The authors' took great effort to investigate this relationship, and ultimately could not detect a behavioral effect of bilateral inhibition on behavior in well-trained mice. The authors provide several potential and valid justifications in their responses to reviewers for why the manipulation may have had no effect on behavior. I think the authors should reconsider including this negative data as a supplement, with all the potential caveats (that are present in most optogenetic experiments), so that follow up studies in the field can take this into account when designing experiments aimed at understanding the role of Sst44 in behavior. Given that the authors have gone to great lengths to validate the virus and show that inhibition does increase PPC network activity (with the suggested caveats), I also think a discussion of the other more conceptual reasons for lack of effect remain interesting and worth reporting. Indeed, while not necessary, a simpler, chronic manipulation such as cell ablation/synaptic silencing throughout all early phases of learning may provided some support to the hypothesis put forth that Sst44 neurons are important for learning. Furthermore, since the authors have silicon probe recordings when the silencing was performed, it would be worthwhile to show the effects of their silencing on network activity in these mice, if the recordings were performed during the behavioral task.

Author Rebuttals to First Revision:

Referees' comments:

Referee #1 (Remarks to the Author):

The authors have performed a large number of new experiments to address my concerns and the concerns of other reviewers. The analysis has also been performed in a more detailed way and the findings are more convincing.

In particular I appreciate the discovery of gap junctions between these SST4+ positive neurons as a potential mechanism for their synchronization. One question that remains for me is whether the proportion and strength of gap junctions between SST+ neurons in general is much lower. Can the authors discuss what has been found in the literature?

This is an interesting question. The rate of coupling between Sst neurons varies in the literature, likely depending on the layer, transgenic line, and the type of measurements and analyses that were made. For example, Fanselow 2008 report 66% electrical coupling in layer 2/3 of somatosensory cortex using the GIN line. The GIN transgenic line labels a subset of layer 2/3 Martinotti (layer 1-projecting) neurons^{1,2}, similar to the subset labeled by our Sst44 enhancer (although the degree of overlap is unknown). Indeed, our measured rate of electrical coupling in Sst44 neurons (54%) roughly agrees with their result. We agree that knowing the overall rate of coupling in layer 2/3 Sst neurons would be useful as a comparison, but we were unable to find a measurement across the broader layer 2/3 Sst population. Other studies in layer 4 report a range of coupling (56-93%³⁻⁵). We expect these different rates are due to differences in slice preparation, the subtype of Sst neuron, and the cortical area in which they reside.

We added a sentence in the Discussion that expands on this point (line 335).

As far as the causal experiments, the authors have performed multiple experiments to address this issue but as they say the results are difficult to interpret. I do think the experiments that they propose for future studies are excellent, but given the large effort that they will take to perform are best relegated to a different paper.

I don't have any further concerns and feel that the paper makes a major contribution for understanding interneuron function in the cortex.

We appreciate the feedback and suggestions that helped us to improve the manuscript.

Referee #2 (Remarks to the Author):

The authors have done an exceptional job of responding to the reviewer comments. The revised manuscript offers substantially more insight regarding the role of cortical interneurons in behavior and now provides a more complete story. I believe this will add key new insights regarding how cell-types in the cortex support behavior. Moreover, it provides a much needed bridge between genetically and functionally defined cell types in the context of behavior.

Thank you for these positive comments and the helpful suggestions throughout the peer review process.

Referee #3 (Remarks to the Author):

I applaud the author's considerable efforts to address my (and other reviewers) comments, including the additional experiments exploring the anatomy and connectivity of Sst44+ cells, the role of visual flow, and new analyses disambiguating movement related activity and error correction. It is my opinion that the alterations to the manuscript have added clarity. I continue to believe that the scientific question is of great interest and that the experimental design is superb. That said, I believe there are remaining issues that should be addressed in rebuttal or a revised manuscript.

Given differences in the size of Sst44+, Sst44-, and non-Sst populations one thing I found myself wondering while going through the manuscript this time was what peri-event plots (e.g. figure e5g-j, etc.) and corresponding statistics

would look like if the groups were matched for total number of cells. Have the authors looked into this for some of the main findings in the manuscript?

Thank you for raising this point. To address the reviewer's concern, we sub-selected cells from each cell type population using the number of cells from the least represented population (which was Sst44 cells), and re-ran the same analysis. In order to fairly sample the other populations, which as the reviewer points out are more numerous (especially non-Sst cells), we repeated this analysis 100 times per population per session, and took the average result for each triggered event (thus, the number of triggered events is the same, which also makes statistical comparisons fair). We observed the same result (Referee 3 Figure 1 below) as with averaging the entire populations (Figure 5). Statistics were also similar (see legend).

Referee 3 Figure 1

Figure 5g-j analysis repeated by subsampling each cell type population with the number of cells in the least represented population (Sst44 cells). **g**, Activity and behavior averaged over large bursts of Sst44 cell activity (smoothed Sst44 cell activity > 0.4). **h**, Activity and behavior as the mouse entered the T-junction, split based on whether this was followed by a large ($> \pi/6$) or a small ($< \pi/12$) heading deviation. Left and right deviations were pooled after inverting behavior for left deviations. Selection criteria were evaluated at $+1.5$ s after entering the T-junction. **i**, Same as **h**, splitting trials with a high deviation based on whether the mouse corrected strongly (turning acceleration > 1 rad/s² in the opposite direction) or weakly (< 0.5 rad/s²). **j**, Triggered on turning accelerations, with high turning velocity (to match the turning profile across conditions), split by large and small deviations. Left and right turning accelerations were pooled after inverting behavior for right accelerations. Wilcoxon rank-sum test across trials, Sst44 cell activity solid vs dashed line (**h-j**): $p < 1e-10$. Wilcoxon signed-rank test across trials, Sst44 cell activity vs other cell types (**h-j**): $p < 1e-10$.

The inactivation experiments presented in response to reviewer 1 comment 1 and echoed by reviewer 4 are critical in my opinion. I personally think the corresponding figures should be included in the manuscript since they make it clear that, although Sst44+ neurons have activity correlated with error corrections, they are not required for such behavior. Otherwise, I think readers may get the incorrect impression that these cells are mediating error corrections. I think this is especially important given the substantial discussion of a role for Sst44+ cells in providing a learning signal vs. commanding corrections in the discussion.

We now include the optogenetic inhibition experiments in the paper (Extended Figure Data 12).

Are there examples of Sst44+ activation when the mouse recovers from making a more extreme navigational error such as navigating to the end of an incorrect reward arm? If the animal recovers by navigating from the incorrect to correct reward arm there would potentially exist an entire period in which the error correction primarily involves linear

movement rather than the distinct angular velocity and accelerations that are primarily observed. It would be interesting to see the activity of Sst44+ neurons under such conditions.

The reviewer proposes an interesting idea. We looked for events where the mouse enters the wrong arm and goes back into the correct arm. However, we were unable to find such events where the mouse did not turn at some point during this trajectory – for this event to take place, the mouse would have to walk backwards in a straight line, which is not only difficult for the mouse, but also unlikely, since the mouse prefers to walk or run forward. In these events, the mouse instead typically turns around in some fashion to get to the other arm. We agree that it is an interesting point to consider other movements beyond turning, and that this will be interesting to investigate in the future.

Minor points.

Figure 1f - marking approximate layer depths might be useful here.

We added approximate layer depths in this panel (now Extended Data Figure 1d). We agree this improves the interpretability of the figure.

Figure 2g,3g,3h, etc. - I think indications of significance would improve readability.

Thank you for this suggestion. We added indications of significance to Figure 2, 3, and other relevant figures.

Extended data figure 3c - what bar plot?

Thank you for pointing this out. This statement referred to the bar plot in panel b. We corrected the figure legend accordingly.

I'm still struggling with some of the logic pertaining to separating coding for movement versus error correction. For example, on L228 (and other related analyses) the authors report that "Activity was intermediate for deviations accompanied by an intermediate correction (Extended Data Figure 10a). Thus, Sst44 cells are activated during course corrections rather than heading deviations alone." - An intermediate correction is still a course correction, no? If the activity is related to correction and not movement associated with the correction shouldn't it be similar regardless of the heading deviation required to correct?

As the reviewer suggests, one could think of an error correction signal as a binary variable that is either on or off. However, one can also imagine a form of this signal that is graded, and that depends on the strength of the error and of the correction. Each form (binary or graded) of this signal may be suited for different functions – for example, it may be that in order to perform their function the inhibition provided by Sst44 cells must be tuned appropriately and that the appropriate tuning is related to the magnitude of the error and the correction. We therefore still find that the term "error correction signal" remains appropriate, given that we observe activity during error corrections – even though graded – but not under low error conditions or during non-corrective turning actions. We added a section in the Discussion to further highlight this point (line 346).

The new experiments pertaining to Figure 4 and Ext. Figure 5 are really well executed with super interesting results relating to the connectivity of the Sst44+ network. I have only one minor question which is why did the authors elect to show the delays of only a subset of cell pairs in figure 4d?

Thank you for highlighting this point for further clarification. We were concerned that it may be difficult to accurately estimate the onset of the membrane potential change in the follower cell (the cell in which we measure current flowing from the hyperpolarized driver cell) for the pairs with the weakest electrical coupling. We instead decided to focus on the pairs with the strongest coupling with the reasoning that the largest membrane potential changes in the follower cell would give the most accurate estimates of the delay. Including only the pairs with the strongest coupling will give an accurate estimate of the delay and does not pose any risk of creating an artifactually short delay. In contrast, the pairs with weaker coupling may lead us

to estimate a delay that is artifactually long simply due to signal to noise issues for detection of the signal. We therefore chose connections with the strongest signal (e.g. largest deviation, > 3 mV) to optimize the signal to noise ratio and minimize any over-estimation of the delay. We now include the description of this analysis in the Extended Data Figure 3 legend and in the Methods (line 968) and include the motivation for selecting strong connections for this analysis.

Referee #4 (Remarks to the Author):

The authors did an excellent and very thorough job in addressing the comments, and the manuscript was substantially improved in revision. Multiple reviewers raised the lack of demonstration of a causal relationship between Sst44 cells and behavior, which was addressed but kept out of the current revision. The authors' took great effort to investigate this relationship, and ultimately could not detect a behavioral effect of bilateral inhibition on behavior in well-trained mice. The authors provide several potential and valid justifications in their responses to reviewers for why the manipulation may have had no effect on behavior. I think the authors should reconsider including this negative data as a supplement, with all the potential caveats (that are present in most optogenetic experiments), so that follow up studies in the field can take this into account when designing experiments aimed at understanding the role of Sst44 in behavior. Given that the authors have gone to great lengths to validate the virus and show that inhibition does increase PPC network activity (with the suggested caveats), I also think a discussion of the other more conceptual reasons for lack of effect remain interesting and worth reporting. Indeed, while not necessary, a simpler, chronic manipulation such as cell ablation/synaptic silencing throughout all early phases of learning may provided some support to the hypothesis put forth that Sst44 neurons are important for learning. Furthermore, since the authors have silicon probe recordings when the silencing was performed, it would be worthwhile to show the effects of their silencing on network activity in these mice, if the recordings were performed during the behavioral task.

We thank the reviewer for the positive comments and the helpful suggestions in the previous round of review. We included the optogenetic inhibition experiments in the paper.